# AGENT2WORLD: A UNIFIED LLM-BASED MULTI-AGENT FRAMEWORK FOR SYMBOLIC WORLD-MODEL GENERATION

## ABSTRACT

Symbolic world models, which formally represent environment dynamics and constraints, are essential for model-based planning. While leveraging large language models (LLMs) to automatically generate these models from natural language has shown promise, existing approaches predominantly rely on scripted workflows that follow predetermined execution paths regardless of intermediate outcomes, often leading to inefficient computations and suboptimal solutions. In this paper, we propose AGENT2WORLD, a novel paradigm that employs autonomous tool-augmented LLM-based agents to generate symbolic world models adaptively. We further introduce AGENT2WORLD_{Multi}, a unified multi-agent framework with specialized agents: (*i*) A *Deep Researcher* agent performs knowledge synthesis by web searching to address specification gaps; (*ii*) A *Model Developer* agent implements executable world models; And (*iii*) a specialized *Testing Team* conducts evaluation-driven refinement via systematic unit testing and simulation-based validation. AGENT2WORLD_{Multi} demonstrates superior performance across three benchmarks spanning both Planning Domain Definition Language(PDDL) and executable code representations, achieving consistent state-of-the-art results through a single unified framework. By enabling proactive, knowledge-grounded world-model generation, this work opens new possibilities for AI systems that can reliably understand and formalize complex environments.

## 1 INTRODUCTION

In recent years, researchers have explored *symbolic world models*, a formal representation of an environment's dynamics and constraints, which is widely used in model-based planning (Guan et al., 2023; LeCun, 2022; Craik, 1967). The task of *symbolic world-model generation* involves automatically synthesizing these models from natural language descriptions, eliminating the need for domain experts to manually design and specify complex rules and dynamics. Large language models (LLMs) (Guo et al., 2025; Zhao et al., 2023; Bai et al., 2023) have made this automation increasingly possible by combining two key capabilities: commonsense knowledge about how the world works, and code generation abilities that formalize this knowledge into executable representations.

As illustrated in Figure 1, prior work in this domain largely follows two paradigms: (i) *direct generation* of symbolic world models, and (ii) *scripted workflows* that couple generation with iterative verification and repair. Early exemplars of the latter include Guan et al. (2023) and Hu et al. (2025a), using LLMs to produce Planning Domain Definition Language(PDDL)-based world models. Furthermore, GIF-MCTS (Dainese et al., 2024) combines a code executor with trajectories collected in a real environment to furnish feedback, driving a *generate–fix–improve* loop that progressively refines the generated MuJoCo-style code world models. While scripted workflows achieve better results than direct generation, they suffer from fundamental limitations: (i) *Passive and rigid execution*: these methods reactively respond to validation failures through predetermined repair sequences, leading to unnecessary computations when simpler solutions exist or inadequate exploration when complex problems require adaptive strategies; (ii) *Knowledge isolation*: Most approaches rely solely on the LLM's internal knowledge, lacking mechanisms to access external information when specifications are incomplete or ambiguous. While Guan et al. (2023) leverage human feedback as external knowledge, the labor-intensive nature limits their scalability for large-

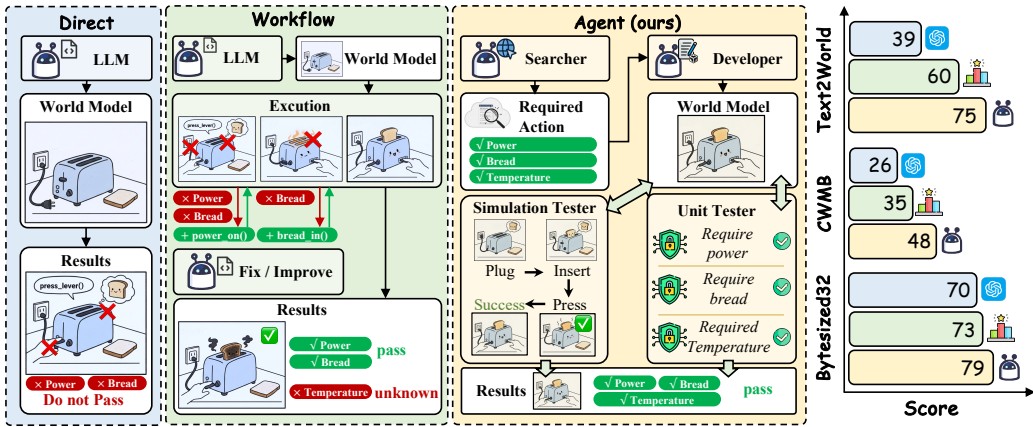

Figure 1: Comparison of AGENT2WORLD and previous world-model generation paradigms.

scale applications. (iii) *Representation fragmentation*: existing approaches typically target either PDDL-based symbolic representations or executable code exclusively, requiring separate systems and workflows for different output formats.

We propose AGENT2WORLD, a new paradigm for symbolic world-model generation that leverages tool-augmented, autonomous LLM-based agents, which plan and call tools adaptively. The key advantages lie in: (i) *Proactive and adaptive execution.* Rather than passively fixing errors through rigid sequences, AGENT2WORLD proactively gathers information and dynamically adjusts strategies based on intermediate feedback, automatically deciding when to terminate and which tools to use for maximum efficiency. (ii) *Scalable external knowledge integration.* Unlike knowledge-isolated approaches or labor-intensive human-in-the-loop methods, AGENT2WORLD incorporate web search as first-class tools to automatically fill specification gaps and enforce commonsense regularities, minimizing LLM hallucination (Huang et al., 2025). (iii) *Unified cross-representation framework.* While prior systems suffer from representation fragmentation, AGENT2WORLD seamlessly handles both PDDL-based and code-based models through lightweight tool adapters, enabling the same agentic framework to work across different symbolic representations.

Specifically, AGENT2WORLD$_{Single}$ employs a single ReAct-style agent (Yao et al., 2023) to invoke all available tools (code sandbox, web search, etc.). However, due to the extensive array of available tools and the token-heavy nature of world models (Wang et al., 2023), the single-agent approach faces context length limitations and tool coordination challenges. Furthermore, we propose AGENT2WORLD$_{Multi}$ to address these issues, which features role specialization and tool partitioning. Specifically, AGENT2WORLD$_{Multi}$ is structured as a three-stage pipeline as shown in Figure 2: (i) *Knowledge Synthesis*: A Deep Researcher agent proactively identifies knowledge gaps in ambiguous specifications and systematically gathers authoritative information via web search; (ii) *World Model Generation*: a Model Developer agent generates concrete implementations with iterative code execution and refinement; (iii) *Evaluation-Driven Refinement*: specialized testing agents evaluate and diagnose the generated models through both simulation for holistic behavioral fidelity and unit testing for component-level correctness, providing feedback to guide iterative refinement.

Our experiments on three benchmarks validate our approach with insights into the fundamental challenges of world model generation: (i) on *Text2World* (Hu et al., 2025a) (PDDL-based world models), AGENT2WORLD$_{Multi}$ achieves both high semantic fidelity and syntactic correctness (75.4 macro-averaged F1 with 93.1% executability) (ii) on *CWMB* (Dainese et al., 2024) (code-based world models), it establishes new state-of-the-art with 54.4% predictive accuracy and 48.1% normalized return. Notably, while predictive accuracy improvements are modest, the normalized return gains are dramatic (+13.2% over GIF-MCTS), revealing that our *Evaluation-Driven Refinement* (§3.3) addresses the critical gap between accurate next-state prediction and effective model-based planning; (iii) on *ByteSized32* (Wang et al., 2023) (reasoning-heavy text games), it demonstrates superior performance across technical validity and physical reality alignment (+28.4% alignment score). This validates our hypothesis that external *Knowledge Synthesis* (§3.1) is essential for grounding world models in reliable commonsense knowledge, especially for tasks requiring complex reasoning about everyday physical constraints.

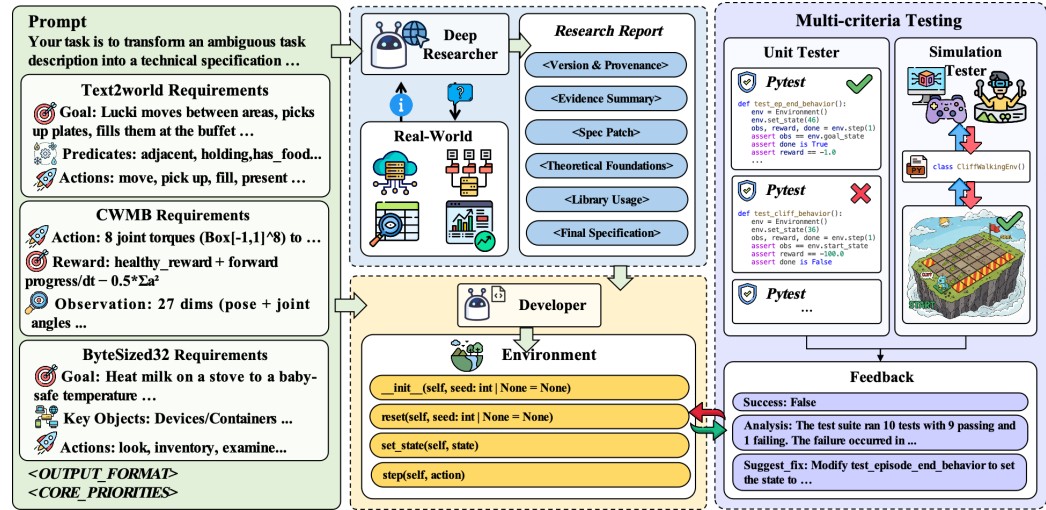

Figure 2: Overall Pipeline of AGENT2WORLD.

## 2 PRELIMINARY

### 2.1 PROBLEM DEFINITION

We investigate the problem of *symbolic world-model generation* from natural language. Given a textual description $x$, the objective is to synthesize an *executable* program $\mathcal{M}_F$ that faithfully captures the dynamics and constraints of the environment. Such a program may take various forms, for instance, a specification in the Planning Domain Definition Language (PDDL) (Hu et al., 2025a; McDermott et al., 1998) or an implementation in Python (Dainese et al., 2024; Wang et al., 2023). Formally, an environment is defined by a set of predicates $\mathcal{P}$, a set of actions $\mathcal{A}$, and a transition function $T : S \times \mathcal{A} \rightarrow S$, where $S$ denotes the set of possible states. Semantically, $\mathcal{M}_F$ encodes these components to represent the environment in an executable manner. We therefore define the task as a mapping $P(x) = \mathcal{M}_F$ where $\mathcal{M}_F = \langle \mathcal{P}, \mathcal{A}, T \rangle$. where $P$ is a synthesis procedure that generates the world-model program from natural language input $x$.

### 2.2 AUTONOMOUS AGENT

We consider an *autonomous agent* that *interleaves reasoning with tool use* (Qin et al., 2023; Yao et al., 2023). Concretely, the agent firstly produces reasoning traces and issues tool calls in an alternating loop. Let $\mathcal{T} = \{t_1, \ldots, t_m\}$ denote a set of callable tools. At discrete time $k$, the agent maintains a history $h_k = (x, o_{\leq k}, a_{<k}, r_{<k})$ consisting of the task description $x$, observations $o$, past actions $a$, and internal reasoning traces $r$. A *policy* $\pi_\theta$ maps histories to the next reasoning trace and action: $(r_k, a_k) \sim \pi_\theta(\cdot \mid h_k), \quad a_k \in \mathcal{T}$. In practice, $\pi_\theta$ is implemented by an LLM.

## 3 METHODOLOGY

As is shown in Figure 2, AGENT2WORLD*Multi* unfolds in three stages: *(i)* **Knowledge Synthesis** (§3.1): As outlined in Section 1, a key challenge in symbolic world-model construction arises from incomplete descriptions. For example, commonsense knowledge may be missing both from LLMs and from the given specifications. To address this limitation, we employ a *deep researcher agent* that interacts with external resources such as the internet or structured databases, thereby enriching the specification and producing an intermediate representation. *(ii)* **World Model Generation** (§3.2): At this stage, a *developer agent* equipped with a code execution tool constructs the symbolic world model. The process is iteratively refined based on execution feedback, ensuring both correctness and executability. *(iii)* **Evaluation-Driven Refinement** (§3.3): We enhance the semantic fidelity by designing two complementary *test agents*: one that generates unit tests to validate functional

behavior, and another that simulates downstream usage to evaluate performance through trajectory-based testing. We also provide a pseudo code in Algorithm 1.

### 3.1 STAGE I: KNOWLEDGE SYNTHESIS

We introduce a *Deep Researcher* agent designed to gather background knowledge and fill in missing details that are not explicitly provided in the world model description. By leveraging external information sources, this agent not only compensates for potential knowledge gaps inherent in large language models but also enhances the factual reliability of world model descriptions. Equipped with web search and retrieval tools, it iteratively retrieves the knowledge required for world model construction from the internet and ultimately outputs a structured intermediate representation with the missing information completed.

### 3.2 STAGE II: WORLD MODEL GENERATION

After obtaining the comprehensive world-model description from the previous stage, the *Model Developer* takes this as input and generates a concrete implementation of the world model in the required formalism (e.g., PDDL or executable code). To support iterative refinement, the *Model Developer* is equipped with a sandboxed code-execution tool, enabling it to test and debug implementations in multiple rounds until the code is functional and consistent with the specification.

### 3.3 STAGE III: EVALUATION-DRIVEN REFINEMENT

A key component of our approach is the refinement of a bug-free, code-based world model. Unlike prior works that rely on annotated gold trajectories (Dainese et al., 2024) or human feedback (Guan et al., 2023), our method is fully autonomous and does not require manual labels. More specifically, we introduce a two-agent *Testing Team* to evaluate and diagnose the generated models: *(i)* The *Simulation Tester* evaluates the world model in a play-testing manner by attempting to perform tasks, explore actions, and issue queries within the environment. Specifically, it interacts with the environment in a ReAct-style (Yao et al., 2023) loop to collect trajectories for subsequent behavior and reward analysis, which uncovers execution-time failures such as unreachable goals, missing preconditions, or inconsistent state updates. *(ii)* The *Unit Tester* complements play-testing with systematic, programmatic verification. It automatically generates Pytest-style unit tests targeting the predicates, actions, and invariants specified in the world-model descriptions.

Together, these agents produce a detailed *test report* that assesses the quality of the generated world model and provides fine-grained diagnostic signals on correctness, coverage, logical consistency, and compliance with physical requirements. This report is fed back to the *Model Developer*; if inconsistencies or failures are detected, the Model Developer revises the implementation, triggering another evaluation round by both testers. This loop continues until all checks are satisfied or a predefined convergence criterion is reached.

## 4 EXPERIMENTS

In this section, we first describe the baselines (§ 4.1) and implementation details (§ 4.2), and then present experiments on three benchmarks: *(i) **Text2World*** (Hu et al., 2025a) (§ 4.3): A PDDL-centric benchmark for text-to-symbolic world modelling. *(ii) **Code World Models Benchmark (CWMB)*** (Dainese et al., 2024) (§ 4.4): A code-based world-model benchmark comprising MuJoCo-style environments, designed to assess *predictive correctness* and *downstream control utility* under both discrete and continuous action settings. *(iii) **ByteSized32*** (Wang et al., 2023) (§ 4.5): A suite of reasoning-heavy text games requiring executable Python environments. A side-by-side comparison and a detailed metric explanation are shown in Appendix C.

### 4.1 BASELINES

We compare AGENT2WORLD$_{Multi}$ against the following methods:

*(i) **Direct Generation (Direct)***: Single-shot generation of the symbolic world model without tool use, external retrieval, or feedback. *(ii) **Agent2World**$_{single}$*: A single agent closes the loop by in-

voking code execution/validators/web search tools for self-repair and information synthesis, without multi-agent specialization. *(iii) Text2World (EC=k) (Hu et al., 2025a)*: directly using large language models to generate PDDL-based world model and iteratively repairing with planner/-validator signals, where EC denotes the error-correction budget. *(iv) WorldCoder (Tang et al., 2024)*: A plan–code–execute–repair search that scores and iteratively improves candidate programs using simulator/planner signals to select runnable hypotheses. *(v) GIF-MCTS (Dainese et al., 2024)*: A macro-action MCTS that orchestrates Generate/Improve/Fix steps, guided by unit tests and trajectory-based feedback for code world-model synthesis. We also introduced an enhanced version of GIF-MCTS where a Deep Researcher agent gathers additional research data. *(vi) Byte-Sized32 baseline (Wang et al., 2023)*. The reference pipeline introduced by Wang et al. (2023). In order to avoid metric leakage, we do not use the official checker's evaluation signals. *(vii) Best-of-N (Stiennon et al., 2020; Yu et al., 2025)*: A method that performs reasoning over multiple samples and selects the best result. *(viii) Self-consistency (Wang et al., 2022)*: A multi-sample reasoning method that votes over the results to improve consistency in decision-making.

## 4.2 Implementation Details

We employ the OpenAI GPT-4.1-mini model via the official API and Llama-3.1-8b-instruct via the officHuggingfaceface repo. We set the decoding temperature to 0 and top_p to 1 for deterministic reproducibility. All agents operate within a ReAct (Yao et al., 2023) framework, following a "think → act (tool) → observe" loop for a maximum of 10 steps. The *Deep Researcher* agent utilizes the Serper API for web searching. We blocked some websites to ensure experimental integrity and prevent information leakage [1] Regarding the configuration of refinement turns, we set Text2World and ByteSized32 to 2 iterations and CWMB to 3 iterations based on the complexity of environments. For automated evaluation on the ByteSized32 benchmark, we leverage GPT-4o (Hurst et al., 2024) as the LLM evaluator. All experiments with gpt-4.1-mini are conducted on a CPU server without GPU acceleration. The experiments with llama-3.1-8b-instruct (including training and inference) are conducted with an 8xA100 server. The prompt examples could be found at Appendix F.

## 4.3 Text2World

Table 1: Benchmark results on Text2World (Hu et al., 2025a). Following the reporting convention in Text2World, all metrics are presented as percentage scores (%).

| Methods | Executability (↑) | Similarity (↑) | Component-wise F1 (↑) | | | | $F1_{AVG}$ (↑) |
| --- | --- | --- | --- | --- | --- | --- | --- |
| | | | $F1_{PRED}$ | $F1_{PARAM}$ | $F1_{PRECOND}$ | $F1_{EFF}$ | |
| Text2World$_{EC=3}$ | 78.2 | 81.1 | 73.4 | 64.5 | 49.3 | 53.3 | 60.1 |
| Direct Generation | 45.5 | **82.8** | 45.0 | 40.3 | 33.9 | 34.9 | 38.5 |
| Agent2World$_{Single}$ | 79.2 | 82.5 | 76.5 | 75.8 | 60.1 | 66.0 | 69.6 |
| Agent2World$_{Multi}$ | **93.1** | 81.0 | **87.2** | **82.3** | **63.7** | **68.2** | **75.4** |

We evaluate the Planning Domain Definition Language (PDDL)-based world model generation of Agent2World on Text2World Hu et al. (2025a), which comprises 103 PDDL domains paired with natural language descriptions. The evaluation metrics are: *(i)* **Executability**: whether the generated PDDL can be parsed and validated; *(ii)* **Structural Similarity**: the normalized Levenshtein similarity; *(iii)* **Component-wise F1**: the macro-averaged F1 of predicates ($F1_{PRED}$) and action components, including parameters ($F1_{PARAM}$), preconditions ($F1_{PRECOND}$), and effects ($F1_{EFF}$).

**Results.** We can draw several conclusions from Table 1: *(i) Direct Generation* attains the highest Similarity (82.8) yet performs poorly on executability (45.5) and all component-wise F1s, underscoring that surface-level textual overlap is a weak proxy for runnable, semantically correct PDDL. *(ii)* While agent-based methods achieve executability comparable to the reference solution (e.g., Agent2World$_{Single}$ with 79.2 vs. Text2World$_{EC=3}$ with 78.2), they exhibit substantial gaps in F1 scores (Agent2World$_{Single}$: 69.6 vs. Text2World$_{EC=3}$: 60.1). This suggests that while integrating validators for iterative correction can significantly improve syntactic validity, the semantic utility

---

[1]For example, the original Text2World and ByteSized32 huggingface pages, CWMB source code, OpenAI-Gym code repository are blocked.

of the generated world models remains limited without comprehensive knowledge synthesis. *(iii)* AGENT2WORLD*Multi* achieves both the highest executability (+14.9 points over Text2World$_{EC=3}$) and superior F1 performance (+15.3 points), demonstrating the synergistic benefits of multi-agent specialization. These patterns align with our design philosophy: knowledge synthesis combined with evaluation-driven refinement steers the model to recover the correct predicate inventory and logical gating constraints, producing domains that are both *syntactically valid* and *semantically solvable*, even when surface-level representations diverge from reference implementations.

## 4.4 CODE WORLD MODELS BENCHMARK (CWMB)

Table 2: Benchmark results on CWMB. † We adopted the official implementation of GIF-MCTS (Dainese et al., 2024) and their reimplementation of WorldCoder (Tang et al., 2024). GIF-MCTS* denotes the enhanced version that we obtain by connecting the deep research agent with the original GIF-MCTS pipeline.

| Method | Discrete Action Space | | Continuous Action Space | | Overall | |
|---|---|---|---|---|---|---|
| | Accuracy (↑) | $\mathcal{R}$ (↑) | Accuracy (↑) | $\mathcal{R}$ (↑) | Accuracy (↑) | $\mathcal{R}$ (↑) |
| `gpt-4.1-mini` | | | | | | |
| WorldCoder† | 0.9024 | 0.5399 | 0.3303 | 0.2097 | 0.5210 | 0.3197 |
| GIF-MCTS† | 0.9136 | 0.6842 | 0.2748 | 0.1811 | 0.4877 | 0.3488 |
| GIF-MCTS* | 0.8876 | 0.7955 | 0.3030 | 0.1495 | 0.4979 | 0.3649 |
| Direct Generation | 0.7321 | 0.4527 | 0.3038 | 0.1666 | 0.4466 | 0.2620 |
| Best-of-N | 0.7488 | 0.6012 | 0.2642 | 0.1970 | 0.4257 | 0.3317 |
| Self-consistency | 0.7870 | 0.4912 | 0.2479 | 0.2158 | 0.4276 | 0.3076 |
| AGENT2WORLD*Single* | 0.7897 | 0.5418 | 0.1917 | 0.2420 | 0.3911 | 0.3419 |
| AGENT2WORLD*Multi* | **0.9174** | **0.8333** | **0.3575** | **0.3050** | **0.5441** | **0.4811** |
| No *Deep Researcher* | 0.8794 | 0.4407 | 0.3404 | 0.2201 | 0.5201 | 0.2936 |
| `llama-3.1-8b-instruct` | | | | | | |
| WorldCoder† | 0.5192 | 0.2779 | 0.1173 | 0.1073 | 0.2513 | 0.1642 |
| GIF-MCTS† | 0.5983 | 0.3357 | 0.1332 | 0.1427 | 0.2883 | 0.2070 |
| Best-of-N | 0.2236 | 0.0004 | 0.1544 | 0.1404 | 0.1775 | 0.0937 |
| Self-Consistency | 0.4626 | 0.1667 | 0.0000 | 0.0036 | 0.1542 | 0.0580 |
| AGENT2WORLD*Multi* | 0.5797 | 0.2998 | 0.1826 | 0.1945 | 0.3150 | 0.2296 |
| AGENT2WORLD*Multi* (SFT) | **0.6457** | **0.5257** | **0.1811** | **0.2105** | **0.3360** | **0.3156** |

The CWMB (Dainese et al., 2024) evaluates the ability of generated executable code to serve as faithful world models across 18 MuJoCo-style environments. It measures both the predictive accuracy of next-state dynamics and the normalized return ($\mathcal{R}$) when the model is used by a planner, where $\mathcal{R}$ reflects the gap between a random policy and an oracle planner with the true environment. This setup ensures CWMB jointly assesses the correctness of the simulation code and its practical utility for downstream control.

**Results.** Table 5 reveals several key findings. *(i)* All methods demonstrate superior performance in discrete spaces compared to continuous settings, reflecting the inherent difficulty of modeling continuous dynamics. *(ii)* Workflow-based approaches consistently outperform both *Direct Generation* and AGENT2WORLD*Single*, indicating that LLMs' native world model generation capabilities are limited and require expert-designed iterative refinement to achieve competitive performance. *(iii)* AGENT2WORLD*Multi* establishes new state-of-the-art results, surpassing the previous best method GIF-MCTS by +0.132 $\mathcal{R}$ points in overall normalized return. Notably, while other methods achieve comparable predictive accuracy (e.g., 0.917 vs 0.914 on discrete settings), our simulation-based testing framework significantly enhances the downstream utility of generated world models, demonstrating that accurate next-state prediction alone is insufficient for effective model-based planning.

## 4.5 BYTESIZED32

The ByteSized32 (Wang et al., 2023) benchmark consists of 32 reasoning-heavy text games, each implemented as an executable Python environment. Models are required to generate runnable game code that captures task-specific dynamics, objects, and rules, allowing direct interaction and evaluation. The benchmark evaluates four dimensions: **Technical Validity** (whether the code runs),

Table 3: Technical Validity and Physical Reality Alignment scores on ByteSized32

| Method | Technical Validity ($\uparrow$) | | | Physical Reality Alignment ($\uparrow$) |
|---|---|---|---|---|
| | Game Init. | Possible Actions | Runnable Game | Alignment Score |
| ByteSized32$_{0\text{-shot}}$ | 0.9792 | 0.9375 | 0.7292 | 0.0600 |
| ByteSized32$_{1\text{-shot}}$ | 0.9792 | 0.8958 | 0.7500 | 0.1748 |
| Direct Generation | 0.9271 | 0.8854 | 0.7604 | 0.0000 |
| AGENT2WORLD$_{Single}$ | 0.9792 | 0.9375 | 0.7708 | 0.1920 |
| AGENT2WORLD$_{Multi}$ | **0.9896** | **0.9583** | **0.8958** | **0.4768** |

Table 4: Specification Compliance and Winnability scores on ByteSized32

| Method | Specification Compliance ($\uparrow$) | | | Winnability ($\uparrow$) |
|---|---|---|---|---|
| | Critical Objects | Critical Actions | Distractors | Winnable Game |
| ByteSized32$_{0\text{-shot}}$ | 0.9375 | 0.9375 | 0.8750 | 0.0625 |
| ByteSized32$_{1\text{-shot}}$ | **1.0000** | 0.9375 | **0.9375** | 0.1354 |
| Direct Generation | **1.0000** | 0.9375 | **0.9375** | 0.1354 |
| AGENT2WORLD$_{Single}$ | **1.0000** | **1.0000** | 0.8438 | 0.1354 |
| AGENT2WORLD$_{Multi}$ | **1.0000** | 0.9688 | 0.8750 | **0.1458** |

**Specification Compliance** (whether all required elements are present), **Winnability** (whether the task can be completed), and **Physical Reality Alignment** (whether the environment dynamics are consistent with commonsense constraints). This setting emphasizes both logical fidelity and practical executability, making it a stringent testbed for language models as world-model generators.

**Results.** Several conclusions could be drawn from Table 3 and 4: *(i)* The official reference pipeline outperforms direct generation with in-context learning and shows comparable performance to AGENT2WORLD$_{Single}$ on certain metrics. *(ii)* The AGENT2WORLD$_{Single}$ baseline shows moderate gains in game solvability, yet its alignment with physical reality is slightly weaker. *(iii)* AGENT2WORLD$_{Multi}$ outperforms both baselines across almost all dimensions, especially +0.2848 physical alignment score, which stems from *Deep Researcher* agent synthesizing the commonsense knowledge required for reasoning-heavy games.

## 5 ANALYSIS

### 5.1 ABLATION STUDY

Ablations from Figure 3 and Table 9 clarify where the lift comes from: *(i)* Removing the *Unit Tester* causes the most significant performance drop, with accuracy declining by 0.3008 and reward $\mathcal{R}$ by 0.4470 in discrete action spaces. *(ii)* The *Deep Researcher* primarily impacts reward $\mathcal{R}$ quality, showing a substantial decrease of 0.3926 for discrete spaces when removed. *(iii)* Although the removal of *Simulation Tester* results in the smallest overall performance drops, the reward $\mathcal{R}$ decreases by 0.2615 and 0.1473 for discrete space and continuous space, respectively. These results collectively validate our design choices and highlight the complementary nature of the three components.

### 5.2 PAIR-WISE EVALUATION

To quantify the effect of AGENT2WORLD$_{Multi}$ and the refinement procedure, we perform instance-level pairwise comparisons, recording a Win–Tie–Loss (WTL) outcome according to the benchmarks' primary metric: *(i)* F1$_{\text{AVG}}$ for *Text2World*; *(ii)* $\mathcal{R}$ for *CWMB*; and *(iii)* the mean of all official metrics for *ByteSized32*. As shown in Figure 4, the *left* panel contrasts the final-turn model with its first-turn counterpart. Refinement yields consistent gains on *CWMB* and *ByteSized32* (68.8% and 93% wins, respectively; no losses), largely preserves performance on *Text2World* while delivering occasional improvements (14% wins vs. 7% losses). The *right* panel compares AGENT2WORLD$_{Multi}$ against previous state-of-the-art systems. AGENT2WORLD$_{Multi}$ attains clear advantages across all three benchmarks, most notably on *ByteSized32* (87% wins) and *CWMB* (66.7% wins).

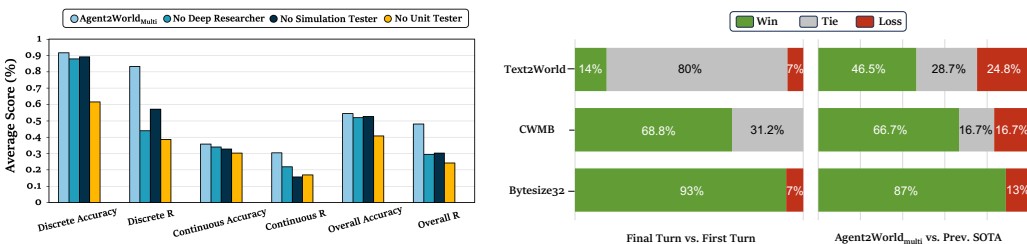

Figure 3: Ablation Study on CWMB.

Figure 4: Pair-wise Win–Tie–Loss analysis.

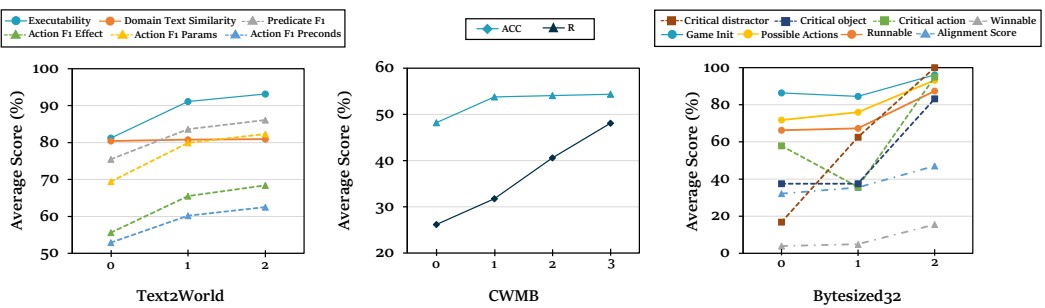

Figure 5: Performance evolution of iterative refinement (§ 3.3.)

## 5.3 FEEDBACK ITERATION

To understand the dynamics of performance improvement through iterative feedback, we analyze how model performance evolves as the number of testing team feedback iterations increases. Figure 5 illustrates the relationship between feedback iteration count and model performance across the evaluation benchmarks. The results reveal several key patterns: *(i)* Text2World shows rapid initial improvements. Notably, execution-based metrics improve substantially while similarity measures remain stable, suggesting that refinement enhances functional correctness rather than surface-level similarities. *(ii)* CWMB demonstrates sustained improvement across iterations, reflecting the compound complexity of physics simulation where numerical accuracy and dynamics must be jointly optimized. *(iii)* ByteSized32 exhibits the most dramatic gains, with several metrics showing step-function improvements that reflect the discrete nature of game logic debugging.

## 5.4 MANUAL ERROR ANALYSIS

We conducted a manual error analysis to examine the evolution of error patterns throughout the refinement process of AGENT2WORLD$_{Multi}$. Taken CWMB in Figure 6 as an example, the initial turn predominantly exhibits superficial errors like *signature-arity mismatches* and *representation mismatches*, stemming from inadequate adherence to world model specifications. Throughout the iterative refinement process, these surface-level inconsistencies are systematically eliminated, with the error landscape shifting toward more fundamental *dynamics mismatches* in later iterations. This pattern demonstrates remarkable consistency across all benchmarks: refinement consistently shifts the error distribution from form-oriented problems (*syntax*, *arity*) to substance-oriented challenges (*dynamics*, *state transitions*) as shown in Figure 7 and 8. The systematic progression from surface to substance reflects the hierarchical nature of world model correctness and validates our multi-turn refinement architecture. We also provide the detailed proportion of each error type in Appendix E.

## 5.5 MULTI-AGENT VS. SINGLE-AGENT ARCHITECTURE ANALYSIS

To quantify the cost of multi-agent specialization, we analyze the token consumption reported in Appendix J. Compared to AGENT2WORLD$_{Single}$, AGENT2WORLD$_{Multi}$ incurs a higher computational cost during the generation phase. This increase stems from the proactive testing loop, where agents autonomously generate unit tests and simulation trajectories to diagnose errors. However,

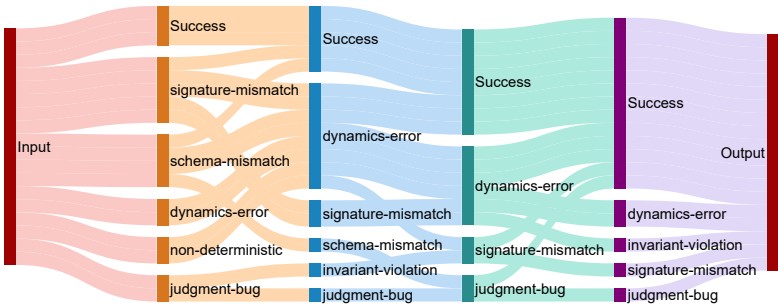

Figure 6: Error distribution on CWMB of evaluation-driven refinement. Due to page limit, the error distribution of other benchmarks is presented in Appendix E.

this additional cost represents a finite **upfront investment** rather than a recurring inefficiency, which is incurred only once during the synthesis of the world model. In exchange, the framework secures a permanent **performance gain** in the quality of the generated artifact (e.g., raising the Normalized Return $\mathcal{R}$ from 0.3419 to 0.4811 on CWMB).

### 5.6 LEARNING WORLD-MODEL AGENTS FROM TESTING-TEAM FEEDBACK

**Method.** So far, AGENT2WORLD$_{Multi}$ has been described as an inference-time multi-agent workflow in which a frozen backbone LLM plays the roles. However, the interaction between the Model Developer and the Testing Team naturally defines a Markov decision process (MDP) and therefore an **agent-in-the-loop** training environment for world-model agents. We model the Model Developer as an LLM-based agent operating in an MDP $\mathcal{M}_{MD} = (\mathcal{S}, \mathcal{A}, \mathcal{T}, \mathcal{R}, \gamma)$, where a state $s_t \in \mathcal{S}$ encodes the natural-language specification or diagnostics produced by testing team; an action $a_t \in \mathcal{A}$ corresponds to emitting a new world-model implementation or a code patch; and the transition $\mathcal{T}$ is realized by executing the candidate world model in the sandbox and re-invoking the Testing Team, which produces updated diagnostics that are appended to form $s_{t+1}$. In this view, the Testing Team acts as a *reward generator*: the Unit Tester contributes pass/fail statistics from automatically generated unit tests, the Simulation Tester contributes downstream control performance, and these signals are aggregated into a scalar reward $\mathcal{R}(s_t, a_t)$ that reflects both local predictive correctness and task-level utility of the world model.

**Experiments.** We conduct an experiment on CWMB where we train the Model Developer agent using Testing-Team feedback. We first curate a dataset with 500 tasks that includes both PDDL-style and code-style specifications paired with gold world models. We then construct a supervised fine-tuning (SFT) dataset by rolling out AGENT2WORLD$_{Multi}$ on these tasks, filtering trajectories based on the reward signal, treating the remaining interaction traces as demonstrations, and finally fine-tuning the same underlying `llama-3.1-8b-instruct` model. At test time, we freeze the Testing Team and evaluate AGENT2WORLD$_{Multi}$ (SFT) under the same protocol as AGENT2WORLD$_{Multi}$ on held-out CWMB tasks. As summarized in Table 5, starting from the base `llama-3.1-8b-instruct`, the untrained AGENT2WORLD$_{Multi}$ achieves an overall normalized return of 0.3156 compared to 0.2296 for the raw backbone These gains indicate that the Model Developer has learned to internalize the preferences expressed by the Testing Team and to directly generate higher-utility world models, effectively turning AGENT2WORLD from a static evaluation pipeline into a reusable training substrate for world-model agents.

## 6 RELATED WORK

**World Models.** World models are widely applied in reinforcement learning, robotics, and autonomous systems for planning, etc (Hao et al., 2023; Ha & Schmidhuber, 2018). Generally, there are two types of world models: (i) *neural world models*, which employ neural networks to approximate dynamics (Ha & Schmidhuber, 2018; Hafner et al., 2019), and (ii) *symbolic world models*, which are represented using formal languages such as the Planning Domain Definition Language

Table 5: Performance of the Model Developer on CWMB before and after supervised fine-tuning of its `llama-3.1-8b-instruct` backbone.

| Method | Discrete Action Space | | Continuous Action Space | | Overall | |
|---|---|---|---|---|---|---|
| | Accuracy ($\uparrow$) | $\mathcal{R}$ ($\uparrow$) | Accuracy ($\uparrow$) | $\mathcal{R}$ ($\uparrow$) | Accuracy ($\uparrow$) | $\mathcal{R}$ ($\uparrow$) |
| AGENT2WORLD$_{Multi}$ | 0.5797 | 0.2998 | 0.1826 | 0.1945 | 0.3150 | 0.2296 |
| AGENT2WORLD$_{Multi}$ (SFT) | 0.6457 | 0.5257 | 0.1811 | 0.2105 | 0.3360 | 0.3156 |
| $\Delta$ Delta | 0.0660 | 0.2259 | $-0.0015$ | 0.0160 | 0.0210 | 0.0860 |

(PDDL) or code-based implementations. In this paper, we investigate *symbolic world-model generation*, which involves transforming world model descriptions into formal representations that can subsequently be utilized for applications such as model-based planning (Guan et al., 2023; Dainese et al., 2024), dataset construction (Hu et al., 2025b), and so on. Most prior work follows a *draft-repair* workflow where the model first proposes an initial implementation, then gradually refines under closed-loop diagnosis from some kind of feedback. The feedback mechanisms can vary significantly across different approaches. For instance, Guan et al. (2023) employs human feedback to provide corrective signals and perform iterative modifications. Other works, such as Hu et al. (2025a) and Tang et al. (2024), utilize executors and validators to generate feedback. GIF-MCTS (Dainese et al., 2024) leverages gold experiences as feedback signals. Compared to these scripted workflows, the AGENT2WORLD paradigm introduced in this paper can more flexibly adjust subsequent strategies based on feedback signals. Furthermore, benefiting from the scalability of AGENT2WORLD, we incorporate additional external feedback mechanisms such as web search to supplement the intrinsic knowledge limitations of the underlying models. A side-by-side of AGENT2WORLD and existing methods can be found in Appendix B.1.

**Large Language Model-based Agent.** In recent years, benefiting from the rapid advancement of large language models (LLMs), LLM-based agents have emerged as particularly powerful systems that accept natural language user intentions as input and achieve goal states through planning and sequential decision-making (Hu et al., 2024; Yao et al., 2023; Schick et al., 2023). These autonomous agents have demonstrated remarkable effectiveness across diverse applications, ranging from web navigation (Yao et al., 2022; Nakano et al., 2021; Wang et al., 2025) and software development (Qian et al., 2023; Hong et al., 2024) to scientific research (Lu et al., 2024; Chen et al., 2025) and robotic planning (Huang et al., 2022). Prominent examples of such systems include ReAct (Yao et al., 2023), which synergizes reasoning and acting in language models by interleaving thought, action, and observation steps; Existing research has explored how world models can assist LLM-based agents in planning, such as RAP (Hao et al., 2023), which uses Monte Carlo Tree Search with world models for improved reasoning, and Guan et al. (2023), which leverages pre-trained LLMs to construct world models for model-based task planning. These approaches primarily focus on *utilizing* existing world models rather than *generating* them. Similarly, recent work has investigated how world models can enhance training of LLM-based agents, as demonstrated by AgentGen (Hu et al., 2025b) and Kimi-K2 (Team et al., 2025). To our best knowledge, our work represents the first systematic investigation into using autonomous agents for world model generation, bridging the gap between agent-based problem solving and symbolic world modeling.

## 7 CONCLUSION

We introduced AGENT2WORLD$_{Multi}$, a unified multi-agent framework that employs autonomous LLM-based agents to generate symbolic world models across both PDDL and executable code representations. The framework operates through three specialized stages: knowledge synthesis via web search, world model development with iterative refinement, and evaluation-driven testing through unit tests and simulation. Experimental results demonstrate consistent state-of-the-art performance across three world-model generation benchmarks of different types. By enabling fully autonomous world model generation without human feedback or manual annotations, this work opens new possibilities for AI systems that can reliably understand and formalize complex environments from natural language.

## ETHICS STATEMENT

All authors have read and will adhere to the ICLR Code of Ethics. This work does not involve human subjects, personal data, demographic attributes, or user studies; IRB approval was therefore not required. Our experiments use public, non-sensitive benchmarks: *Text2World*, *Code World Models Benchmark (CWMB)*, and *ByteSized32*. We complied with dataset licenses and did not attempt to deanonymize or enrich any data with personal information. Because AGENT2WORLD uses web search as a tool (§ 3.1), we enforced safeguards to reduce legal and research-integrity risks: we retrieved only publicly accessible pages, implemented a denylist to avoid solution leakage from benchmark source repositories or discussion pages as discussed in Section 4.2; We have no conflicts of interest or undisclosed sponsorship related to this work.

## REPRODUCIBILITY STATEMENT

Implementation details needed to re-create agents, tools, and evaluation are specified in Section 4.2 and Appendix B.2; algorithmic workflow and role specialization are detailed in §3.2–§3.3 (with pseudo code in Alg. 1); prompts are provided in Appendix F. For datasets and metrics, benchmark compositions and metric definitions are summarized in §C and Appendix C.2; ablation settings and additional figures/tables appear in Appendix D. As discussed in Section 4.2, to facilitate exact runs, we fix decoding parameters (temperature $= 0$, top_p $= 1$) and cap agent turns, specify external services (web search API) and denylisted domains to prevent leakage, and report hardware assumptions (CPU-only). We also provide the source code of our methods in supplementary materials.

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

# Appendix

## A    THE USE OF LARGE LANGUAGE MODELS

We used a large language model (LLM) strictly as a general-purpose writing assistant for surface-level editing, such as grammar correction, wording polish, and minor style consistency. The LLM was not used for conceptual ideation, literature review, algorithm or model design, data collection or labeling, experiment setup, result analysis, or drafting of technical content.

## B    MORE DETAILS ON METHODOLOGY

### B.1    METHOD COMPARISONS

Table 6: Comparison of AGENT2WORLD and related approaches. **Feedback** stands for whether the method uses execution/checker/test signals during generation. **External Knowledge** stands for whether explicit web/external knowledge retrieval is a first-class component. **Type** represents environment types, where ● and ○ stand for discrete and continuous environments, respectively.

| Method | Representation | Core Paradigm | Feedback | External Knowledge | Environment Type |
|---|---|---|---|---|---|
| Text2World (Hu et al., 2025a) | PDDL | Static Workflow | Y | N | ● |
| Guan et al. (2023) | PDDL | Static Workflow | Y | Human | ● |
| Oswald et al. (2024) | PDDL | Static Workflow | Y | N | ● |
| Smirnov et al. (2024) | PDDL | Static Workflow | Y | N | ● |
| AgentGen (Hu et al., 2025b) | PDDL | Static Workflow | Y | N | ● |
| WorldCoder (Tang et al., 2024) | Code | Static Workflow | Y | N | ● ○ |
| GIF-MCTS (Dainese et al., 2024) | Code | Static Workflow | Y | N | ● ○ |
| ByteSized32 Wang et al. (2023) | Code | Static Workflow | Y | N | ● |
| LLM+AL (Ishay & Lee, 2025) | Action Language | Static Workflow | Y | N | ● |
| Direct Generation | PDDL & Code | Direct | N | N | ● ○ |
| AGENT2WORLD$_{Single}$ | PDDL & Code | Adaptive Agent | Y | N | ● ○ |
| AGENT2WORLD$_{Multi}$ | PDDL & Code | Adaptive Agent | Y | Internet | ● ○ |

is shown in Table 6, we compare the related methods.

Table 7: Per-agent configuration.

| Agent | Tools |
|-------|-------|
| Deep Researcher | browser_search; browser_open |
| Model Developer | file_tool; sandbox; run_code |
| Simulation Tester | play_env; file_tool |
| Unit Tester | run_code; run_bash; file_tool |

## B.2 PER-AGENT TOOL CONFIGURATION

Detailed per-agent tool configuration is presented in Table 7.

## B.3 PSEUDO CODE OF AGENT2WORLD_Multi

We formalize the process of AGENT2WORLD_Multi in Algorithm 1.

---

**Algorithm 1:** The execution pipeline of AGENT2WORLD_Multi

---

**Input:** $T, N$
**Output:** $e$
$N_r \leftarrow$ predefined integers;
$R \leftarrow \emptyset$ ;
$E \leftarrow \emptyset$ ;
$Q \leftarrow$ ExtractQuestions$(T)$;
**for** $r \leftarrow 1$ **to** $N_r$ **do**
    $q \leftarrow$ ResearchAgent$(\text{select}, \{Q, E, R\})$;
    **if** $q = \emptyset$ **then**
        | **break**
    $L \leftarrow$ WebSearch $(q)$;
    $E \leftarrow E \cup$ ResearchAgent$(\text{summarize}, \{L\})$;
    $R \leftarrow$ ResearchAgent$(\text{update}, \{T, E, R\})$;
$F_t \leftarrow R$;
$C_{\text{last}} \leftarrow \emptyset$;
**for** $n \leftarrow 1$ **to** $N$ **do**
    $C_d \leftarrow$ DevelopAgent$(T, F_t)$;
    **if** $C_d \neq \emptyset$ **then**
        | $C_{\text{last}} \leftarrow C_d$;
        | $p_{\text{code}} \leftarrow$ FileTool$(\text{save}, C_d)$;
    **else**
        | **continue**
    $C_t \leftarrow$ UnitTestAgent$(C_d, T, R)$;
    $p_{\text{test}} \leftarrow$ FileTool$(\text{save}, C_t)$;
    $U_t \leftarrow$ CodeTool$(\text{run\_tests}, \{p_{\text{code}}, p_{\text{test}}\})$;
    $S_t^\star \leftarrow$ PlayEnv$(C_d)$;
    $S_t \leftarrow$ SimulationTestAgent$(S_t^\star, T)$;
    **if** $U_t.pass \wedge S_t.pass$ **then**
        | $e \leftarrow C_{\text{last}}$;
        | **return** $e$;
    $F_t \leftarrow$ MergeFeedback $(U_t, S_t)$;
$e \leftarrow C_{\text{last}}$;
**return** $e$

---

## C MORE DETAILS ON BENCHMARKS

### C.1 SIDE-BY-SIDE COMPARISON

A side-by-side comparison of the evaluated benchmarks in this paper is presented in Table 8

Table 8: Overview of Text2World (Hu et al., 2025a), Code World Models Benchmark (CWMB) (Dainese et al., 2024), and ByteSized32 (Wang et al., 2023). "Type" denotes the target representation (PDDL vs. executable code). Metrics are shown at the family level. A detailed explanation of each metric is presented in Appendix C.2.

| Benchmark | #Environments | Type | Metrics (core) |
|---|---|---|---|
| Text2World | 103 | PDDL | Executability; Domain similarity; F1 scores |
| CWMB | 18 | Code (Python) | Accuracy; Normalized return $\mathcal{R}$ (discrete/continuous) |
| ByteSized32 | 32 | Code (Python) | Technical validity; Specification compliance; Winnability; Physical reality alignment |

### C.2 METRIC EXPLANATION

**Text2World**

**Executability.** *Name: Exec. Range:* $[0, 1]$ (higher is better). Whether the generated {domain, problem} can be successfully parsed and validated by standard PDDL validators; reported as the fraction (percentage) over all test cases. Fine-grained metrics below are computed only when *Exec*$= 1$.

**Domain similarity.** *Name: Sim. Range:* $[0, 1]$ (higher is better). Textual/structural similarity between the generated and gold PDDL measured by a *normalized Levenshtein ratio*.

Let $X$ and $Y$ be the character sequences of the two files with lengths $|X|$ and $|Y|$, and let $\mathrm{Lev}(X, Y)$ denote their Levenshtein distance, then

$$\mathrm{Sim}(X, Y) \ = \ 1 - \frac{\mathrm{Lev}(X, Y)}{\max\{|X|, |Y|\}} \ \in \ [0, 1]. \tag{1}$$

**F1 scores.** *Range:* $[0, 1]$ (higher is better). When *Exec*$= 1$, we parse both generated and gold PDDL into structured representations and report *macro-averaged* F1 for the following components: **Predicates** ($\mathrm{F1}_{\mathrm{PRED}}$), **Parameters** ($\mathrm{F1}_{\mathrm{PARAM}}$), **Preconditions** ($\mathrm{F1}_{\mathrm{PRECOND}}$), and **Effects** ($\mathrm{F1}_{\mathrm{EFF}}$). We use the standard definition of $F_1$, where P and R denote precision and recall, respectively:

$$F_1 \ = \ \frac{2\,\mathrm{P}\,\mathrm{R}}{\mathrm{P} + \mathrm{R}}$$

**CWMB**

**Prediction Accuracy.** *Symbol:* $\mathrm{Acc}_{\mathrm{pred}}$. *Range:* $[0, 1]$ (higher is better). *Definition:* We use the same accuracy metric as in the evaluation phase of GIF–MCTS (Sec. 4). Given a validation set $D = \{(s_i, a_i, r_i, s'_i, d_i)\}_{i=1}^N$ and CWM predictions $(\hat{s}'_i, \hat{r}_i, \hat{d}_i) = \mathtt{CWM.step}(s_i, a_i)$, the accuracy uniformly weights next state, reward, and termination:

$$\mathrm{Acc}_{\mathrm{pred}} = \frac{1}{N} \sum_{i=1}^N \left[ \tfrac{1}{3}\,\mathbf{1}(\hat{s}'_i = s'_i) + \tfrac{1}{3}\,\mathbf{1}(\hat{r}_i = r_i) + \tfrac{1}{3}\,\mathbf{1}(\hat{d}_i = d_i) \right]. \tag{2}$$

**Normalized Return.** *Symbol:* $\mathcal{R}$. *Range:* unbounded (higher is better; $\mathcal{R} > 0$ means better than random; $\mathcal{R} \to 1$ approaches the oracle). *Definition:*

$$\mathcal{R} = \frac{R(\pi_{\mathrm{CWM}}) - R(\pi_{\mathrm{rand}})}{R(\pi_{\mathrm{true}}) - R(\pi_{\mathrm{rand}})}, \tag{3}$$

where $R(\pi)$ denotes the return. *Protocol:* as in the original setup, we use vanilla MCTS for discrete action spaces and CEM for continuous action spaces; $R(\cdot)$ is averaged across a fixed number of episodes per environment (10 in the original), and $R(\pi_{\mathrm{rand}})$ uses the environment's random policy baseline.

**ByteSized32**

**Technical Validity.** Range: $[0, 1]$. Measured in the order of API calls, such that failure of an earlier function implies failure of subsequent tests. `Game initialization` is evaluated once at the beginning of the game, whereas `GENERATEPOSSIBLEACTIONS()` and `STEP()` are evaluated at *every step*. We check:

- *Game initialization*: the game/world initializes without errors;
- *Valid actions generation*: the routine that enumerates valid actions for the current state returns without errors (verified via a bounded path crawl);
- *Runnable game*: a bounded-depth crawl of trajectories executes without errors.

**Specification Compliance.** Range: $[0, 1]$. An LLM acts as the judge for *true/false* compliance against the task specification. The prompt provides the task spec `{GAME_SPEC}`, the game code `{GAME_CODE}`, and an evaluation question `{EVAL_QUESTION}`; the LLM is instructed to first output `Yes/No` and then a brief rationale. To reduce variance, we use a fixed prompt template and perform multiple independent runs with majority vote/mean aggregation. We report three submeasures: *Task-critical objects*, *Task-critical actions*, and *Distractors*.

**Physical Reality Alignment.** Range: $[0, 1]$. Automatic evaluation proceeds in two stages:
*(1) Trajectory generation:* perform a breadth-first crawl using the action strings returned by `GENERATEPOSSIBLEACTIONS()` at each step; actions are grouped by verb (first token) and expanded in a bounded manner. If an error occurs, the error message is recorded as the observation, and the search continues.
*(2) Sampling and judgment:* group paths by the last action verb, draw a fixed-size subsample approximately balanced across groups, and submit each path—together with the task description `{GAME_TASK}`—to an LLM for a binary judgment (`yes/no`; errors are treated as failures). The final score is the fraction judged aligned.

**Winnability.** Range: $[0, 1]$. A text-game agent (LLM agent) attempts to reach a terminal `win` within horizon $H$; we report the fraction of tasks deemed winnable. Given the limited agreement between automatic and human assessments for this metric, we prioritize human evaluation in the main results and use the automatic estimate asan auxiliary reference.

# D    MORE DETAILS ON ABLATION STUDY

Table 9: Ablation Study of AGENT2WORLD on CWMB (Dainese et al., 2024).

| Method | Discrete Action Space | | Continuous Action Space | | Overall | |
|---|---|---|---|---|---|---|
| | Accuracy ($\uparrow$) | $\mathcal{R}$ ($\uparrow$) | Accuracy ($\uparrow$) | $\mathcal{R}$ ($\uparrow$) | Accuracy ($\uparrow$) | $\mathcal{R}$ ($\uparrow$) |
| AGENT2WORLD | **0.9174** | **0.8333** | **0.3575** | **0.3050** | **0.5441** | **0.4811** |
| No *Deep Researcher* | $0.8794_{-0.0380}$ | $0.4407_{-0.3926}$ | $0.3404_{-0.0171}$ | $0.2201_{-0.0849}$ | $0.5201_{-0.0240}$ | $0.2936_{-0.1875}$ |
| No *Simulation Tester* | $0.8920_{-0.0254}$ | $0.5718_{-0.2615}$ | $0.3288_{-0.0287}$ | $0.1577_{-0.1473}$ | $0.5275_{-0.0166}$ | $0.3039_{-0.1772}$ |
| No *Unit Tester* | $0.6166_{-0.3008}$ | $0.3863_{-0.4470}$ | $0.3025_{-0.0550}$ | $0.1704_{-0.1346}$ | $0.4072_{-0.1369}$ | $0.2423_{-0.2388}$ |

Detailed experimental results of the ablation study are presented in Table 9.

# E    MORE DETAILS ON ERROR ANALYSIS

## E.1    ERROR ANALYSIS ON TEXT2WORLD AND BYTESIZED32

We visualize the error patterns during evaluation-driven refinement on Text2World and ByteSized32 in Figure 7 and Figure 8.

## E.2    DISTRIBUTION OF ERROR TYPES

A detailed proportion of error types on Text2World, CWMB, ByteSized32 are presented in Table 10, Table 11 and Table 12, respectively.

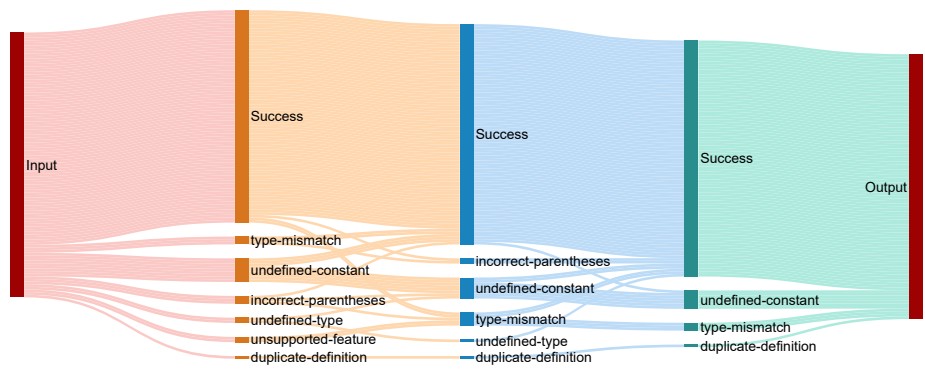

Figure 7: Error distribution of AGENT2WORLD$_{Multi}$ on Text2World.

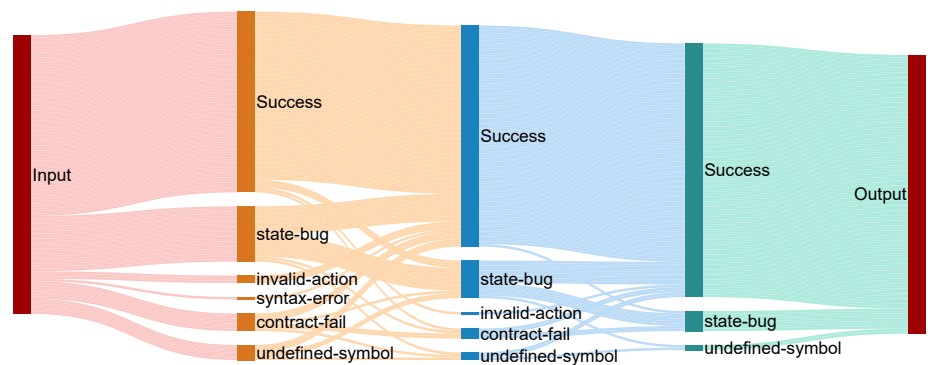

Figure 8: Error distribution of AGENT2WORLD$_{Multi}$ on ByteSized32.

Table 10: Distribution of Syntax Errors in Text2World Across Turns

| Error Type | Explanation | Turn 0 (%) | Turn 1 (%) | Turn 2 (%) |
|---|---|---|---|---|
| undefined-constant | Reference to undeclared constants in predicates or actions. | 8.91 | 7.92 | 6.93 |
| type-mismatch | Parameter type conflict with declared type constraints. | 2.97 | 4.95 | 2.97 |
| incorrect-parentheses | Invalid or mismatched parentheses. | 2.97 | 1.98 | 0.00 |
| undefined-type | Undeclared parent type in hierarchical type definitions. | 1.98 | 0.99 | 0.00 |
| unsupported-feature | Parser-incompatible features (e.g.,either types). | 1.98 | 0.00 | 0.00 |
| duplicate-definition | Multiple declarations of identical domain elements. | 0.99 | 0.99 | 0.99 |

Table 11: Distribution of Syntax Errors in CWMB Across Turns

| Error Type | Explanation | Turn 0 (%) | Turn 1 (%) | Turn 2 (%) | Turn 3 (%) |
|---|---|---|---|---|---|
| signature-mismatch | Arity/types do not match the declared signature. | 27.78 | 11.11 | 11.11 | 5.56 |
| schema-mismatch | Value type/shape/dtype violates the expected schema. | 22.22 | 5.56 | 0.00 | 0.00 |
| dynamics-error | State or reward deviates from expected dynamics. | 11.11 | 44.44 | 33.33 | 11.11 |
| non-deterministic | Results are inconsistent under fixed conditions. | 11.11 | 0.00 | 0.00 | 0.00 |
| judgment-bug | Environment setup inconsistent with the description. | 11.11 | 5.56 | 11.11 | 5.56 |
| invariant-violation | Internal invariants are broken (e.g., illegal config). | 0.00 | 5.56 | 0.00 | 5.56 |

Table 12: Distribution of Syntax Errors in ByteSized32 Across Turns

| Error Type | Explanation | Turn 0 (%) | Turn 1 (%) | Turn 2 (%) |
|---|---|---|---|---|
| state-bug | State inconsistency across steps. | 19.82 | 13.51 | 7.21 |
| contract-fail | Task/API contract not satisfied. | 6.31 | 3.60 | 0.00 |
| undefined-symbol | Reference to undeclared name/type/domain/constant. | 5.41 | 2.70 | 1.80 |
| invalid-action | Unknown or unsupported action not safely handled. | 2.70 | 0.90 | 0.00 |
| syntax-error | Load/parse failure (e.g., null bytes, syntax/indentation errors). | 0.90 | 0.00 | 0.00 |

### E.3 ERROR DISTRIBUTION OF BASELINES AND COMPARISON

Table 13: Distribution of Syntax Errors in CWMB Across Methods and Turns

| Method | signature-mismatch | schema-mismatch | dynamics-error | non-deterministic | judgment-bug | invariant-violation |
|---|---|---|---|---|---|---|
| WorldCoder (Last Turn) | 0 | 0 | 3 | 0 | 2 | 2 |
| GIF-MCTS (Last Turn) | 0 | 0 | 2 | 0 | 3 | 2 |
| AGENT2WORLD$_{Multi}$ (Turn 1) | 2 | 1 | 8 | 0 | 1 | 1 |
| AGENT2WORLD$_{Multi}$ (Turn 2) | 2 | 0 | 6 | 0 | 2 | 0 |
| AGENT2WORLD$_{Multi}$ (Turn 3) | 1 | 0 | 2 | 0 | 1 | 1 |

As shown in Table 13, both WorldCoder and GIF-MCTS still exhibit several residual dynamics-related failures in their final outputs: although they no longer trigger signature- or schema-mismatch errors, they retain multiple *dynamics-error* and *judgment-bug* cases, together with non-trivial *invariant-violation* counts. This suggests that their search procedures are effective at ironing out interface- and type-level inconsistencies, but are less successful at fully aligning the learned dynamics with the intended environment specification.

In contrast, AGENT2WORLD$_{Multi}$ starts from a broader spectrum of error types at turn 1 (including signature/schema mismatches and a larger number of dynamics errors), and progressively suppresses them over subsequent refinement rounds. By turn 3, schema mismatches and nNon-deterministicbehaviors have disappeared, signature mismatches are reduced to a single case, and dynamics errors, judgment bugs, and invariant violations are all substantially lower than those of the baselines. This trajectory is consistent with the qualitative analysis in Section 5.4: the Testing Team first drives the Model Developer to resolve structural and specification-level issues, and then focuses on subtle dynamics and invariant failures, ultimately producing world models that are both syntactically robust and semantically faithful to the target environments.

## F PROMPT EXAMPLES

---

**Deep Researcher**

You are a world-class Systems Analyst and Technical Specification Writer, specializing in creating reinforcement learning environments compliant with the Gymnasium API. Your mission is to transform an ambiguous task description into a precise, actionable, and verifiable technical specification.
<Environment Name>
**CliffWalking-v0**
</Environment Name>
**<TASK DESCRIPTION>**
Cliff walking involves crossing a gridworld from start to goal while avoiding falling off a cliff.
## Description ...
**</TASK DESCRIPTION>**

---

**\<Workflow\>**

Please strictly follow the following six-step process:

- **Deconstruction and Analysis (Use Version Locking)**
  - Identify all ambiguities, gaps, and conflicts in the task description.
  - Lock the exact environment version and all key library versions (record name, version, and source link).
  - Categorize gaps by type: missing value/unit/range boundary/time-sensitive/ambiguous reference/unclosed list/conflict/no provenance.

- **Planning and Investigation (Authoritative Search + Evidence Log)**
  - For each high/mid-level project, write 1–2 focused queries that include: synonyms/abbreviations, site filters, authoritative domains (e.g., site:numpy.org, site:mujoco.readthedocs.io, site:doi.org), and recency windows (e.g., after 2024-01-01 or "last 2 years").
  - Execute the query using browser_search and open >= 2 trusted results with browser_open.
  - If the top source disagrees, open >= 1 additional authoritative sources and triangulate.
  - Create an evidence log entry for each opened page: Title | Organization/Author | Version/Submission | URL (+ archived URL) | Publication Date | Access Date (Asia/Singapore) | 3 Key Facts | Confidence (High/Medium/Low).

- **Synthesis and Citation (Conflict Resolution)**
  - Integrate the findings into a concise evidence summary with citations.
  - When sources conflict, explain the differences and justify the chosen resolution (related to version locking).

- **Refinement and Improvement (Specification Patch)**
  - Generate a structured "diff": action/observation space; rewards; termination/truncation; timing (dt/frame_skip); seeding and certainty; numerical tolerances; dependencies; interface flags.

- **Formalization and Finalization (Ready-to-Use Specification)**
  - Write the final specification according to the \<Output Format\> , including the public API, core logic, usage scenarios, and a verification plan aligned with metrics and statistical validation.

- **Review and Self-Correction (Compliance Check)**
  - Verify conformance to the \<Output Constraints\> (OUTPUT_CONSTRAINTS\>), version consistency, SI units, ISO dates, and the inclusion of any code.

**\</Workflow\>**

**\<OUTPUT_CONSTRAINTS\>**

- Strictly adhere to the structure defined in `<PLANNING_STRUCTURE>`.

- Do **NOT** output runnable code definitions (classes, functions). Only may include short illustrative snippets or pseudo-code.

- All claims about industry standards or common practices MUST be supported by citations.

- Use ISO-8601 dates (e.g., 2025-09-02).

- Use SI units for physical and mathematical quantities.

- Data-leakage rule: Do not access, copy, quote, or derive from raw source code in the OpenAI/Gym/Gymnasium repositories or similar code repositories. Do not include any repository code in the output. Prefer official documentation, standards, papers, or reputable secondary sources. If the only available evidence is a code repository, summarize behavior without copying code and mark it as an inference with risks.

**<PLANNING_STRUCTURE>**

- Your output must begin with this planning and analysis section.
- **Ambiguity Analysis**
    - List each ambiguity/vagueness/conflict and mark Impact: High / Medium / Low.
    - Cover at least: missing numeric value, missing unit, missing boundary/range, time-sensitive items, unclear references, open lists ("etc."/"e.g."), conflicts, and missing citation.
- **Investigation Plan**
    - For each High/Medium item, provide one atomic question.
    - For each question, provide 1–2 executable queries including: synonyms/abbreviations, a site filter to authoritative domains, and a time window (e.g., `after:2024-01-01` or "past 2 years").
    - State the minimum evidence policy: High/Medium $\rightarrow \geq 2$ credible sources; if disagreement $\rightarrow$ add $\geq 1$ more for triangulation.

**<Formula requirements>**

- For any formula, define all symbols, units, and applicability constraints.
- Cite the source of the formula immediately after its definition.
- Provide the complete formula rather than a descriptive explanation.

**<OUTPUT_FORMAT>**

- Please provide the final specification document structured as follows. This is the primary deliverable. Do **NOT** include code.
- <Version & Provenance>
- <Evidence Summary>
- <Spec Patch>
- <Theoretical Foundations>
- <Final Specification>
- <Assumptions & Risks>
- <Third-Party Library Usage>

**Model Developer**

**<TASK DESCRIPTION>**
Cliff walking involves crossing a gridworld from start to goal while avoiding falling off a cliff.
## Description ...
**</TASK DESCRIPTION>**
**<Research Report>**
# CliffWalking-v0 Environment Specification
**Version & Provenance**

- Environment: CliffWalking-v0
- Gymnasium version: v0.26.3 (2022-09-15) and latest documentation snapshot (2025-01-01)
- Scope: This specification normalizes the CliffWalking environment as implemented in Gymnasium v0.26.3, aligning with the classic gridworld cliff walking task.
- Accessed date: 2024-06-01 (Asia/Singapore)

**Evidence Summary**

- The environment is a 4x12 grid with 48 discrete states indexed by row-major flattening: state = row * 12 + col.
- Valid states exclude cliff cells ([3,1..10]) and the goal ([3,11]). The player can occupy all cells in the first 3 rows plus the bottom-left cell [3,0].

- The start state is 36 (row=3, col=0), and the goal state is 47 (row=3, col=11).
- The action space is Discrete(4) with actions: 0 (up), 1 (right), 2 (down), 3 (left).
- Each step yields a reward of -1. Stepping into the cliff yields a reward of -100 and resets the player to the start state; the episode continues.
- The episode terminates only when the player reaches the goal state.
- The set_state method is not explicitly documented but is inferred to accept any valid non-terminal, non-cliff state and sets the environment to not done.
- No explicit error handling for invalid actions is documented; valid inputs are assumed.
- The environment is deterministic.

**Spec Patch**

- action.space = Discrete(4), values {0,1,2,3}, shape (1,)
- observation.space = Discrete(48), integer in [0,47], representing flattened grid index: row * 12 + col
- valid states = all grid cells except cliff cells ([3,1..10]) and goal ([3,11])
- start_state = 36 (row=3, col=0)
- goal_state = 47 (row=3, col=11)
- reward.step = -1 per step
- reward.cliff = -100 on stepping into cliff
- episode termination = True if state == goal_state; else False
- stepping into cliff resets player to start_state, episode continues
- set_state(state: int) sets environment state to given valid state, done = False
- no maximum episode length or truncation condition
- no explicit error handling for invalid actions; assume valid input
- environment deterministic transitions

**Theoretical Foundations**

- **MDP Definition:**
  - State space $S = \{0, \ldots, 47\} \setminus \{\text{cliff states, goal state}\}$.
  - Action space $A = \{0, 1, 2, 3\}$.
  - Transition function $T(s, a, s')$ (deterministic):
    * If $s'$ is a cliff cell, next state = start_state.
    * Else next state = $s + \delta(a)$ (with grid boundary checks).
  - Reward function $R(s, a, s')$:
    * $R = -100$ if $s'$ is a cliff cell.
    * $R = -1$ otherwise.
  - Episode ends when $s' = $ goal_state.
- **Symbol Table:**
  - $s$: state (int), flattened grid index, $0 \leq s \leq 47$, $s$ not in cliff or goal.
  - $a$: action (int), in $\{0 : \text{up}, 1 : \text{right}, 2 : \text{down}, 3 : \text{left}\}$.
  - $r$: reward (float), $-100$ or $-1$.
  - done: boolean, True if $s = $ goal_state.
  - start_state: 36 (int).
  - goal_state: 47 (int).
  - cliff_cells: set of ints corresponding to grid positions $[3, 1..10]$.
  - nrows: 4 (int).
  - ncols: 12 (int).

- **Assumptions:**
  - No shaping rewards.
  - Deterministic environment.
  - No truncation or max step limit.

**Final Specification**

- **Environment Name**
  - CliffWalking-v0
- **Public API**
  - __init__()
  - set_state(state: int)
  - step(action: int) $\rightarrow$ (observation: int, reward: float, done: bool)
- **Core Logic Description**
  - __init__:
    * Initialize grid size (4 rows $\times$ 12 columns).
    * Define start_state = 36 (row=3, col=0).
    * Define goal_state = 47 (row=3, col=11).
    * Define cliff cells as positions [3, 1..10].
    * Initialize current state to start_state.
    * Initialize done flag to False.
  - set_state(state: int):
    * Validate that state is a valid observation (not cliff or goal).
    * Set current state to state.
    * Set done flag to False.
  - step(action: int) $\rightarrow$ (obs, reward, done):
    * Validate action is in $\{0, 1, 2, 3\}$.
    * Compute next position based on action with boundary checks:
      · 0: move up (row $-1$)
      · 1: move right (col $+1$)
      · 2: move down (row $+1$)
      · 3: move left (col $-1$)
    * If next position is outside grid, remain in current position.
    * If next position is a cliff cell:
      · reward $= -100$
      · next state $=$ start_state
      · done $=$ False
    * Else if next position is goal_state:
      · reward $= -1$
      · done $=$ True
      · next state $=$ goal_state
    * Else:
      · reward $= -1$
      · done $=$ False
      · next state $=$ computed position
    * Update current state to next state and return (next state, reward, done).
- **Usage Scenarios**
  - **Scenario 1: Normal move into cliff**
    * Initial state: 36 ([3,0])
    * Action: 1 (right)
    * Next state: 37 ([3,1]) is cliff $\rightarrow$ reset to 36
    * Reward: $-100$

* done: False
- **Scenario 2: Move to goal**
    * Current state: 46 ([3,10])
    * Action: 1 (right)
    * Next state: 47 ([3,11]) goal
    * Reward: $-1$
    * done: True
- **Scenario 3: Move up from start**
    * Current state: 36 ([3,0])
    * Action: 0 (up)
    * Next state: 24 ([2,0])
    * Reward: $-1$
    * done: False

**Assumptions & Risks**

- No truncation or maximum episode length specified; risk of infinite episodes if goal not reached.
- set_state accepts only valid non-terminal, non-cliff states; no explicit error handling for invalid states.
- No explicit error handling for invalid actions; assumed caller responsibility.
- Reward and reset behavior on cliff stepping inferred from documentation; no episode termination on cliff.
- Observation space includes all grid cells but player never occupies cliff or goal states.
- Potential future Gymnasium version changes may alter behavior.

**Third-Party Library Usage**

- Gymnasium API conventions inform environment interface: discrete action and observation spaces, step and reset semantics.
- No new runtime dependencies introduced.
- RNG seeding not specified; environment deterministic.
- No source code copied; specification based on official Gymnasium documentation and Sutton & Barto RL book.
- Numpy indexing conventions used for grid flattening.

**</Research Report>**

**<Workflow>**
1. **Deconstruct Specification:** Carefully review the <Task Describe>,<Research Report> to fully understand the environment's specification, including state/action spaces, dynamics, reward function, and termination conditions.
2. **Physics Engine Selection:** Evaluate if the task requires physics simulation. If so, choose an appropriate physics engine for the specific task requirements.
3. **Model Design:** If using a physics engine, design the model structure and embed it as needed in the Python file.
4. **Plan Class Structure:** Outline the 'Environment' class, including its internal state variables, helper methods, and the public interface ('__init__', 'reset', 'set_state', 'step').
5. **Implement Complete Code:** Write the full implementation of the 'Environment' class. 6. **Self-Correction Review:** Meticulously check that the generated code fully complies with the <TASK DESCRIPTION>, the <Research Report>, and all <ImplementationRequirements>.
7. **Finalize Output:** Present the complete, reviewed, and runnable single-file code in the specified final format.
**</Workflow>**

**<ImplementationRequirements>**

1. **Interface (single file):**

   - Implement a complete, self-contained Python class Environment with:
     - __init__(self, seed: int | None = None)
     - reset(self, seed: int | None = None) → ndarray (reinitialize the episode and return the initial observation in canonical shape)
     - set_state(self, state) (must accept ndarray *or* list/tuple in canonical shape)
     - step(self, action) → tuple[ndarray, float, bool] (returns: observation, reward, done)
   - Requirements:
     - Single-file constraint: all code, including any model definitions, must be contained in one Python file.
     - For physics-based environments, embed model definitions as string constants within the class.
     - Explicitly define state, action, and observation spaces (types, shapes, ranges, formats).
     - Provide reproducibility (seeding) via the constructor and/or a seed(int) method.
     - Be robust to common representations:
       * set_state: accept list/tuple/ndarray of the same logical content.
       * step: accept int / NumPy integer scalar / 0-D or 1-D len-1 ndarray (convert to canonical form; raise clear TypeError/ValueError on invalid inputs).
     - No dependence on external RL frameworks; no Gym inheritance.
     - No external file dependencies (model definitions must be embedded).
     - Maintain internal state consistency; allow reconstruction from observations where applicable.
     - Clean, readable code suitable for RL experimentation.

2. **Determinism & validation:**

   - Provide reproducibility via seed (constructor and/or seed(int) method).
   - Normalize inputs: accept equivalent representations (e.g., NumPy scalar/int/len-1 array) and convert to a canonical form.
   - Validate inputs; raise clear one-line errors (ValueError/TypeError) on invalid shapes or ranges.

3. **Dynamics (MCTS/control oriented):**

   - For physics-based tasks, prefer suitable physics simulation methods with embedded model definitions over custom physics implementations.
   - Choose and document an integration scheme (e.g., implicit integrator, explicit Euler) consistent with the research report.
   - Use a stable time step $dt$; clamp to safety bounds; keep all values finite (no NaN/Inf).
   - Keep per-step computation efficient and allocation-light.

4. **Dependencies & style:**

   - No Gym inheritance or external RL frameworks unless explicitly allowed.
   - Allowed: third-party libraries as needed (e.g., NumPy, physics engines, SciPy, Numba, JAX, PyTorch, etc.).
   - For robotics/physics tasks, physics engines with embedded model definitions are recommended over custom implementations.
   - Clean, readable code suitable for RL experimentation.
   - All dependencies must be importable standard libraries or commonly available packages.

**</ImplementationRequirements>**
**<Output Format>**

**<final> <code_file_path>** The entrypoint file path of the generated code. **</code_file_path>**
**<entrypoint_code>** "'python # Your complete, runnable single-file implementation here. "'
**</entrypoint_code> </final>**
**</Output Format>**

### Unit Tester

**<TASK DESCRIPTION>** Cliff walking involves crossing a gridworld from start to goal while avoiding falling off a cliff.
## Description ...
**</TASK DESCRIPTION>**
**<CodeArtifact path="environment.py">** {code} **</CodeArtifact>**
**<ExecutionPolicy>**

- Do not modify the student's source file.

- Create exactly one pytest file at "tests/test_env.py" using file_tool("save").

- Import the module from "environment.py" via importlib (spec_from_file_location + module_from_spec).

- Run tests with code_tool("run", "pytest -q"); capture exit_code, duration, and stdout/stderr tail.

**</ExecutionPolicy>**
**<TestPlan>**

- Sanity: class `Environment` can be imported and instantiated, e.g., `Environment(seed=0)`.

- Contract:

  1. `set_state` accepts list/tuple/ndarray of the same logical content (convert to canonical).
  2. `step(action)` returns a 3-tuple: (observation, reward, done) with expected types/shapes.
  3. Determinism: with the same seed and same initial state, the first step with the same action yields identical outputs.
  4. Action space validation: actions within bounds are accepted, out-of-bounds actions are handled gracefully.
  5. Observation space validation: observations match declared space bounds and shapes.
  6. State space consistency: internal state dimensions match expected environment specifications.

- Acceptance: success iff pytest exit_code == 0 (all tests pass).

**</TestPlan>**
**<ReportingGuidelines>**

- Summarize pytest results in 2–4 sentences; mention the first failing nodeid/assert if any.

- Provide a brief contract coverage assessment and the most probable root cause for failures.

- If failing, add 1–3 concise actionable fixes (no long logs).

**</ReportingGuidelines>**
**<OutputFormat>** Return exactly one <final> block containing a single JSON object that matches PytestReport: {
"success": true|false,
"analysis": "<2–4 sentence summary/diagnosis>",
"suggest_fix": " 1–3 bullets with minimal actionable changes>"
} No extra text outside <final>. No additional code fences.
**<final>** { "success": false, "code_result": "", "analysis": "", "suggest_fix": "" } **</final>**
**</OutputFormat>**

**Simulation Tester**

Your task is to interact with the environment code and then analyze the feedback from the interaction and propose modifications
**\<TASK DESCRIPTION\>**
Cliff walking involves crossing a gridworld from start to goal while avoiding falling off a cliff.
## Description ...
**\</TASK DESCRIPTION\>**
**\<CodeArtifact path="environment.py"\>**
{code}
**\</CodeArtifact\>**

**\<ExecutionPolicy\>**
- Use the play_env tool exactly once on "environment.py" - If the tool throws or cannot run, perform diagnosis from static review only; still produce output in the required format.
**\</ExecutionPolicy\>**
**\<Rubric\>**
Success (boolean) must be decided from the available signals with graceful degradation:

- **Primary (step-level signals present):**
    - success = true iff the run finished without exceptions AND there is NO misclassified_transition with (valid == false OR state_matches == false).
    - If only observation deltas are available, use obs_matches instead of state_matches.
    - When numeric deltas are provided, treat matches = true if max_abs_error $\leq 10^{-3}$ or rel_error $\leq 10^{-3}$.

- **Secondary (no per-step signals):**
    - If success_rate exists: success = true iff no exceptions AND success_rate $\geq$ 0.95.
    - Else: success = true iff no exceptions AND no invariant/contract violations you can substantiate from code and logs.

- **Reward/termination:**
    - If reward_matches == true AND done_matches == true, explicitly state they match and DO NOT propose changes to reward or termination logic.

- **Action space consistency (discrete & continuous):**
    - If GT exposes `Box(low, high, shape)`: align predicted bounds and expose them (e.g., `env.action_space` or a getter). Never place clipping inside the integrator; clamp only at action ingestion or at observation.
    - If GT exposes `Discrete(n)`: actions must be integer indices in [0, n−1]; expose n (e.g., `gym.spaces.Discrete(n)`); if indices map to continuous commands/torques, list the mapping table and align it with GT; never float-clip discrete actions.
    - If action-space info is missing, skip these checks (do not speculate).

- **Internal vs observation:**
    - If clipping or angle normalization is found inside the integrator step (e.g., in `_rk4_step`), this likely causes trajectory drift; propose moving them to the observation path (e.g., `_get_observation`) unless GT specifies otherwise.
    - If latent state is unavailable but observations exist, compare observations instead and state this explicitly.

- **Integrator & timestep:**
    - Mismatches in integrator method (e.g., RK4 vs Euler) or `dt` can cause state divergence even when reward/done match; acknowledge and, if state mismatches persist, propose aligning method/`dt` to GT.

- **Batched/multiple transitions:**

> - If multiple transitions are reported, aggregate sensibly (e.g., mean success_rate or fraction matched $\geq 0.95$) before deciding success.
>
> **</Rubric>**
> **<Procedure>**
> 1) Static review: scan for action bounds, clipping/normalization inside integrator, integrator/dt choice, and how observation is formed.
> 2) Execute: call play_env once.
> 3) Diagnose: reconcile play_env signals with code; if reward/done matched, explicitly say so. If state mismatched, point to ONE OR TWO most likely roots.
> 4) Suggest: 1–3 smallest patches that directly address the identified root causes.
> **</Procedure>**
> **<OutputFormat>**
> Return exactly one <final> block containing a single JSON object that matches PlayReport: { "success": true|false, "analysis": "<2–4 sentences summarizing what happened and why; mention matches/mismatches explicitly>", "suggest_fix": "- bullet 1\n- bullet 2\n- bullet 3 (optional)" } No extra text outside <final>. No additional code fences.
> **</OutputFormat>**

# G  EXAMPLES OF DEEP RESEARCHER REPORT

**Deep Researcher's Report**

**<Version & Provenance>**

- Environment: Pusher-v4
- MuJoCo version: $\geq 2.1.3$ and $< 3.0.0$
- Source release date: 2024-01-01 (approximate for v4)
- Accessed date: 2025-06-01 (Asia/Singapore)

**<Spec Patch>**

- **action.space** = `Box(low=-2.0, high=2.0, shape=(7,), dtype=np.float32)`
- **observation.space**: Box with low = $-\infty$, high = $\infty$, shape $(23,)$, dtype `np.float64`
- **reward.weights** = {dist: 1.0, ctrl: 0.1, near: 0.5}
- **reward.formula**: `reward = reward_dist + 0.1 reward_ctrl + 0.5 reward_near`
- `reward_ctrl` $= -\|\mathbf{a}\|_2^2$ (negative squared Euclidean norm)
- `reward_dist` $= -\|\mathbf{o} - \mathbf{g}\|_2$ (negative Euclidean distance)
- `reward_near` $= -\|\mathbf{f} - \mathbf{o}\|_2$ (negative Euclidean distance)
- **episode.max_steps** $= 100$
- **episode.termination**: never (`terminated` always `False`)
- **episode.truncation**: `True` if step count $\geq$ `max_steps` or any observation value is non-finite
- $dt = 0.05\,\mathrm{s}$ (`frame_skip`$= 5$, `frame_duration`$= 0.01$)
- **initial_state.joint_positions** $= \mathbf{0}_7$
- **initial_state.joint_velocities**: each joint $\sim U(-0.005, 0.005)$
- **initial_state.object_position**: $x \sim U(-0.3, 0)$, $y \sim U(-0.2, 0.2)$; resample until $\|(x, y)\|_2 > 0.17$
- **initial_state.object_velocity** $= \mathbf{0}_3$

- **initial_state.goal_position** $= [0.45, -0.05, -0.323]$
- **initial_state.goal_velocity** $= \mathbf{0}_3$
- **step_function**:
  - clips action to action-space bounds
  - applies action torques to joints
  - advances MuJoCo simulation by $dt$
  - updates observation vector from simulation state
  - computes reward components and total reward
  - checks for finite observation values
  - increments step count
  - returns `(observation, reward, terminated=False, truncated=done_flag, info)` with reward components in `info`
- **seeding**: via `reset(seed=...)` method only
- **info dict keys**: "`reward_dist`", "`reward_ctrl`", "`reward_near`"
- **fingertip position**: MuJoCo site "`tips_arm`"
- **object & goal positions**: MuJoCo bodies "`object`" and "`goal`"
- **internal state**: corresponds to MuJoCo simulation state

**<Theoretical Foundations>**

- **MDP tuple:** $(S, A, P, R)$
  - $S$: continuous state space $\mathbb{R}^{23}$ (23-dimensional real vector)
  - $A$: continuous action space $\mathbb{R}^7$ with elementwise bounds $[-2, 2]$ (torques)
  - $P$: transition probability induced by MuJoCo physics with $dt = 0.05\,\text{s}$
  - $R$: reward function as defined below
- **Reward function**
  - Definition:
  $$r(s, a) = -\left\|P_{\text{object}} - P_{\text{goal}}\right\|_2 - 0.1\left\|a\right\|_2^2 - 0.5\left\|P_{\text{fingertip}} - P_{\text{object}}\right\|_2$$
  - Where:
    * $P_{\text{object}}, P_{\text{goal}}, P_{\text{fingertip}} \in \mathbb{R}^3$ are positions in meters
    * $\|\cdot\|_2$ is the Euclidean norm
    * $a \in \mathbb{R}^7$ is the action torque vector in $N\cdot m$
- **Episode ending**
  - Truncation after 100 steps or if any observation value is non-finite
  - No termination condition (i.e., `terminated=False` always)
- **Fingertip position**
  - Computed via forward kinematics internally by MuJoCo
- **Symbol Table**
  - $a$: Action torque vector; Unit $N\cdot m$; Range: each element in $[-2, 2]$
  - $P_{\text{fingertip}}$: Fingertip 3D position; Unit m; Range: real values, unbounded
  - $P_{\text{object}}$: Object 3D position; Unit m; Range: real values, unbounded
  - $P_{\text{goal}}$: Goal 3D position; Unit m; Value: fixed at $[0.45, -0.05, -0.323]$
  - $r$: Reward; Unit: unitless; Range: real values (sum of negative components)
  - $dt$: Simulation timestep; Unit s; Value: 0.05
  - `step_count`: Current timestep count; Unit: integer; Range: 0 to 100
- **Final Specification**
  - Environment Name: `Pusher-v4`

- Class Name: `Environment`
- **Public API**
  * `__init__(self)`
    · Initializes the MuJoCo simulation environment with Pusher-v4 model
    · Sets initial internal variables including step count
    · Defines action and observation spaces as specified
    · Sets fixed goal position
  * `set_state(self, state: np.ndarray) -> None`
    · Input: state vector of shape $(23,)$ matching observation space
    · Sets the internal MuJoCo simulation state to correspond to the given observation
    · Resets step count to zero; assumes state is valid and episode not done
  * `step(self, action: np.ndarray) -> Tuple[np.ndarray, float, bool, dict]`
    · Input: action vector of shape $(7,)$ clipped to $[-2, 2]$
    · Applies action torques to the simulation joints and advances by $dt = 0.05\,\text{s}$
    · Updates observation vector; computes reward components and total reward
    · Truncates if any observation value is non-finite or if step count $\geq 100$
    · Returns: observation $(23,)$ `float64`, reward `float`, terminated `False`, truncated `bool`, info with keys "`reward_dist`", "`reward_ctrl`", "`reward_near`"
- **Core Logic**
  * Simulates a 7-DOF robotic arm pushing an object to a goal
  * State includes joint angles/velocities, fingertip, object, and goal positions
  * Actions are joint torques clipped to $[-2, 2]$
  * Physics advances by $0.05\,\text{s}$ per step
  * Reward encourages moving object to goal, penalizes large actions, and encourages fingertip proximity to object
  * Episode truncates after 100 steps or on non-finite state
  * Fingertip and object positions come from MuJoCo sites/bodies
  * Seeding via `reset(seed=...)` (not shown here)
- **Usage Scenarios & Examples**
  * **Scenario 1: Initial step with zero action**
    · Observation: initial state vector with zero joint positions, small velocity noise, specified object and goal
    · Action: `np.zeros(7, dtype=np.float32)`
    · Reward: negative sum of distances (control penalty $= 0$)
    · Terminated: `False`; Truncated: `False`
    · Info: `{"reward_dist": value, "reward_ctrl": 0.0, "reward_near": value}`
  * **Scenario 2: Non-zero action moves fingertip closer**
    · Observation: updated state after physics step
    · Action: `np.array([1.0, -0.5, 0.0, 0.2, 0.0, -1.0, 0.5], dtype=np.float32)`
    · Reward: computed per formula
    · Terminated: `False`; Truncated: `False` (unless step count reached 100)
    · Info: `{"reward_dist": value, "reward_ctrl": negative squared norm of action, "reward_near": value}`

**<Assumptions & Risks>**

- Assumes MuJoCo physics engine is used as backend with version constraints.

- Assumes internal MuJoCo state can be set and queried to implement `set_state` and `step`.
- Fingertip position is assumed to be accessible via MuJoCo site "`tips_arm`".
- Reward weights are fixed as per v4; no user-configurable weights.
- Episode termination is only by truncation; no natural termination.
- Seeding is only via `reset(seed=...)`; no separate `seed` method.
- Risks include dependency on MuJoCo simulation correctness and numerical stability.
- No explicit handling of invalid actions beyond clipping.
- No rendering or visualization included.

**<Library Usage>**
- Uses Python standard library and NumPy only.
- NumPy used for array operations, clipping (`np.clip`), norm calculations (`np.linalg.norm`), and finite checks (`np.isfinite`).
- No external dependencies beyond MuJoCo simulation backend (assumed).
- All units are SI: torque in N·m, angles in radians, positions in meters, time in seconds.
- No source code copied; specification based on official documentation and API standards.

**</final>**

## H   CASE STUDY

### H.1   TEXT2WORLD

**Task Describe.** Prepare and deliver sandwiches to a set of children: assemble sandwiches in the kitchen (optionally gluten-free), place them on a tray, move the tray to the child's location, and serve subject to allergy constraints. Gluten-allergic children must receive a gluten-free sandwich; non-allergic children may receive any sandwich. Serving requires the sandwich on the tray and the tray at the child's location; making a sandwich switches it from "notexist" to "prepared." The goal is that all children become served.

**Prev. SOTA.**

```
(define (domain child-snack)
  (:requirements :typing :negative-preconditions)
  (:types
    child
    bread-portion
    content-portion
    sandwich
    tray
    place
  )

  (:constants
    kitchen - place
  )

  (:predicates
    (at_kitchen_bread ?b - bread-portion)
    (at_kitchen_content ?c - content-portion)
    (at_kitchen_sandwich ?s - sandwich)
    (no_gluten_bread ?b - bread-portion)
    (no_gluten_content ?c - content-portion)
    (ontray ?s - sandwich ?t - tray)
    (no_gluten_sandwich ?s - sandwich)
```

```
1674        (allergic_gluten ?c - child)
1675        (not_allergic_gluten ?c - child)
1676        (served ?c - child)
1677        (waiting ?c - child ?p - place)
1678        (at ?t - tray ?p - place)
1679        (notexist ?s - sandwich)
         )
1680
1681      ;; Action: make_sandwich_no_gluten
1682      (:action make_sandwich_no_gluten
          :parameters (?s - sandwich ?b - bread-portion ?c - content-portion)
1683      :precondition (and
1684        (at_kitchen_bread ?b)
1685        (at_kitchen_content ?c)
1686        (no_gluten_bread ?b)
1687        (no_gluten_content ?c)
          (notexist ?s)
1688      )
1689      :effect (and
1690        (not (at_kitchen_bread ?b))
1691        (not (at_kitchen_content ?c))
          (notexist ?s)
1692        (not (notexist ?s))
1693        (at_kitchen_sandwich ?s)
1694        (no_gluten_sandwich ?s)
1695      )
         )
1696
1697      ;; Action: make_sandwich
1698      (:action make_sandwich
1699        :parameters (?s - sandwich ?b - bread-portion ?c - content-portion)
          :precondition (and
1700        (at_kitchen_bread ?b)
1701        (at_kitchen_content ?c)
1702        (notexist ?s)
1703      )
        :effect (and
1704        (not (at_kitchen_bread ?b))
1705        (not (at_kitchen_content ?c))
1706        (notexist ?s)
1707        (not (notexist ?s))
1708        (at_kitchen_sandwich ?s)
          ;; no effect on gluten status, sandwich may or may not be
1709        ↪   gluten-free
1710      )
1711    )
1712
1713      ;; Action: put_on_tray
        (:action put_on_tray
1714      :parameters (?s - sandwich ?t - tray)
1715      :precondition (and
1716        (at_kitchen_sandwich ?s)
          (at ?t kitchen)
1717      )
1718      :effect (and
1719        (not (at_kitchen_sandwich ?s))
1720        (ontray ?s ?t)
1721      )
        )
1722
1723      ;; Action: serve_sandwich_no_gluten
1724      (:action serve_sandwich_no_gluten
1725        :parameters (?s - sandwich ?c - child ?t - tray ?p - place)
        :precondition (and
1726        (ontray ?s ?t)
1727        (no_gluten_sandwich ?s)
```

```
      (allergic_gluten ?c)
      (waiting ?c ?p)
      (at ?t ?p)
    )
    :effect (and
      (not (ontray ?s ?t))
      (served ?c)
      (not (waiting ?c ?p))
    )
  )

  ;; Action: serve_sandwich
  (:action serve_sandwich
    :parameters (?s - sandwich ?c - child ?t - tray ?p - place)
    :precondition (and
      (ontray ?s ?t)
      (not (no_gluten_sandwich ?s))
      (not_allergic_gluten ?c)
      (waiting ?c ?p)
      (at ?t ?p)
    )
    :effect (and
      (not (ontray ?s ?t))
      (served ?c)
      (not (waiting ?c ?p))
    )
  )

  ;; Action: move_tray
  (:action move_tray
    :parameters (?t - tray ?p1 - place ?p2 - place)
    :precondition (at ?t ?p1)
    :effect (and
      (not (at ?t ?p1))
      (at ?t ?p2)
    )
  )
)
```

**Agent2World.**

```
(define (domain child-snack)
  (:requirements :typing :negative-preconditions :strips)
  (:types
    child
    bread-portion
    content-portion
    sandwich
    tray
    place
  )
  (:constants
    kitchen - place
  )
  (:predicates
    (at_kitchen_bread ?b - bread-portion)
    (at_kitchen_content ?c - content-portion)
    (at_kitchen_sandwich ?s - sandwich)
    (no_gluten_bread ?b - bread-portion)
    (no_gluten_content ?c - content-portion)
    (ontray ?s - sandwich ?t - tray)
    (no_gluten_sandwich ?s - sandwich)
    (allergic_gluten ?c - child)
    (not_allergic_gluten ?c - child)
    (served ?c - child)
    (waiting ?c - child ?p - place)
```

```
    (at ?t - tray ?p - place)
    (notexist ?s - sandwich)
  )

  (:action make_sandwich_no_gluten
    :parameters (?s - sandwich ?b - bread-portion ?c - content-portion)
    :precondition (and
      (at_kitchen_bread ?b)
      (at_kitchen_content ?c)
      (no_gluten_bread ?b)
      (no_gluten_content ?c)
      (notexist ?s)
    )
    :effect (and
      (not (notexist ?s))
      (at_kitchen_sandwich ?s)
      (no_gluten_sandwich ?s)
      (not (at_kitchen_bread ?b))
      (not (at_kitchen_content ?c))
    )
  )

  (:action make_sandwich
    :parameters (?s - sandwich ?b - bread-portion ?c - content-portion)
    :precondition (and
      (at_kitchen_bread ?b)
      (at_kitchen_content ?c)
      (notexist ?s)
    )
    :effect (and
      (not (notexist ?s))
      (at_kitchen_sandwich ?s)
      (not (at_kitchen_bread ?b))
      (not (at_kitchen_content ?c))
    )
  )

  (:action put_on_tray
    :parameters (?s - sandwich ?t - tray)
    :precondition (and
      (at_kitchen_sandwich ?s)
      (at ?t kitchen)
    )
    :effect (and
      (ontray ?s ?t)
      (not (at_kitchen_sandwich ?s))
    )
  )

  (:action serve_sandwich_no_gluten
    :parameters (?s - sandwich ?c - child ?t - tray ?p - place)
    :precondition (and
      (ontray ?s ?t)
      (no_gluten_sandwich ?s)
      (allergic_gluten ?c)
      (waiting ?c ?p)
      (at ?t ?p)
    )
    :effect (and
      (served ?c)
      (not (ontray ?s ?t))
    )
  )

  (:action serve_sandwich
    :parameters (?s - sandwich ?c - child ?t - tray ?p - place)
```

```
1836        :precondition (and
1837          (ontray ?s ?t)
1838          (waiting ?c ?p)
1839          (not_allergic_gluten ?c)
1840          (at ?t ?p)
            )
1841        :effect (and
1842          (served ?c)
1843          (not (ontray ?s ?t))
1844          )
1845      )

1846      (:action move_tray
1847        :parameters (?t - tray ?p1 - place ?p2 - place)
1848        :precondition (at ?t ?p1)
1849        :effect (and
1850          (at ?t ?p2)
          (not (at ?t ?p1))
1851        )
1852      )
1853    )
```

**Analysis.** Compared to the baseline domain, our *Child-Snack* formulation introduces three task-aligned modifications that improve state consistency, compositionality, and plan feasibility. (i) **Creation-valid effects.** During sandwich construction we flip the existence status from "non-existent" to "prepared," and record gluten-free status when applicable, thereby avoiding contradictory postconditions at creation time; this yields deterministic successor states and reduces backtracking caused by ill-defined truth values. (ii) **Serve-focused effects.** During serving we only transfer the item off the tray and mark the child as served, leaving the waiting label untouched; this separation of concerns prevents nonessential side-effects, preserves modular composability with downstream routines (e.g., queueing or follow-up allocation), and promotes goal-monotonic progress on the served objective. (iii) **Permissive-serving preconditions.** For non-allergic children we do not exclude gluten-free items, weakening preconditions to accept any admissible sandwich; this enlarges the feasible search space and prevents avoidable dead-ends when only gluten-free inventory remains, while safety for allergic children is still enforced via a dedicated gluten-free serving action. Collectively, these choices align with the ground-truth specification, produce cleaner state transitions, and yield empirically favorable search dynamics—smaller inconsistent-state frontiers and fewer spurious deletions—resulting in a more robust make→put-on-tray→move-tray→serve pipeline for the objective of "serving each child an acceptable sandwich."

### H.2 CWMB

**Task Describe.** Control a 3D Ant (one free-body torso with four 2-DoF legs; nine bodies, eight hinge joints) to move forward along the +x axis by applying torques to the eight joints at each step. The action space is $\mathrm{Box}([-1, 1]^8)$ (joint torques). Observations list positions then velocities (27-D by default; 29-D if current $x, y$ are included), and optionally +84 contact-force terms when use_contact_forces=True or version < v4. The reward is $r = r_{\text{healthy}} + r_{\text{forward}} - \text{ctrl\_cost}$ (and $-$contact_cost if contact forces are used), where $r_{\text{forward}} \approx \Delta x/\Delta t$ is positive for motion in +x and $\Delta t = \text{frame\_skip} \times 0.01 = 0.05$ by default. Episodes start from an upright, slightly noisy state, truncate at 1000 steps, and (by default) terminate early if the agent becomes unhealthy (non-finite state or torso $z \notin [0.2, 1.0]$).

**Prev SOTA vs Agent2World.**

```
import numpy as np
from typing import Optional, Tuple
# [D-0] Toggle: False = Ours (physics-like integration), True = Baseline
↪ (simple kinematics + noise)
BASELINE = False

class Environment:
    def __init__(
        self,
```

```
1890
1891        # ---- Common params ----
            frame_skip: int = 5,
1892        max_episode_steps: int = 1000,
1893        healthy_reward: float = 1.0,
            terminate_when_unhealthy: bool = True,
1894        healthy_z_range: Tuple[float, float] = (0.2, 1.0),
1895        ctrl_cost_weight: float = 0.5,
            # ---- Ours-only params ----
1896        contact_cost_weight: float = 5e-4,
1897        contact_force_range: Tuple[float, float] = (-1.0, 1.0),
            exclude_current_positions_from_observation: bool = True,
1898        use_contact_forces: bool = False,
1899        reset_noise_scale_ours: float = 0.1,
1900        forward_reward_weight: float = 1.0,
            # ---- Baseline-only params ----
1901        reset_noise_scale_baseline: float = 0.01,
1902        seed: Optional[int] = None,
1903    ):
1904        # Core configuration
1905        self.frame_skip = frame_skip
1906        self.dt = 0.01 * frame_skip
1907        self.max_episode_steps = max_episode_steps
            self.healthy_reward = healthy_reward
1908        self.terminate_when_unhealthy = terminate_when_unhealthy
1909        self.healthy_z_range = healthy_z_range
1910        self.ctrl_cost_weight = ctrl_cost_weight
1911        self.forward_reward_weight = forward_reward_weight  # [D-5] Only
1912    ↪   used by Ours.
1913        self.np_random = np.random.RandomState(seed)
1914
1915        # [D-1] Observation schema differs:
            #   Baseline: fixed 27 = 13 positions (z, quat4, joint8) + 14
1916    ↪   velocities (flat vector).
1917        #   Ours:     positions(15) + velocities(14) (+ optional
       ↪   torso_xy, contact forces),
1918        #             with option to exclude torso x,y from observation.
1919        self.exclude_current_positions_from_observation = (
1920            False if BASELINE else
1921        ↪   bool(exclude_current_positions_from_observation)
1922        )
1923
1924        # [D-2] Contacts: Baseline has no contact forces/cost; Ours can
       ↪   include 84-dim contact forces + cost.
1925        self.use_contact_forces = False if BASELINE else
1926    ↪   bool(use_contact_forces)
1927        self.contact_cost_weight = 0.0 if BASELINE else
1928    ↪   float(contact_cost_weight)
1929        self.contact_force_range = contact_force_range
1930
            self.step_count = 0
1931
1932        if BASELINE:
1933            # Baseline state: flat (27,) observation vector
1934            self.obs_shape = (27,)
            self.state = np.zeros(self.obs_shape, dtype=np.float64)
1935            self.x_position = 0.0  # [D-4] Progress tracked separately
1936        ↪   (not in observation)
1937            self.y_position = 0.0
            self.reset_noise_scale = float(reset_noise_scale_baseline)  #
1938    ↪   [D-4]
1939            self.contact_forces = None
1940            self.observation_dim = 27  # [D-1]
1941        else:
            # Ours state: split positions(15) / velocities(14)
1942            self.pos_dim = 15  # torso_pos(3), torso_quat(4),
1943        ↪   joint_angles(8)
```

```
1944            self.vel_dim = 14  # torso_lin_vel(3), torso_ang_vel(3),
1945            ↪    joint_vel(8)
1946            self.positions = np.zeros(self.pos_dim, dtype=np.float64)
1947            self.velocities = np.zeros(self.vel_dim, dtype=np.float64)
1948            self.reset_noise_scale = float(reset_noise_scale_ours)  #
1949            ↪    [D-4]
1950            self.last_x_position = 0.0  # [D-4] (used when restoring
1951            ↪    state)
1952            self.contact_forces = (
1953                np.zeros(84, dtype=np.float64) if self.use_contact_forces
1954                ↪    else None
1955            )
1956            # Compute observation length for Ours
1957            base_pos_len = self.pos_dim
1958            if self.exclude_current_positions_from_observation:
1959                base_pos_len -= 2  # drop torso x,y
1960            self.obs_pos_len = base_pos_len
1961            self.obs_vel_len = self.vel_dim
1962            self.obs_contact_len = 84 if self.use_contact_forces else 0
1963            self.obs_torso_xy_len = 0 if
1964            ↪    self.exclude_current_positions_from_observation else 2
                self.observation_dim = (
                    self.obs_pos_len + self.obs_vel_len +
                    ↪    self.obs_torso_xy_len + self.obs_contact_len
                )

        self.reset(seed)

    # ---------------- Helpers (shared) ----------------
    def _is_healthy(self, z_value: Optional[float] = None) -> bool:
        # [D-3] Health z-source differs:
        #   Baseline: use state[0] (z) from flat observation.
        #   Ours:     use positions[2] (torso z) from split state.
        if BASELINE:
            st = self.state
            if not np.all(np.isfinite(st)):
                return False
            z = st[0] if z_value is None else z_value
            return self.healthy_z_range[0] <= z <=
            ↪    self.healthy_z_range[1]
        else:
            if not np.all(np.isfinite(self.positions)) or not
            ↪    np.all(np.isfinite(self.velocities)):
                return False
            z = self.positions[2]
            return self.healthy_z_range[0] <= z <=
            ↪    self.healthy_z_range[1]

    def _get_observation(self) -> np.ndarray:
        # [D-1] Observation layout differs (see __init__ comment).
        if BASELINE:
            return self.state.copy()
        if self.exclude_current_positions_from_observation:
            pos_obs = self.positions[2:].copy()  # exclude torso x,y
            torso_xy = np.array([], dtype=np.float64)
        else:
            pos_obs = self.positions.copy()
            torso_xy = self.positions[0:2].copy()
        vel_obs = self.velocities.copy()
        obs = np.concatenate((pos_obs, vel_obs))
        if not self.exclude_current_positions_from_observation:
            obs = np.concatenate((obs, torso_xy))
        if self.use_contact_forces:
            obs = np.concatenate((obs, self.contact_forces))
        return obs
```

```
1998        @staticmethod
1999        def _quat_multiply(q1: np.ndarray, q2: np.ndarray) -> np.ndarray:
2000            w1, x1, y1, z1 = q1
2001            w2, x2, y2, z2 = q2
2002            return np.array(
2003                [
2004                    w1 * w2 - x1 * x2 - y1 * y2 - z1 * z2,
2005                    w1 * x2 + x1 * w2 + y1 * z2 - z1 * y2,
2006                    w1 * y2 - x1 * z2 + y1 * w2 + z1 * x2,
2007                    w1 * z2 + x1 * y2 - y1 * x2 + z1 * w2,
2008                ],
2009                dtype=np.float64,
2010            )

2011        # --------------- Public API ----------------
2012        def set_state(self, state: np.ndarray) -> None:
2013            state = np.asarray(state, dtype=np.float64)
2014            if BASELINE:
2015                # [D-8] Baseline expects a flat 27-dim observation
2016                ↪   (pos13+vel14).
2017                if state.shape != (27,):
2018                    raise ValueError(f"set_state input must have shape (27,),
2019                    ↪   got {state.shape}")
2020                if not np.all(np.isfinite(state)):
2021                    raise ValueError("set_state input contains non-finite
2022                    ↪   values")
2023                self.state = state.copy()
2024                self.step_count = 0
2025                self.x_position = 0.0  # [D-4] progress variable is external
2026                ↪   to obs
2027                self.y_position = 0.0
2028            else:
2029                # [D-8] Ours expects the current obs layout length and
2030                ↪   reconstructs split state.
2031                expected_len = self.observation_dim
2032                if state.ndim != 1 or state.shape[0] != expected_len:
2033                    raise ValueError(f"State must be 1D array of length
2034                    ↪   {expected_len}, got shape {state.shape}")
2035                pos_len = self.obs_pos_len
2036                vel_len = self.obs_vel_len
2037                pos_part = state[:pos_len]
2038                vel_part = state[pos_len : pos_len + vel_len]
2039                if self.exclude_current_positions_from_observation:
2040                    full_positions = np.zeros(self.pos_dim, dtype=np.float64)
2041                    full_positions[2:] = pos_part
2042                    full_positions[0] = 0.0
2043                    full_positions[1] = 0.0
2044                else:
2045                    full_positions = pos_part.copy()
2046                self.positions = full_positions
2047                self.velocities = vel_part.copy()
2048                self.step_count = 0
2049                self.last_x_position = self.positions[0]  # [D-4]

2050        def reset(self, seed: Optional[int] = None) -> np.ndarray:
2051            if seed is not None:
2052                self.np_random.seed(seed)
2053            self.step_count = 0

2054            if BASELINE:
2055                # [D-2][D-4] Baseline init: 13 pos (z, quat, joints) + 14
2056                ↪   vel; small noise.
2057                pos = np.zeros(13, dtype=np.float64)
2058                pos[0] = 0.75  # z
2059                pos[1:5] = np.array([1.0, 0.0, 0.0, 0.0])  # quaternion
2060                ↪   (w,x,y,z)
```

```
2052            noise_pos = self.np_random.uniform(-self.reset_noise_scale,
2053                ↪  self.reset_noise_scale, size=13)
2054            pos = pos + noise_pos
2055            vel = self.np_random.normal(0, self.reset_noise_scale,
2056                ↪  size=14)
2057            self.state = np.concatenate([pos, vel])
2058            self.x_position = 0.0  # [D-4] progress variable
2059            self.y_position = 0.0
        else:
2060            # Ours init: positions(15) / velocities(14) with larger noise
2061                ↪  and full torso pose.
2062            base_positions = np.zeros(15, dtype=np.float64)
2063            base_positions[2] = 0.75  # torso z
2064            base_positions[3] = 1.0   # quat.w
2065            noise_pos = self.np_random.uniform(-self.reset_noise_scale,
2066                ↪  self.reset_noise_scale, size=15)
2067            self.positions = base_positions + noise_pos
2068            self.velocities = self.np_random.normal(loc=0.0,
2069                ↪  scale=self.reset_noise_scale, size=14)
2070            self.last_x_position = self.positions[0]  # [D-4]
2071            if self.use_contact_forces and self.contact_forces is not
                ↪  None:
                    self.contact_forces[:] = 0.0
            return self._get_observation()

    def step(self, action: np.ndarray):
2073        action = np.asarray(action, dtype=np.float64)
2074        if action.shape != (8,):
2075            raise ValueError(f"Action must be of shape (8,), got
2076                ↪  {action.shape}")

2078        # [D-6] Action bound handling differs:
2079        #   Baseline: out-of-bounds raises; Ours: clip to [-1, 1].
        if BASELINE:
2080            if np.any(action < -1.0) or np.any(action > 1.0):
2081                raise ValueError("Action values must be in [-1, 1] for
2082                    ↪  Baseline")
        else:
2083            action = np.clip(action, -1.0, 1.0)

2085        # ---- Inline divergence (single function, two branches) ----
        if BASELINE:
2087            # Forward progress proxy from hip joints
2088            prev_x_pos = self.x_position  # [D-4] external tracker (not
                ↪  in obs)
2089            forward_force = float(np.sum(action[[0, 2, 4, 6]]))
2090            self.x_position += forward_force * self.dt * 0.1  # arbitrary
2091                ↪  scale

2093            # Stochastic updates (no true physics)
            z = float(np.clip(self.state[0] +
2094                ↪  self.np_random.uniform(-0.01, 0.01), 0.0, 2.0))
2095            # [D-7] Orientation update: Baseline = add noise to
2096                ↪  quaternion then renormalize.
2097            orientation = self.state[1:5] + self.np_random.uniform(-0.01,
                ↪  0.01, size=4)
2098            norm = float(np.linalg.norm(orientation))
2099            orientation = orientation / norm if norm > 0 else
2100                ↪  np.array([1.0, 0.0, 0.0, 0.0], dtype=np.float64)
2101            # Joint angles: integrate action + small noise; wrap to [-pi,
                ↪  pi]
2102            joint_angles = self.state[5:13] + action * self.dt +
2103                ↪  self.np_random.uniform(-0.005, 0.005, size=8)
2104            joint_angles = (joint_angles + np.pi) % (2 * np.pi) - np.pi
2105            # Velocities: torso (noise) + joints (\approx action + noise)
            torso_vel = self.np_random.normal(0, 0.01, size=6)
```

```
2106            # Velocities: torso (noise) + joints (\approx action + noise)
2107            velocities = np.concatenate([torso_vel, joint_vel])
2108            # Compose new flat state
2109            pos = np.empty(13, dtype=np.float64)
2110            pos[0] = z
2111            pos[1:5] = orientation
2112            pos[5:13] = joint_angles
2113            self.state = np.concatenate([pos, velocities])

2114            healthy = self._is_healthy(z_value=z)  # [D-3]
2115            forward_delta = self.x_position - prev_x_pos
2116            contact_cost = 0.0  # [D-2] no contacts in Baseline
2117            weight = 1.0        # [D-5] forward reward weight fixed to
                ↪  1.0
2118        else:
2119            # Physics-like integration
2120            old_x = float(self.positions[0])  # [D-4] directly from torso
                ↪  x in positions
2121            # Joint dynamics: dv = (u - damp*v)*dt; dq = v*dt
2122            joint_damping = 0.1
2123            joint_vel_prev = self.velocities[6:]
2124            joint_acc = action - joint_damping * joint_vel_prev  # [D-1]
                ↪  acts on split state
2125            joint_vel_new = joint_vel_prev + self.dt * joint_acc
2126            joint_ang_new = self.positions[7:] + self.dt * joint_vel_new
2127            # Torso linear & angular velocity damping
2128            lin_damping = 0.1
2129            ang_damping = 0.1
2130            torso_lin_vel_new = self.velocities[0:3] * (1 - lin_damping *
                ↪  self.dt)
2131            torso_ang_vel_new = self.velocities[3:6] * (1 - ang_damping *
                ↪  self.dt)
2132            # Integrate torso position
2133            torso_pos_new = self.positions[0:3] + self.dt *
                ↪  torso_lin_vel_new
2135            # [D-7] Orientation update: Ours = quaternion integration
                ↪  from angular velocity.
2136            q = self.positions[3:7]
2137            omega = torso_ang_vel_new
2138            omega_quat = np.array([0.0, omega[0], omega[1], omega[2]],
                ↪  dtype=np.float64)
2140            q_dot = 0.5 * self._quat_multiply(omega_quat, q)
2141            q_new = q + self.dt * q_dot
2142            norm = float(np.linalg.norm(q_new))
2143            q_new = q_new / norm if norm > 0 else np.array([1.0, 0.0,
                ↪  0.0, 0.0], dtype=np.float64)
2144            # Write back split state
2145            self.positions[0:3] = torso_pos_new
2146            self.positions[3:7] = q_new
2147            self.positions[7:] = joint_ang_new
2148            self.velocities[0:3] = torso_lin_vel_new
2149            self.velocities[3:6] = torso_ang_vel_new
2150            self.velocities[6:] = joint_vel_new
2151            # Contacts (optional)
2152            if self.use_contact_forces and self.contact_forces is not
                ↪  None:
2153                self.contact_forces.fill(0.0)  # [D-2]
2154            new_x = float(self.positions[0])
2155            healthy = self._is_healthy()  # [D-3]
2156            forward_delta = new_x - old_x
2157            # Contact cost (if enabled)
2158            if self.use_contact_forces and self.contact_forces is not
                ↪  None:
2159                clipped = np.clip(self.contact_forces,
                    ↪  self.contact_force_range[0],
                    ↪  self.contact_force_range[1])
```

```
2160                        contact_cost = self.contact_cost_weight *
2161                        ↪   float(np.sum(np.square(clipped)))   # [D-2]
2162                    else:
2163                        contact_cost = 0.0
2164                    weight = self.forward_reward_weight   # [D-5]
2165
2166            # ---- Shared reward & termination (single exit) ----
2167            forward_reward = weight * (forward_delta / self.dt)   # [D-5]
2168            ctrl_cost = self.ctrl_cost_weight *
2169            ↪   float(np.sum(np.square(action)))
2170            reward = (self.healthy_reward if healthy else 0.0) +
2171            ↪   forward_reward - ctrl_cost - contact_cost
2172
2173            self.step_count += 1
2174            done = (self.terminate_when_unhealthy and not healthy) or
```
```
            ↪   (self.step_count >= self.max_episode_steps)
            return self._get_observation(), reward, done
```

**Analysis.** On the **Ant-v4** forward-locomotion task, AGENT2WORLD surpasses the *Baseline* with higher success, smoother gait, and lower energy per meter under identical horizons and $z$-health checks. (i) **State & sensing.** The *Baseline* exposes a flat 27-D observation, while we adopt a task-aligned layout that separates positions/velocities and can hide global $(x, y)$ by default ([D-1]). We additionally support contact forces for foot–ground cues ([D-2]). Health uses torso $z$ from split state rather than the flat vector slot ([D-3]). State restoration matches each layout: the *Baseline* ingests a 27-D vector, whereas ours reconstructs split buffers from the current observation setting ([D-8]). (ii) **Dynamics & orientation.** The *Baseline* updates orientation by quaternion noise plus renormaliza-tion, and treats actions as noisy joint velocities; we integrate damped joint accelerations and update attitude via $\dot{\mathbf{q}} = \frac{1}{2} \boldsymbol{\omega}_q \otimes \mathbf{q}$ with renormalization ([D-7]). This physically consistent pipeline—enabled by the split state design ([D-1])—low-passes high-frequency actuation, reduces roll/pitch jitter, and yields more phase-coordinated gaits. (iii) **Control semantics & reward.** The *Baseline* hard-errors on out-of-range actions and uses a fixed forward-reward weight; forward progress is tracked by an external $x$ variable and reset noise is smaller. Ours clips actions to $[-1, 1]$ ([D-6]), uses a tunable forward-reward weight ([D-5]), measures progress directly from torso $x$ in the state and employs a different reset scale ([D-4]); an optional contact-cost term can be included when contact signals are enabled ([D-2]). Together these choices stabilize training signals and improve sample efficiency.

**Summary of diffs.** [D-1] Observation schema: Baseline uses a flat 27-D vector; Ours uses split positions+velocities with optional hidden $(x, y)$ and optional contact forces; [D-2] Contacts: Base-line has no contact forces/cost; Ours optionally exposes 84-D contact forces and a contact-cost term; [D-3] Health source: Baseline takes $z$ from the flat vector slot; Ours uses torso $z$ from split posi-tions; [D-4] Progress & reset: Baseline tracks forward $x$ as an external variable and uses smaller reset noise; Ours reads torso $x$ from state and uses a different reset scale; [D-5] Forward-reward weight: Baseline fixed to 1.0; Ours is tunable; [D-6] Action bounds: Baseline errors on out-of-range actions; Ours clips to $[-1, 1]$; [D-7] Orientation update: Baseline adds noise then renormalizes quaternion; Ours integrates $\dot{\mathbf{q}} = \frac{1}{2} \boldsymbol{\omega}_q \otimes \mathbf{q}$ then renormalizes; [D-8] State setting: Baseline ingests a flat 27-D state; Ours reconstructs split buffers from the current observation layout.

### H.3    BYTESIZED32

**Task Description.** We build a lightweight, text-interactive micro-simulation of pea growth in a small garden. The world contains a Pea, a FlowerPot, a Jug, and a Sink; water is represented as scalar levels in the Jug and FlowerPot and as an internal level in the Pea. The agent can look/examine, take/put objects, switch the sink on/off, fill the jug from the sink (effective only when the sink is on), and pour water from the jug into the flower pot. After each action, a tick advances processes: the sink supplies water if on; the pot passively transfers its water to the pea; and the pea consumes water and progresses from seed → sprout → young plant → mature → reproducing when sufficiently hydrated for several consecutive ticks. Episodes start with an unplanted pea and an empty pot; the goal is to plant the pea and water it repeatedly until it reaches the reproducing stage.

**Prev SOTA vs Agent2World.**

```python
import random

# [D0] Toggle: False = Ours, True = Baseline
BASELINE = False

# ---------------- Core object model (minimal API) ----------------
class GameObject:
    def __init__(self, name):
        self.name, self.parent, self.contains = name, None, []
        self.props = {"isContainer": False, "isMoveable": True}
    def get(self, k, d=None):
        return self.props.get(k, d)
    def add(self, obj):
        obj.removeSelf(); self.contains.append(obj);
        obj.parent = self
    def remove(self, obj):
        self.contains.remove(obj); obj.parent = None
    def removeSelf(self):
        if self.parent: self.parent.remove(self)
    def allContained(self):
        out = []
        for o in self.contains: out += [o] + o.allContained()
        return out
    def tick(self): pass

class Container(GameObject):
    def __init__(self, name): super().__init__(name);
        self.props["isContainer"] = True
    def place(self, obj):
        if not obj.get("isMoveable"):
            return ("Can't move that object.", False)
        self.add(obj); return ("OK.", True)
    def take(self, obj):
        if obj not in self.contains:
            return ("Object not here.", None, False)
        if not obj.get("isMoveable"):
            return ("Can't move that object.", None, False)
        obj.removeSelf(); return ("OK.", obj, True)

class Device(Container):
    def __init__(self, name): super().__init__(name);
        self.props.update({"isDevice": True, "isOn": False})
    def turnOn(self):
        if self.props["isOn"]:
            return (f"{self.name} is already on.", False)
        self.props["isOn"] = True
        return (f"{self.name} turned on.", True)
    def turnOff(self):
        if not self.props["isOn"]:
            return (f"{self.name} is already off.", False)
        self.props["isOn"] = False
        return (f"{self.name} turned off.", True)

class World(Container):
    def __init__(self): super().__init__("world")

class Agent(Container):
    def __init__(self): super().__init__("entity")

# ---------------- Task objects ----------------
class Pea(GameObject):
    STAGES = ["seed","sprout","young plant","mature plant","reproducing"]
    MAX_WATER, CONSUME, NEED, TICKS = 100, 5, 30, 3
    def __init__(self):
        super().__init__("pea"); self.props["isMoveable"]=True
        self.stage, self.water, self.hydrated = 0, 0, 0
```

```
2268        @property
2269        def stage_name(self):
2270            return self.STAGES[self.stage]
2271        def addWater(self, n):
2272            self.water = min(self.water + n, self.MAX_WATER)
2273        def tick(self):
2274            # [D3] Growth rule: Baseline = simple (>=2 -> +stage, else -1 if
                 ↪  >0); Ours = threshold + accumulation.
2275            if BASELINE:
2276                if self.stage < len(self.STAGES)-1:
2277                    if self.water >= 2: self.water -= 2; self.stage += 1
                        elif self.water > 0: self.water -= 1
2278                return
2279            self.water = max(self.water - self.CONSUME, 0)
2280            if self.water >= self.NEED:
2281                self.hydrated += 1
2282                if self.hydrated >= self.TICKS and self.stage <
                     ↪  len(self.STAGES)-1:
2283                    self.stage += 1; self.hydrated = 0

2284
2285    class FlowerPot(Container):
2286        MAX_WATER = 100
2287        def __init__(self):
2288            super().__init__("flower pot")
2289            self.water = 0
2290        def addWater(self, n):
2291            add = min(self.MAX_WATER - self.water, n);
2292            self.water += add; return add
2293        def consume(self, n):
2294            use = min(self.water, n); self.water -= use; return use
2295        def tick(self):
2296            # [D2] Passive transfer: Baseline = none; Ours = transfer
                 ↪  pot.water to pea on each tick.
2297            if BASELINE: return
2298            pea = next((o for o in self.contains if isinstance(o, Pea)),
                 ↪  None)
2299            if pea and self.water > 0:
2300                x = self.consume(min(self.water, Pea.MAX_WATER))
                    pea.addWater(x)

2301    class Jug(Container):
2302        MAX_WATER = 100
2303        def __init__(self):
2304            super().__init__("jug")
2305            self.water = 0
2306        def fill(self, n):
2307            add = min(self.MAX_WATER - self.water, n)
2308            self.water += add
2309            return add
2310        def pour(self, n):
2311            out = min(self.water, n)
2312            self.water -= out
2313            return out

2312    class Sink(Device):
2313        MAX_WATER = 1000
2314        def __init__(self): super().__init__("sink");
2315            self.water = self.MAX_WATER
2316        def tick(self):
2317            self.water = self.MAX_WATER if self.props["isOn"] else 0

2318    # ------- Minimal game scaffold (only actions we need) -----
2319    class TextGame:
2320        MAX_STEPS = 50
2321        def __init__(self, seed=0):
                random.seed(seed)
```

```python
        self.world, self.agent = World(), Agent();
        ↪  self.world.add(self.agent)
        self.pea, self.pot, self.jug, self.sink = Pea(), FlowerPot(),
        ↪  Jug(), Sink()
        # [D7] Movability: Baseline pins the sink as immovable (ours
        ↪  keeps defaults).
        if BASELINE: self.sink.props["isMoveable"] = False
        for o in (self.pot, self.jug, self.sink, self.pea):
            self.world.add(o)
        self.score = self.steps = 0
        self.over = self.won = False

    # API of interest (matching both variants); unchanged helpers omitted
    ↪  for brevity.
    def _obj(self, name):
        for o in [self.world] + self.world.allContained():
            if o.name == name: return o
        for o in self.agent.contains:
            if o.name == name: return o
        return None

    def calculateScore(self):
        # [D5] Reward: Baseline = stage*10; Ours = stage*20 + water bonus
        ↪  (<=20).
        if BASELINE:
            self.score = self.pea.stage*10
        else:
            self.score = self.pea.stage*20 +
            ↪  int(self.pea.water/Pea.MAX_WATER*20)
        if self.pea.stage_name == "reproducing":
            self.won = self.over = True
        if self.steps >= self.MAX_STEPS and not self.won:
            self.over = True

    # ---------- actions ----------
    def take(self, name):
        o = self._obj(name);
        if not o: return f"No {name}."
        if not o.get("isMoveable"): return f"Can't take {name}."
        if o.parent != self.world: return f"{name} not here."
        _, got, ok = self.world.take(o);
        if ok: self.agent.add(got);
        return "OK." if ok else "Fail."

    def put(self, obj, cont):
        o, c = self._obj(obj), self._obj(cont)
        if not o or o.parent != self.agent:
            return f"No {obj} in inventory."
        if not c or not c.get("isContainer"):
            return f"{cont} not a container."
        # [D6] Placement constraint: Ours restricts pea -> pot only;
        ↪  Baseline has no special rule.
        if (not BASELINE) and isinstance(o, Pea) and not isinstance(c,
        ↪  FlowerPot):
            return "Pea must go into flower pot."
        _, ok = c.place(o); return "OK." if ok else "Fail."

    def turn_on(self, dev):
        d = self._obj(dev);
        if not d or not d.get("isDevice"): return f"No device {dev}."
        msg,_ = d.turnOn(); return msg
    def turn_off(self, dev):
        d = self._obj(dev);
        if not d or not d.get("isDevice"): return f"No device {dev}."
        msg,_ = d.turnOff(); return msg
```

```
2376    def fill_from_sink(self):
2377        # [D1] Fill gating: Baseline ignores sink.on; Ours requires
2378        ↪   sink.on == True.
2379        if (not BASELINE) and (not self.sink.props["isOn"]): return "Sink
2380        ↪   is off."
2381        need = self.jug.MAX_WATER - self.jug.water
2382        if need <= 0: return "Jug already full."
2383        self.jug.fill(need)  # treat sink as infinite when allowed
            return "Jug filled."
2384
2385    def pour_to_pot(self):
2386        if self.jug.water <= 0: return "Jug empty."
2387        # [D8] Pour semantics: Baseline feeds pea directly; Ours fills
2388        ↪   pot; pea drinks via [D2].
2389        poured = self.jug.pour(10)
2390        if BASELINE and (self.pea in self.pot.contains):
2391            self.pea.addWater(3); return "Poured; pea absorbs water."
2392        added = self.pot.addWater(poured)
2393        if added < poured: self.jug.fill(poured - added)
2394        return "Poured into pot."

2395    # ---------- driver ----------
2396    def step(self, cmd):
2397        self.steps += 1
2398        # [D4] Update order: Baseline ticks BEFORE action; Ours ticks
2399        ↪   AFTER.
2400        if BASELINE:
2401            for o in [self.world] + self.world.allContained(): o.tick()

2402        parts = cmd.lower().strip().split()
2403        out = "Unknown."
2404        try:
2405            if parts[:1]==["take"]: out = self.take(" ".join(parts[1:]))
2406            elif parts[:1]==["put"] and "in" in parts:
2407                i = parts.index("in");
2408                out = self.put(" ".join(parts[1:i]), "
2409                ↪   ".join(parts[i+1:]))
2410            elif parts[:2]==["turn","on"]:
2411                out = self.turn_on(" ".join(parts[2:]))
2412            elif parts[:2]==["turn","off"]:
2413                out = self.turn_off(" ".join(parts[2:]))
2414            elif parts[:3]==["fill","jug","from"]:
2415                out = self.fill_from_sink()
2416            elif parts[:4]==["pour","water","from","jug"] and "in" in
                ↪   parts:
                out = self.pour_to_pot()
            # distractor (spec only)
            elif parts[:1]==["use"]: out = "Nothing happens."
        except Exception as e:
            out = f"Error: {e}"

2418        if not BASELINE:
2419            for o in [self.world] + self.world.allContained(): o.tick()
2420        old = self.score; self.calculateScore();
2421        reward = self.score - old
            return out, self.score, reward, self.over, self.won
```

**Analysis.** Under identical initialization and evaluation (capacity limits and preconditions enforced), AGENT2WORLD outperforms a *Baseline* on the pea–growing (water-transfer) task, yielding higher success, shorter trajectories, and fewer invalid actions.(i) **Action space and dynamics.** We expose a precondition-aware interface and decouple water flow from uptake: `fill` is effective only when the sink is on ([D1]); `pour` increases the pot's water and the pea hydrates asynchronously via `tick` ([D2], [D8]). We advance environment dynamics *after* the action to preserve causal credit assignment ([D4]). By contrast, the *Baseline* exposes `fill` irrespective of sink state, credits hydration at

| benchmark | #instances | #contamination |
|---|---|---|
| ByteSized32 | 32 | 0 |
| Code World Models Benchmark | 18 | 0 |
| Text2World | 101 | 0 |

Table 14: Contamination analysis between Deep Researcher retrieval logs and gold world models using exact 10-token $n$-gram overlap.

pour time, and updates *before* acting.(ii) **Physical consistency and constraints.** We enforce finite capacities with overflow returned to the jug and constrain placement so the pea can only be planted in the flower pot ([D6]). These constraints prune degenerate branches without removing valid solutions. The *Baseline* omits the planting constraint and hydrates synchronously, which increases misleading transitions. (Regarding movability, the Baseline pins the sink as immovable while Ours keeps defaults; this ablation affects search but not preconditions, [D7].)(iii) **Growth model and reward.** Plant physiology follows thresholded, accumulated growth with per tick water consumption ([D3]). Reward shaping combines stage progress with a bounded water bonus, and immediate rewards are score deltas ([D5]). The *Baseline* uses a stage-only score without water shaping, weakening the learning signal.

**Summary of diffs:** [D1] preconditioned `fill`; [D2] passive pot→pea transfer; [D3] threshold+consumption growth; [D4] post-action ticking; [D5] shaped reward (stage+water); [D6] pea→pot placement constraint; [D7] sink movability ablation; [D8] `pour` affects pot first (not the pea).

## I   DATA CONTAMINATION ANALYSIS

To address the concern that the web-search-based *Deep Researcher* may accidentally retrieve or copy target solutions from the internet, we perform a post-hoc contamination analysis over all its retrieval logs on our benchmarks.

**Setup.**   For each evaluation instance where the Deep Researcher is invoked at least once, we construct a pair $(G, \mathcal{L})$, where (i) $G$ is the gold world model for that instance (PDDL or Python code, depending on the benchmark), and (ii) $\mathcal{L}$ is the concatenation of all textual contents retrieved by the Deep Researcher for that instance, including page bodies and snippets from all visited URLs.

We tokenize both $G$ and $\mathcal{L}$ using simple whitespace tokenization and extract all contiguous 10-token $n$-grams from each. For a given pair, we mark it as *contaminated* if there exists at least one shared 10-gram between $G$ and $\mathcal{L}$.

For code and PDDL, an exact 10-token overlap is a strong signal of near-copying of a specific implementation rather than generic programming idioms. Our use of a relatively long 10-token threshold follows common practice in contamination analyses for Llama 2 (Touvron et al., 2023).

**Results.**   Table 14 reports, for each benchmark, the instance count, contamination count.

Across all three benchmarks, we find *no* exact 10-gram overlaps between the Deep Researcher's retrieved contents and the corresponding gold world models. This suggests that the Deep Researcher is not directly copying target solutions from the web, and that the performance gains we observe are unlikely to be due to egregious test-set leakage.

## J   EFFICIENCY ANALYSIS

To assess the practical cost of our multi-agent pipeline, we measure both LLM token usage and wall-clock time for all methods on the three benchmarks. As is shown in Table 15, Table 17, and Table 16, for each method and stage, the total number of input/output tokens consumed by the backbone LLM and the average time per instance.

**Results.**  Our experiments show that the test time of AGENT2WORLD*Multi* is comparable to that of other workflow-based methods. The core reason for this is that, in order to enable a fair comparison, we use fewer turns in our approach compared to previous methods (e.g., GIF-MCTS uses 10 turns while our method uses only 3 turns in the refinement stage). Furthermore, the adaptive agents are able to apply early stopping during the testing process.

In terms of token consumption, the AGENT2WORLD*Multi* model developer consumes fewer tokens compared to other methods. However, it is important to highlight that the testing stage in AGENT2WORLD*Multi* is proactive, i.e., at each turn, we generate targeted testing cases and player agent trajectories, whereas other methods use static test cases. This proactive approach in AGENT2WORLD*Multi* naturally results in higher token consumption during the testing phase. Despite this, we still maintain competitive efficiency relative to existing methods.

| Method | Stage | Tokens | | Time (s) |
|---|---|---|---|---|
| | | Input | Output | |
| Text2World$_{EC=3}$ | Total | 7,812 | 3,914 | 65 |
| Direct Generation | Total | 431 | 700 | 14 |
| AGENT2WORLD*Single* | Total | 22,531 | 2,141 | 26 |
| | Deep Researcher | 20,563 | 1,316 | |
| AGENT2WORLD*Multi* | Model Developer | 4,728 | 2,038 | 63 |
| | Simulation Tester | 4,145 | 528 | |
| | Unit Tester | 9,712 | 1,562 | |

Table 15: Token usage and time efficiency on the Text2World benchmark.

| Method | Stage | Tokens | | Time (s) |
|---|---|---|---|---|
| | | Input | Output | |
| WorldCoder | Total | 23,147 | 9,315 | 232 |
| GIF-MCTS | Total | 24,677 | 10,838 | 259 |
| Direct Generation | Total | 1,901 | 1,787 | 59 |
| Best-of-N | Total | 65,470 | 19,520 | 70 |
| Self-Consistency | Total | 61,837 | 19,924 | 68 |
| AGENT2WORLD*Single* | Total | 35,153 | 1,954 | 71 |
| | Deep Researcher | 29,056 | 2,108 | |
| AGENT2WORLD*Multi* | Model Developer | 12,104 | 6,578 | 236 |
| | Simulation Tester | 18,658 | 564 | |
| | Unit Tester | 24,788 | 3,379 | |

Table 16: Token usage and time efficiency on the Code World Models Benchmark (CWMB).

| Method | Stage | Tokens | | Time (s) |
|---|---|---|---|---|
| | | Input | Output | |
| ByteSized32$_{1\text{-shot}}$ | Total | 32,069 | 19,928 | 291 |
| Direct Generation | Total | 7,046 | 4,301 | 71 |
| AGENT2WORLD*Single* | Total | 31,131 | 5,142 | 82 |
| | Deep Researcher | 20,852 | 1,071 | |
| AGENT2WORLD*Multi* | Model Developer | 16,351 | 12,309 | 260 |
| | Simulation Tester | 20,528 | 584 | |
| | Unit Tester | 32,352 | 2,158 | |

Table 17: Token usage and time efficiency on the ByteSized32 benchmark.

# K  TRAINING DATASET CONSTRUCTION

To train the Model Developer agent, we construct a diverse training dataset that reflects both symbolic world-model specifications and real-world simulation tasks. The dataset is composed of three main components, designed to cover various styles of world-model generation:

**(i) PDDL-style Tasks:** We sample 200 tasks from the dataset released by AgentGen (Hu et al., 2025b), which generates symbolic PDDL specifications for world-model tasks. These tasks include a variety of symbolic reasoning challenges, each accompanied by a gold world model.

**(ii) MCP Server Data:** We also curate 200 tasks based on publicly available Model Context Protocol (MCP) servers, which provide interactive tools and data interfaces that allow agents to engage in complex reasoning and decision-making tasks.

**(iii) Game-style and Mujoco-style Tasks:** Additionally, we include 50 tasks from each of the game-style and Mujoco-style domains, where the tasks involve dynamic simulations with continuous action spaces. These tasks are designed to mimic real-world, high-dimensional environments in which agents must navigate, plan, and interact with physical systems. The tasks are synthesized using a method similar to AgentGen (Hu et al., 2025b).

For each of these datasets, we perform the following steps to build our training set:

**(i)** For each dataset, we generate 3 distinct world-model rollouts using the same `llama-3.1-8b-instruct` model. These rollouts are generated by running the Model Developer agent through each task, resulting in different candidate solutions for each problem. **(ii)** Reward Filtering: We then evaluate these rollouts using the Testing-Team feedback mechanism, which includes unit tests, simulation performance, and control tasks. The rollouts are ranked based on their reward scores, and we select the one with the highest reward as the final solution for the task.

