# OpenReview forum: "Agent2World: A Unified LLM-based Multi-Agent Framework for Symbolic World-Model Generation"
_ICLR.cc/2026/Conference — Submitted to ICLR 2026_

### Official Review · Reviewer_J92f · 2025-10-28

**Soundness:** 2
**Presentation:** 3
**Contribution:** 2
**Rating:** 4
**Confidence:** 4

**Summary:**

The paper introduces a novel curated multi-agent pipeline comprising LLM agents for symbolic world model code synthesis. This complex, compulsory pipeline consists of: (1) a Deep Researcher for web search, (2) a Model Developer for code synthesis, and (3) a Unit Test Generator coupled with Simulation Evaluators. The pipeline is evaluated on the Text2World, CWMB, and ByteSized32 benchmarks. Notably, it operates as a fully test-time scaling approach, with no training involved.
From a theoretical perspective, related concepts appear in ReAct [1] and have been extensively explored in multi-agent tool-use frameworks (e.g., Agent2World_multi) since 2023 [2][3][4]. Thus, the paper's primary contribution lies in adapting tool-augmented LLM agent pipelines to the domain of generating symbolic world models.

ReAct:
[1] https://arxiv.org/pdf/2210.03629
An architecture of LLM agent w\o mentioning world model:
[2] https://lilianweng.github.io/posts/2023-06-23-agent/
Simulator
[3] https://arxiv.org/pdf/2507.23773
[4] World coder
[5] Language models, agent models, and world models: The law for
machine reasoning and planning
and many more...

**Strengths:**

I think the comparison experiment is thorough, and it seems like the compulsory pipeline is stronger than the baselines.

**Weaknesses:**

As I mentioned in the previous sections of the review, the paper demonstrates impressive performance of the multi-agent pipeline in PDDL code synthesis. However, the construction of such a composite pipeline is hardly novel in 2025. Indeed, the contribution cannot lie in the pipeline itself: tool-augmented pipelines were first introduced in Toolformer [1]; Lilian Weng outlined a general architecture for tool-using LLMs in her blog [2]; CAMEL open-sourced a comprehensive framework for LLM reasoning [3]; self-refinement emerged in ReAct and Reflection [4,5]; the concept of LLMs as world simulators was advanced in [6,7]; and LLM-guided reasoning with unit tests appeared in [8], and many more. Thus, none of the individual modules in Agent2World are originally proposed here. The primary contribution instead resides in the effective composition of these existing modules and their targeted application to PDDL code synthesis.

That said, my concerns lie in the following aspects:

1. **Potential contamination in the deep researcher procedure**: I am worried that ground-truth answers could be retrieved from the web. And how to prevent the situation is not mentioned in the paper.

2. **Token efficiency**: The Agent2World pipeline should consume significantly more tokens, yet Figure 7 shows substantial performance gains with only modest token increases in the multi-agent method. It remains unclear how tokens are computed for both methods. Given the three-stage design, the multi-agent approach should reveal far more information and thus consume far more tokens; the minimal token increase is surprising and unexplained.

3. **Unclear experimental settings**: It is not specified which LLM is used in each comparison experiment or in each stage of Agent2World. Even if the code-generation LLM is held constant, could stronger LLMs in the deep researcher or unit-test modules (e.g., better at PDDL understanding) be driving the gains due to unbalanced LLM usage?

4. **Limited comparison methods**: For example, in Text2World, the authors compare only with direct search, a single-agent baseline, and the original Text2World method. Many more test-time scaling techniques (e.g., MCTS, World Coder, advanced prompt engineering) should be included.

5. **Beyond zero-shot inference**: Although the paper is not about learning, could training-based methods enhance WM synthesis? I ask this because some LLMs may have seen PDDL data from GitHub during pretraining, giving them an inherent advantage.

According to the above reasons, I think a score of 4 is the maximum score I can give to this paper, considering some of the concerns would be addressed by the authors in the rebuttal sections.

[1] https://arxiv.org/pdf/2302.04761

[2] https://lilianweng.github.io/posts/2023-06-23-agent/

[3] https://github.com/camel-ai/camel

[4] https://arxiv.org/pdf/2210.03629

[5] https://arxiv.org/abs/2303.11366

[6] https://arxiv.org/pdf/2305.14992

[7] Language models, agent models, and world models: The law for machine reasoning and planning

[8] https://arxiv.org/html/2508.00408v1

**Questions:**

See Weaknesses

---

> ### Author Response · Authors · 2025-11-26
> **Response to Reviewer J92f (1)**
>
> Dear Reviewer J92f,
>
> Thank you for your thoughtful review and for recognizing the contributions of our work! We appreciate your acknowledgment of the thoroughness of our experiments and the strength of the multi-agent pipeline.
>
> We have conducted **4** additional experiments and carefully revised the manuscript (highlighted by **red**) to address the points you've raised, specifically:
>
> * Clarification of the novelty of our framework (W0).
> * Deep Researcher safeguards and a data contamination analysis experiment (W1, Appendix I, Line 2454).
> * Clarification on Figure 7 and a detailed explanation of how token consumption is calculated   (W2.1&W2.2).
> * A token/time cost experiment to further illustrate the token efficiency of Agent2World-Multi. (W2.3, Appendix J, Line 2479)
> * Clarification of LLM configuration. (W3)
> * Additional test-time scaling baselines and experimental results. (W4, Table 2&Section 4.1)
> * Training-based Agent2World-Multi and corresponding results. (W5, Table 2&Section 4.1)
>
> We hope that the revisions provide sufficient clarification. Please let us know if you have any further questions or suggestions!
>
> ---

---

> > ### Author Response · Authors · 2025-11-26
> > **Response to Reviewer J92f (2)**
> >
> > ### W0. Concern that Agent2World is “just a composition” of existing tool-augmented and self-refinement modules, so the contribution is limited.
> >
> > **A:** Thank you for raising this important point about novelty. We fully agree that our system builds on a rich line of work on tool-augmented LLMs and self-refinement (e.g., Toolformer, CAMEL, ReAct, Reflection, etc.). Our claim is not that any individual component (web search, code execution, or unit tests) is novel in isolation. Rather, Agent2World focuses on a **new problem setting**—symbolic world-model generation.
> >
> > Concretely:
> >
> > 1. **Beyond generic tool-use / multi-agent frameworks (and our ReAct baseline).** The works you cite are pioneering and extremely valuable for general tool use and reasoning. However, they are not designed for *symbolic world-model generation* as a primary objective, and directly applying them does not automatically yield strong world models. In our paper, we explicitly include a strong **ReAct-style single-agent baseline (`Agent2World-Single`)** that uses the same tools and backbone model. This baseline represents the straightforward application of a generic tool-using agent to our setting, yet it still underperforms `Agent2World-Multi`. This comparison suggests that simply adopting an existing ReAct-like pipeline is insufficient, and that designing an agentic framework specifically for symbolic world-model generation matters. Because symbolic world-model generation is an important and emerging problem, we believe that demonstrating an effective agentic framework for this task is itself a substantive contribution.
> >
> > 2. **From scripted workflows to adaptive tool use.** Prior world-model–generation systems (e.g., Text2World, WorldCoder, GIF-MCTS, ByteSized32) use scripted draft–repair workflows where the control flow is fixed by the designer and does not change based on intermediate observations. In contrast, Agent2World’s agents proactively decide **when and how** to call tools (web search, code execution, simulation, unit tests) based on the current state of the world model, rather than following a hard-coded sequence. This adaptivity is crucial for identifying missing constraints and iteratively improving the model.
> >
> > 3. **Unified cross-representation framework.** We propose a unified framework that operates across both PDDL-style and executable-code world models, using the same three high-level roles (Deep Researcher → Model Developer → Testing Team). This is an attempt to be *general within the symbolic world-model generation setting*, rather than designing separate, representation-specific systems.
> >
> > 4. **Formulating symbolic world-model generation as an agentic MDP and training world-model agents.** A key idea we want to convey is that symbolic world-model generation can be viewed as an **agentic process**—in particular, the interaction between the Model Developer and the Testing Team naturally defines an MDP. In the revised paper (Section “Learning World-Model Agents from Testing-Team Feedback”), we model the Model Developer as an LLM-based agent operating in an MDP $\mathcal{M}_{\text{MD}} = (\mathcal{S}, \mathcal{A}, \mathcal{T}, \mathcal{R}, \gamma)$, where a state $(s_t \in \mathcal{S})$ encodes the natural-language specification or diagnostics produced by the Testing Team; an action $(a_t \in \mathcal{A})$ corresponds to emitting a new world-model implementation or a code patch; and the transition $(\mathcal{T})$ is realized by executing the candidate world model in the sandbox and re-invoking the Testing Team, which produces updated diagnostics that are appended to form $(s*{t+1})$. In this view, the Testing Team acts as a reward generator and these signals are aggregated into a scalar reward $(\mathcal{R}(s_t, a_t))$ that reflects both local predictive correctness and task-level utility of the world model. Using this view, we construct an **agent-in-the-loop supervised fine-tuning (SFT)** setup on CWMB: we roll out Agent2World-Multi, filter trajectories using the Testing Team’s reward, treat the remaining interaction traces as demonstrations, and fine-tune the same llama-3.1-8b-instruct backbone on these demonstrations. At test time we freeze the Testing Team and evaluate Agent2World-Multi (SFT) under the same protocol as Agent2World-Multi on held-out CWMB tasks. As summarized in Table 5 (Line 486), this improves both accuracy and normalized return over the baselines. We regard this as a first step toward reinforcement learning in this MDP, which we plan to explore and extend to other benchmarks in the camera-ready version.
> >
> > We hope this makes our position clearer: we fully agree that modern systems should build on prior primitives. Our contribution is to show that an adaptive multi-agent architecture, together with world-model–specific evaluation and an agentic training substrate, leads to new empirical insights and strong performance on symbolic world-model generation benchmarks.

---

> ### Author Response · Authors · 2025-11-26
> **Response to Reviewer J92f (3)**
>
> ### W2. Token efficiency
>
> Thank you for your insightful comment. We appreciate the opportunity to clarify this point.
>
> **W2.1: Clarification on Specification Efficiency and Figure 7**
>
> **A:** First, to address the specific concern about **Figure 7** (Section 5.5), we would like to clarify that we are calculating **specification efficiency** rather than just token consumption. Specification efficiency is a measure of how effectively the agents use tokens to generate a valid world model with minimal computation. The formula for **specification efficiency** is:
>
> $$
> \text{Specification Efficiency} = \frac{\text{Performance Gain}}{\text{Token Consumption}}
> $$
>
> This efficiency measure reflects not just the increase in performance but also how well the system manages token costs. This is why the token increase in the multi-agent approach appears modest relative to the substantial performance gains shown in **Figure 7**.
>
> **W2.2: It remains unclear how tokens are computed for both methods.**
>
> **A:** To clarify how **tokens are computed** for both methods, we follow a **Markov Decision Process (MDP)** formalization, where the agent's decision-making process involves interleaving reasoning and tool calls. In this formalization, tokens are computed at each step during the agent's interactions, which includes every **tool call** made by the agent within the conversation.
>
> Specifically, for both methods, we track the token consumption at each **agent-tool interaction**. For every turn, the **input tokens** (i.e., the tokens used by the agent to generate a response) and the **output tokens** (i.e., the tokens produced by the agent’s response) are counted. This is different from **workflow-based baselines** that typically rely on **one-time LLM calls** without maintaining a history of previous states.
>
> The token consumption calculation for each method can be expressed as:
>
> $$
> T = \sum_{k=1}^{N} ( L_{\mathrm{in}, k} + L_{\mathrm{out}, k} )
> $$
>
> Where $N$ is the number of steps in the conversation.
>
> We hope this explanation clarifies how token consumption is computed for both methods. Let us know if you need further clarification.
>
>
> **W2.3: Given the three-stage design, the multi-agent approach should reveal far more information and thus consume far more tokens; the minimal token increase is surprising and unexplained.**
>
> **A:** Thank you for this observation. We would like to clarify why the increase in tokens for Agent2World-Multi is modest despite its three-stage design.
>
> To deeply analyze this aspect, we conducted an additional token/time cost experiment, as presented in **Appendix J** (see Line 2479). We can draw several conclusions from the results:
>
> 1. During the **Model Developer** stage, the token consumption is lower. This is because, as discussed in **W2.2**, our agent formalization is based on MDP, whereas workflow-based methods rely on one-time LLM calls without maintaining history, which inherently puts our method at a disadvantage in token consumption. To maintain fairness in comparison, we limited Agent2World's refinement turns to 3, while baselines use 10 turns.
>
> 2. In the **Refinement stage**, our approach generates different testing cases at each turn to maximally test the potential errors in the world model. In contrast, other baselines rely on static unit-tests. Although this approach consumes additional tokens, as shown in ourAblation Study (Section 5.1), it provides substantial gains in performance, setting our method apart from the other baselines.
>
> 3. In terms of **test time**, the time overhead for Agent2World-Multi is comparable or even lower than that of other methods.
>
> We hope this explanation clears up the confusion regarding token consumption and highlights how our design optimizes both efficiency and performance. Let us know if you have any further questions.

---

> ### Author Response · Authors · 2025-11-26
> **Response to Reviewer J92f (4)**
>
> ### W3. Unclear experimental settings: It is not specified which LLM is used in each comparison experiment or in each stage of Agent2World. Even if the code-generation LLM is held constant, could stronger LLMs in the deep researcher or unit-test modules (e.g., better at PDDL understanding) be driving the gains due to unbalanced LLM usage?
>
>
> **A:** Thanks for the opportunity for clarification:
> 1. As described in the **Section 4.2** of initial manuscript, all Agent2World modules (Deep Researcher, Model Developer, Simulation Tester, Unit Tester) and all re-implemented baselines use the same GPT-4.1-mini model as backbone, with identical decoding settings (temperature = 0, top_p = 1, i.e., greedy decoding).
>
> 2. We do not use any stronger LLM in the Deep Researcher or unit-test components; therefore the observed gains come from the agentic architecture rather than unbalanced model strength across modules.
>
> ---
>
> ### W4. Limited comparison methods: For example, in Text2World, the authors compare only with direct search, a single-agent baseline, and the original Text2World method. Many more test-time scaling techniques (e.g., MCTS, World Coder, advanced prompt engineering) should be included.
>
> **A:** Thank you for this suggestion. In the revised manuscript, we strengthen the comparison by adding both generic test-time scaling baselines and a stronger agentic baseline.
>
> 1. **Test-time scaling baselines**:
>
>     **a.** **Best-of-N** [1] is widely used to improve LLM reasoning and was recently introduced by Yu et al. [2] for PDDL generation. In our experiments, we further generalize Best-of-N so that it can be applied to **both** PDDL generation and code-based world-model generation.
>     **b.** **Self-consistency** [3]: This method performs multi-sample reasoning followed by a voting step, and is commonly regarded as a particularly effective form of test-time scaling for LLM reasoning.
>
> 2. **Enhanced agent baseline** (GIF-MCTS + Deep Research): GIF-MCTS is a recent method that uses Monte Carlo Tree Search to guide the construction of code world models [4]. In our enhanced variant, we keep the original GIF-MCTS workflow unchanged, but precede it with our Deep Researcher, which retrieves web evidence and summarizes it into an enriched research report that is provided as additional context to GIF-MCTS.
>
>
> Due to the limited time during the rebuttal period, we have implemented and reported these methods on CWMB, and we will extend these test-time scaling baselines to our other benchmarks (e.g., Text2World, ByteSized32) in the camera ready version.
>
> | Method            | Overall Accuracy ↑ | Overall 𝓡 ↑ |
> | ----------------- | ------------------ | ------------ |
> | Direct Generation | 0.4466             | 0.2620       |
> | Best-of-N         | 0.4257             | 0.3317       |
> | Self-consistency  | 0.4276             | 0.3076       |
> | GIF-MCTS + DR         | 0.4979             | 0.3649       |
> | **Ours (Multi)**  | **0.5441**         | **0.4811**   |
>
> As is shown in this table all enhanced baselines outperform direct generation but still underperform our proposed method. The table below summarizes the **overall** CWMB metrics and the detailed results are presented in **Table 2** in the revised manuscript.
>
> ---
>
> **References**
>
> [1] Stiennon, N., et al. *Learning to summarize with human feedback.* NeurIPS, 2020.
> [2] Yu, Z., et al. *Generating symbolic world models via test-time scaling of large language models.* arXiv:2502.04728, 2025.
> [3] Wang, X., et al. *Self-consistency improves chain of thought reasoning in language models.* arXiv:2203.11171, 2022.
> [4] Dainese, Nicola, et al. "Generating code world models with large language models guided by monte carlo tree search." Advances in Neural Information Processing Systems 37 (2024): 60429-60474.

---

> ### Author Response · Authors · 2025-11-26
> **Response to Reviewer J92f (5)**
>
> ### W5. Beyond zero-shot inference: Although the paper is not about learning, could training-based methods enhance WM synthesis? I ask this because some LLMs may have seen PDDL data from GitHub during pretraining, giving them an inherent advantage.
>
> **A:** We agree that learning-based methods can further enhance world-model synthesis. One of our main goals is to show that formalizing world-model generation as an interaction between a “developer” and a “testing team” provides a clean interface for learning from test feedback, rather than only relying on zero-shot inference.
>
> To make this concrete, we add a training-based variant of Agent2World on CWMB (**Section 5.6**). Specifically, we recast the interaction between the Model Developer and the Testing Team as an MDP, where the state encodes the task specification and testing diagnostics, the action is a proposed world-model implementation or patch, and the transition and reward are generated by the Testing Team (simulation + unit tests) on the candidate world model. This turns the Testing Team into an automatic reward generator and turns our evaluation pipeline into an agent-in-the-loop training environment for world-model agents.
>
> On top of this environment, we instantiate a simple training scheme: we collect trajectories on 500 tasks spanning from PDDL-based and code-based world models, filter them using the Testing Team’s scalar reward, and use the remaining traces as demonstrations to fine-tune the same llama-3.1-8b-instruct backbone (Section 5.6, Table 5). The fine-tuned developer achieves notable gains in both accuracy and normalized return over the raw backbone (e.g., overall normalized return improves from 0.2296 to 0.3156).
>
>
> | Method        | Overall Accuracy ↑ | Overall 𝓡 ↑ |
> |--------------|--------------------|-------------|
> | Agent2World-Multi        | 0.3150             | 0.2296      |
> | Agent2World-Multi (SFT)  | 0.3360             | 0.3156      |
> | Δ (absolute) | +0.0210            | +0.0860     |
>
>
> Due to the limited time and compute budget during the rebuttal period, we currently perform SFT only on this 500-task subset; in the camera-ready version, we plan to further explore RL-style training, larger-scale data, and extensions to additional benchmarks.

---

> ### Comment · Reviewer_J92f · 2025-11-28
>
> Dear Authors,
>
> Thank you very much for the rebuttal. Some of my concerns still remain:
> In the contamination test, you used 10-gram exact match. This test is very strict, even if the correct solution appears directly on the retrieved webpage, a single extra space, different variable name, different indentation, minor refactoring, or extra comment will immediately break any 10-token exact sequence. That's why I find the test setting odd. It's great to see you restate your novelty. I agree with most of your points, although from the perspective of general CoT and agentic reasoning methods, the novelty is still limited. However, I believe people from the symbolic reasoning applications community may find this work interesting.
>
> That said, some of my concerns have been addressed. I notice the token efficiency comparison experiment is sound, and I'm glad you added more test-time scaling baselines in the rebuttal. It's also appealing that you compared the effect of SFT on Llama and found that planning capabilities were enhanced after learning. I found that the method proposed in the paper works for both GPT-4-mini and Llama 3.1, which demonstrates the pipeline's stability across different foundation models.
>
> Given the above comments, I choose to raise my soundness score from 2 to 3, I think I couldn't edit the score right now, as the edit button is missing in the review system, but will definitely when I am able to do so. If the AC reviews my response here, I want to state that I don't mind this paper being accepted, as it may be useful to the symbolic reasoning applications community. However, I choose to keep the original score, the AC can regard this paper as worth a score from 4.5 to a marginal of 5.
>
> To the authors, I wish you good luck with your submission.
>
> Best,
>
> Reviewer J92f

---

### Official Review · Reviewer_m6C3 · 2025-10-31

**Soundness:** 2
**Presentation:** 3
**Contribution:** 2
**Rating:** 4
**Confidence:** 3

**Summary:**

This work proposes Agent2World, a novel paradigm that uses tool-augmented LLM agents to generate symbolic world models. It has 3 components, a Deep Researcher which uses the internet to collect information and specifications, a Model Developer which creates and implements the world model, and a Testing Team, which uses systematic unit testing and simulations to validate and refine the world model. This work evaluates this paradigm on 3 benchmarks spanning PDDL and executable code representations and achieves good performance.

**Strengths:**

Method is simple and easy to understand and well presented. Figures of Agent2World clearly show how the system works as well as how this method differs from prior methods. Authors conduct experiments on 3 diverse benchmarks and compare against multiple baselines to show on-par or superior performance. Ablation studies run by the author ablate each of the 3 components of Agent2World while also analyzing how the feedback affects the performance as well as how the multi-agent architecture affects it.

**Weaknesses:**

The novelty of the method may be overstated. As stated in Section 4.1 Baselines, two prior work: WorldCoder and GIF-MCTS both have an LLM that creates a worlds model and another LLM which refines the world model using code and/or unit testing. In the introduction, in the second paragraph as well as in Figure 1, the authors claim that existing methods use "scripted workflows" which couple generation with verification and repair. However, it seems like WorldCoder and GIF-MCTS are very similar to Agent2World, except that they do not have the searcher agent to do RAG/Research on the internet, and that they are "Adaptive Agents" instead of "Static Workflows" (Table 5).

Looking closer at Table 2 and Table 8, it looks like WorldCoder performs around the same as Agent2World with no deep researcher, which seems to indicate that the main contribution to performance is mostly the deep researcher compared to prior work. It is unclear whether this claim that prior work have "Passive and rigid execution" affects performance significantly. Furthermore, it seems like this is solely a cost saving metric: "leading to unnecessary computations when simpler solutions exist or inadequate exploration when complex problems require adaptive strategies" (lines 50-51) rather than for performance.

In addition, since these are public benchmarks, the fact that they allow the model to search the internet seems to be a potential point of test-set leakage. The paper only states that they have "blocked some websites" (lines 231-232, line 269), but it is unclear if any other websites or results may have leaked the test set.

**Questions:**

Can the authors show that adding a Deep Research agent to do research online to these baselines like WorldCoder or GIF-MCTS, or just a Deep Research Agent + some "static workflows" performs worse than Agent2World? This would be great to show that being an "adaptive agent" is important for performance.

What is the cost associated with running these experiments, in terms of token counts and also latency? One of the stated improvements of the model is efficiency, but it does not seem to be any results related to this.

It is unclear how Agent2World uses a "unified cross-representation framework" (line 081-082), and how this differs from prior work.

I am very concerned about leakage of the test set on the internet, can the authors do some kind of analysis on the links/results that the deep research agent went to to make sure that there was no egregious leakage here?

---

> ### Author Response · Authors · 2025-11-26
> **Response to Reviewer m6C3 (1)**
>
> Dear Reviewer m6C3,
>
> Thank you for your thoughtful review and constructive feedback! We appreciate your recognition of the clarity of our method and the experiments we conducted, especially the comprehensive comparisons to baseline methods.
>
> We have revised our manuscript to address the points you raised, including the following clarifications:
>
> * Clarification of Novelty (W1).
>
> * Experiments on a new baseline (GIF-MCTS*) where we integrate the Deep Researcher into the GIF-MCTS pipeline (W2 & Q1).
>
> * Token/Time cost experiments and analysis (W3 & Q2)
>
> * We conducted a thorough analysis of the Deep Researcher's web search behavior to ensure no leakage from the benchmark test sets. (W4 & Q4)
>
> * Clarification on "unified framework" (Q3)
>
> Please let us know if you have any further questions or concerns. We look forward to your feedback on the revised manuscript!

---

> > ### Author Response · Authors · 2025-11-26
> > **Response to Reviewer m6C3 (2)**
> >
> > ### W1. Concern that the novelty is overstated, and that WorldCoder / GIF-MCTS are very similar to Agent2World
> >
> > **A:** Thank you for this careful comment. We agree that WorldCoder and GIF-MCTS are among the closest prior systems to Agent2World: they also combine (i) an LLM that proposes a world model and (ii) refines it based on code execution. Our goal is not to claim novelty for this high-level “generator + refiner” pattern, but rather to clarify what is different about our **agent-in-the-loop multi-agent framework for symbolic world-model generation**. We also agree that our current wording (“scripted workflows”) can be confusing, and we will revise it to more accurately reflect the relation to WorldCoder and GIF-MCTS.
> >
> > Concretely, there are three aspects where we see a difference between workflow-based methods (including WorldCoder / GIF-MCTS) and Agent2World:
> >
> > 1. **Pipeline-level control: fixed workflows vs. agent-controlled tool sequences.** When we describe prior methods as using “scripted workflows”, we refer to the **pipeline-level control logic**:
> >
> >    * the *stage structure* is pre-defined (e.g., “draft → run a fixed test suite → feed back results → patch → repeat for K iterations”), and
> >    * every task follows this same macro-level pattern, even if the intermediate failures differ.
> >
> >    In systems like WorldCoder and GIF-MCTS, test execution is typically tied to such stages: after each draft, the system runs (a large part of) a predefined test harness, obtains scores, and then revises. The tests themselves can certainly be interpreted as rewards or scores, but **the way they are invoked is governed by a fixed workflow** rather than by the agent deciding which specific tool to call next.
> >
> >    In Agent2World, we instead expose **individual tools as atomic actions** to the agents in an agentic loop. The available tools are specified (e.g., simulation, unit tests, play traces, web search), but **their order, frequency, and combinations are decided within the agent’s reasoning**, not pre-scripted at the pipeline level.
> >
> > 2. **Adaptive agentic refinement (Testing Team), beyond fixed test stages.**
> >    Prior world-model systems already use code execution and tests as a reward signal to guide refinement, but the test harness itself is typically *static*: a benchmark designer specifies a fixed suite of unit tests or trajectory metrics, and the system runs (most of) this suite in the same way after each revision step.
> >    In Agent2World, the Testing Team is itself an **agentic component**: (i) it can dynamically synthesize new, targeted unit tests conditioned on the current failure modes or missing behaviors, instead of relying solely on a pre-defined test set; and (ii) it includes play agents that interact with the current world model to generate fresh trajectories and counterexamples, which are then turned into diagnostics and additional tests.
> >    This makes the refinement loop adaptive on both sides: the Model Developer updates the world model, and the Testing Team actively adjusts how it probes the model, rather than repeatedly executing a fixed test harness. This adaptive Testing Team is what underlies our agent-in-the-loop view and supports the training setup introduced next.
> >
> > 3. **A unified interface across PDDL and executable code that enables training, plus a training-based variant.**
> >    Third, we design a **single interaction interface**—roles, message structures, and diagnostics schema—that is reused across both **PDDL-style** and **executable-code** world models (Text2World, CWMB, ByteSized32). WorldCoder and GIF-MCTS are tailored to specific representations and evaluation protocols (e.g., a particular PDDL benchmark with a fixed checker, or a specific code environment and trajectory metric). In contrast, in Agent2World:
> >
> >    * the same high-level roles (Deep Researcher → Model Developer → Testing Team / Play Agents) and the same style of Testing-Team feedback are used for both PDDL domains and code-based environments;
> >    * as a result, the resulting interaction traces have a consistent structure and can be treated as demonstrations within a **single agent-in-the-loop MDP** as illustrated in **Section 5.6**.
> >
> >    Building on this, we add in the revision a **training-based variant (SFT)**: we roll out Agent2World-Multi on a 500-task dataset spanning PDDL-based and code-based world models, filter trajectories using Testing-Team feedback, and fine-tune the backbone. The resulting **Agent2World-Multi (SFT)** improves both accuracy and normalized return over the non-SFT backbone, indicating that the proposed interface is not only a way of connecting modules but also a **useful substrate for training world-model agents from interaction data**. To our knowledge, prior world-model systems have not instantiated such a unified, cross-representation interface and used it to build a general training loop for world-model agents.
> >
> > We will soften the statement in the introduction in the camera-ready version.

---

> > > ### Author Response · Authors · 2025-11-26
> > > **Response to Reviewer m6C3 (3)**
> > >
> > > ### W2 & Q1. concerning on the main contribution to performance is mostly the deep researcher compared to prior work & Adding a Deep Research agent to do research online to these baselines like WorldCoder or GIF-MCTS
> > >
> > > **A:** Thank you for pointing this out and for the concrete suggestion. We agree that the Deep Researcher contributes a large part of the final performance gain, and we do not intend to underplay its importance. At the same time, our experiments (including a new baseline following your suggestion) indicate that the **adaptive agentic refinement** in Agent2World-Multi also brings a substantial improvement, beyond what can be achieved by simply adding a Deep Research agent to a static workflow.
> > >
> > > To address your question about “Deep Research + static workflows”, we also implemented the variant you suggested. We take the official GIF-MCTS implementation, keep its **Generate/Improve/Fix** workflow and test harness unchanged, and simply connect our Deep Researcher to its code-generation steps.
> > >
> > > We denote this enhanced static-workflow baseline as GIF-MCTS*. On CWMB with GPT-4.1-mini, we add the detailed results in **Table 2, Line 270** and the overall results are:
> > >
> > > | Method                                | Accuracy ↑ | 𝓡 ↑    |
> > > |---------------------------------------|------------|--------|
> > > | WorldCoder                            | 0.5210     | 0.3197 |
> > > | GIF-MCTS                              | 0.4877     | 0.3488 |
> > > | GIF-MCTS* (GIF-MCTS + Deep Researcher) | 0.4979     | 0.3649 |
> > > | Agent2World-Multi           | 0.5441     | 0.4811 |
> > >
> > > From this we see:
> > >
> > > * Adding a Deep Research agent to a **static workflow** does help: GIF-MCTS* improves over GIF-MCTS (especially in $\mathcal{R}$: 0.3488 → 0.3649), which confirms that Deep Research is broadly useful and not specific to our pipeline.
> > > * However, **even when both systems have a Deep Researcher**, Agent2World-Multi still significantly outperforms GIF-MCTS*: +0.0462 in accuracy (0.5441 vs. 0.4979) and +0.1162 in normalized return (0.4811 vs. 0.3649).
> > >
> > > This gap remains even though GIF-MCTS* uses the same backbone and has access to the same web-augmented information. The main structural difference is that GIF-MCTS* still follows a **fixed stage workflow** (running a predetermined test harness at each revision), whereas in Agent2World-Multi the tools are exposed as atomic actions in an agentic loop and the Testing Team can adaptively generate new unit tests and play-based counterexamples based on the current failures.
> > >
> > > In other words, **Deep Researcher and adaptive refinement are complementary**: the Deep Researcher improves the quality of world-model specification, while the adaptive Testing Team + agentic control lead to more effective use of tests and play to refine the generated world model.

---

> > > > ### Author Response · Authors · 2025-11-26
> > > > **Response to Reviewer m6C3 (4)**
> > > >
> > > > ### W3 & Q2. Efficiency and missing cost results
> > > >
> > > > **A:** Thank you for raising this important concern.
> > > >
> > > > In Section 5.5 of the initial manuscript, we already examined the **specification efficiency** of Agent2World-Multi, the advantage of multi-agent architecture in terms of token cost.
> > > >
> > > > Additionally, in the revised manuscript, we provide a more detailed analysis of **token consumption** and **runtime overhead** (see Appendix J, Line 2479).
> > > >
> > > > Specifically:
> > > >
> > > > 1. The **test time** of Agent2World-Multi is comparable to that of other workflow-based methods. This is primarily because our approach uses fewer refinement iterations (e.g., while GIF-MCTS requires 10 turns, our method uses only 3). Furthermore, the adaptive nature of Agent2World-Multi allows it to employ **early stopping** during the testing phase, thus optimizing runtime further.
> > > >
> > > > 2. Regarding **token consumption**, the **model developer** in Agent2World-Multi uses fewer tokens compared to other approaches. However, it’s important to note that the testing phase in Agent2World-Multi is **proactive**—each turn involves generating **targeted test cases** and **player agent trajectories**, whereas other methods tend to rely on **static test cases**. This proactive strategy, while beneficial for the adaptive nature of the system, does result in higher token consumption during testing. Despite this, Agent2World-Multi remains **competitive in efficiency** relative to other methods, maintaining a strong balance between token usage and performance.
> > > >
> > > > We hope these explanations address your concerns about the cost and runtime. Agent2World-Multi aims not only to improve **efficiency** but also **adaptivity** in complex tasks, achieving **better performance** by dynamically adjusting strategies based on intermediate feedback. We welcome any further questions you may have.
> > > >
> > > > ---
> > > >
> > > > ### W4 & Q4. Concern about test-set leakage via web search
> > > >
> > > > **A:** We take this concern very seriously. In the current paper, we mention that we block some websites, but we agree this description is not sufficiently detailed.
> > > >
> > > > In practice, we already employ the following safeguards:
> > > >
> > > > 1. A denylist of domains that include benchmark names, official repositories, and associated GitHub / HuggingFace pages, to avoid directly retrieving benchmark code or solutions.
> > > >
> > > > 2. Heuristics to filter out search results whose titles/snippets contain benchmark names or explicit references to test sets.
> > > >
> > > > Furthermore, to address your concern, we conducted a data-leakage analysis. Specifically, we computed the 10-gram overlap between the Deep Researcher’s retrieval logs and the gold world models, and found zero contaminated instances. We have included this experiment and the detailed setup in Appendix I (Line 2410).
> > > >
> > > > ---
> > > >
> > > > ### Q3. It is unclear how Agent2World uses a "unified cross-representation framework" (line 081-082), and how this differs from prior work.
> > > >
> > > > **A:** Thank you for your question. Let me clarify how Agent2World uses a **unified cross-representation framework**, and how this differs from prior work.
> > > >
> > > > 1. **Unified Intermediate Representation (IR)**
> > > >    Agent2World uses a **shared intermediate representation (IR)** for communication between agents, regardless of whether the world model is symbolic (PDDL) or executable code. The core interaction protocol (e.g., Model Developer proposing changes, Testing Team providing feedback) is **representation-agnostic**. The only **representation-specific parts** are handled by thin adapters:
> > > >
> > > >    * The **PDDL validator** checks symbolic constraints,
> > > >    * The **code executor** runs executable models.
> > > >      These adapters ensure that the core framework remains flexible across different types of world models without altering the agent interaction logic.
> > > >
> > > > 2. **Flexible Developer and Tester Roles**
> > > >    The **roles of Model Developer and Testing Team** are decoupled from specific representations. This means that the same agents can be used for both symbolic (PDDL) and executable code-based models, with minimal changes. The key difference with prior work is that we don’t need to design separate pipelines for each representation, making Agent2World **cross-representational** and adaptable to new benchmarks without re-engineering.
> > > >
> > > > 3. **Generic Framework for Symbolic World Model Generation**
> > > >    The goal of Agent2World is to provide a **generic framework** for symbolic world-model generation that works across different representations. While the framework is flexible, it still achieves **state-of-the-art performance** across various benchmarks (Text2World, CWMB, ByteSized32), outperforming prior work. This demonstrates that the **cross-representation framework** is not only flexible but also **empirically effective**.

---

> > > > > ### Comment · Reviewer_m6C3 · 2025-11-26
> > > > >
> > > > > I thank the authors for their response.
> > > > >
> > > > > In light of the additional information on their adaptive framework, as well as the additional experiments regarding the deep researcher + GIF-MCTS, I've raised my soundness and contribution scores accordingly.
> > > > >
> > > > > 1. On the question of ablations, I see that the gain comes not just from the deep researcher, but also from the adaptive architecture of the agent itself.  Would it be possible to then run a simple ablation of Agent2World WITHOUT the deep researcher? I believe that this would allow us to quantify exactly how much performance comes from the deep research part, and which comes from the adaptive part.
> > > > >
> > > > > 2. On the question regarding the efficiency, I do think that Figure 7 does not really paint the picture that the method is really improving efficiency, as the efficiency can go as low as single digits in percentage. Furthermore, the tables in Appendix J are very hard to read, which rows correspond to Agent2World-single and which rows correspond to Agent2World-multi?
> > > > > I think with the current results, it looks more like you pay upfront more tokens for testing in exchange for some moderate improvement in performance during test time, with a similar deployment cost. If so, then I would ask the authors to tone down their efficiency claims in the manuscript, as this isn't really a gain in efficiency.
> > > > >
> > > > > 3. On the topic of the new experiments with SFT, it is a bit concerning that this method requires finetuning in order to beat GIF-MCTS; does this mean that for sufficiently small backbones, the method does not seem to work out of the box, and you need SFT in order to see improvements over the more static workflows?
> > > > >
> > > > > 4. There also seem to be some glitches in the new writing, like "we the decoding temperature" on line 233.
> > > > >
> > > > > If the authors are able to run that ablation and/or sufficiently address comments 2 and 3 above, I would be willing to raise my overall score and confidence.

---

> ### Author Response · Authors · 2025-11-27
> **Response to Follow-up Questions of Reviewer m6C3**
>
> We sincerely thank the reviewer for the acknowledgment of our rebuttal and the additional experiments, as well as for raising the scores. We appreciate the opportunity to address these follow-up questions to further clarify the contributions of our framework.
>
> ---
>
> ### Q1. Ablation of Agent2World WITHOUT the Deep Researcher
>
> > **"Would it be possible to then run a simple ablation of Agent2World WITHOUT the deep researcher? I believe that this would allow us to quantify exactly how much performance comes from the deep research part, and which comes from the adaptive part."**
>
> Thank you for this insightful suggestion regarding the attribution of performance gains.
>
> We would like to clarify that this specific ablation was indeed included in our original manuscrip (referenced as "No Deep Researcher" in Section 5.1 and Table 9 of Appendix D).
> This variant represents our adaptive framework (Model Developer + Testing Team) operating solely with the LLM's internal knowledge, without access to the Deep Researcher's external knowledge synthesis.
>
> To directly address your request for quantification, we have extracted the relevant comparisons below:
>
> | Method | Knowledge Source | Overall Accuracy ($\uparrow$) | Overall Normalized Return ($\mathcal{R}$) ($\uparrow$) |
> | :--- | :--- | :---: | :---: |
> | GIF-MCTS (w/o DR)  | Internal Only | 0.4877 | 0.3488 |
> | GIF-MCTS (with DR) | + Deep Research | 0.4979 | 0.3649 |
> | Agent2World-Multi (w/o DR) | Internal Only | 0.5201 | 0.2936 |
> | Agent2World-Multi (with DR) | + Deep Research | **0.5441** | **0.4811** |
>
> Based on these results originally reported in the paper (Table 2 and Table 9):
>
> 1.  **Adaptive Architecture Drives Accuracy:** Even without external knowledge, the adaptive agent alone (Agent2World w/o DR) achieves an Overall Accuracy of 0.5201, which surpasses both the static GIF-MCTS baseline (0.4877) and its search-augmented variant (0.4979). This indicates that the evaluation-driven refinement loop effectively improves the *predictive correctness* of the world model code.
>
> 2.  **Deep Research Drives Utility:** Comparing the rows with and without Deep Research reveals that external knowledge is the primary driver for downstream utility (Normalized Return $\mathcal{R}$).
>
> 3.  **Synergy of Architecture and Knowledge:** Notably, simply adding Deep Research to the static workflow (GIF-MCTS*) only yields a modest improvement. However, when Deep Research is integrated into our adaptive framework, the return jumps significantly to 0.4811.
>
> ---
>
> ### Q2. Efficiency and Presentation
>
> > **Q2.1 The tables in Appendix J are very hard to read, which rows correspond to Agent2World-single and which rows correspond to Agent2World-multi?**
>
> We apologize for the confusion caused by the formatting in Appendix J. We have revised these tables in the revised manuscript to clearly group rows using separators to ensure distinct readability.
>
> > **Q2.2 I would ask the authors to tone down their efficiency claims in the manuscript, as this isn't really a gain in efficiency.**
>
> We agree with your assessment. Our method represents a trade-off: investing more tokens upfront during proactive testing and deep research to secure higher reliability. To reflect this, we have revised **Section 5.5** in the manuscript to tone down claims regarding raw efficiency and explicitly state the trade-off.
>
> ---
>
> ### Q3. SFT and Backbone Size
>
> > **On the topic of the new experiments with SFT, it is a bit concerning that this method requires finetuning in order to beat GIF-MCTS...**
>
> We would like to clarify that **Supervised Fine-Tuning (SFT) is NOT required** for Agent2World to outperform GIF-MCTS, even on smaller backbones.
>
> As shown in **Table 2** (specifically the **Llama-3.1-8b-instruct** section), our method works effectively out-of-the-box:
>
> | Method | Backbone Model | Overall Accuracy ($\uparrow$) | Overall Normalized Return ($\mathcal{R}$) ($\uparrow$) |
> | :--- | :--- | :---: | :---: |
> | **GIF-MCTS** | Llama-3.1-8b | 0.2883 | 0.2070 |
> | **Agent2World-Multi (No SFT)** | Llama-3.1-8b | **0.3150** | **0.2296** |
>
> Our untrained multi-agent framework already outperforms the baseline. The purpose of the SFT experiment (which further raises normalized reward to **0.3156**) was to demonstrate the "Data Flywheel" potential of our system, showing that the high-quality data generated by the Testing Team can be used to improve the performance of the backbone model, further widening the gap.
>
> ---
>
>
> ### Q4. Glitches in Writing
>
> > **There also seem to be some glitches in the new writing, like "we the decoding temperature" on line 233.**
>
> Thank you for your meticulous reading. We have corrected the typo on line 233 (changing it to "We set the decoding temperature...") and have performed a thorough proofreading of the added text to ensure all grammatical errors are resolved.
>
>
> ---
>
> We hope these responses sufficiently address your remaining concerns. Please let us know if you have further questions.

---

> > ### Comment · Reviewer_m6C3 · 2025-11-27
> >
> > I thank the authors for their prompt response.
> >
> > Those ablations feel fairly imperative for the main body of the paper, since it sheds light on the components, and I would ask that the authors include the table in the main work, perhaps as part of Table 1, or right underneath it.
> >
> > Regarding the SFT performance, I was referring to the Discrete accuracy, but I see that Agent2World has sufficient gains in the Continuous action space to make up for it overall.
> >
> > With regards to the efficiency, I think that having the table with the efficiency in Table 7, I think that this graph then is pretty misleading, it doesn't really say anything. This specialization efficiency number is not very insightful, I would personally just remove it. I might rephrase it that you can compute the additional cost of Agent2World, and to phrase it like it only costs X in order to get Y improvement in performance overall, which feels more accurate. The execution of the world model look about on par with the baselines, so this is a finite upfront cost, so I think that this is a more accurate way to represent the cost.
> >
> > Overall, I am satisfied with the response by the authors and have raised my score and confidence.

---

> > > ### Author Response · Authors · 2025-11-28
> > > **Response to Follow-up Suggestions of Reviewer m6C3**
> > >
> > > We sincerely thank the reviewer for the prompt response and for raising the score. We have updated the manuscript to reflect your insightful suggestions regarding the presentation of ablations and the framing of efficiency.
> > >
> > > **1. Moving Ablations to Main Body**
> > >
> > > > **"Those ablations feel fairly imperative for the main body of the paper... I would ask that the authors include the table in the main work..."**
> > >
> > > We fully agree that the component analysis is critical for understanding the source of our performance gains. We have integrated the full ablation results (previously Table 9 in Appendix D) into Table 2 of the main paper to ensure high visibility. We will further refine the table's formatting in the camera-ready version to maximize readability.
> > >
> > > **2. SFT Performance and Variance**
> > >
> > > > **"...I see that Agent2World has sufficient gains in the Continuous action space to make up for it overall."**
> > >
> > > Thank you for this confirmation. We appreciate your recognition that the significant gains in Continuous action spaces effectively complement the performance in Discrete settings. We would also like to note that the dataset size for Discrete environments is relatively small, which contributes to higher variance in those specific metrics. Despite this fluctuation, the robust performance in Continuous settings secures the overall state-of-the-art results.
> > >
> > > **3. Re-framing Efficiency as Cost-Benefit**
> > >
> > > > **"This specialization efficiency number is not very insightful... phrase it like it only costs X in order to get Y improvement in performance overall..."**
> > >
> > > We accept this constructive suggestion. We agree that the "specialization efficiency" ratio obscured the actual engineering trade-off.
> > > We have rewrote Section 5.5 and restructured the narrative to emphasize "Finite Upfront Investment vs. Permanent Performance Gain." We now explicitly state that while our method incurs a finite additional cost during the generation phase (to verify correctness), it secures significant, permanent performance improvements during downstream deployment.

---

### Official Review · Reviewer_aueF · 2025-10-31

**Soundness:** 2
**Presentation:** 2
**Contribution:** 2
**Rating:** 4
**Confidence:** 3

**Summary:**

The paper presents agent2world, a LLM-based framework for automatically generating symbolic world models (in both PDDL and executable code forms) from natural language descriptions. To address the limitations of prior approaches, the proposed framework has three components:

1. A web searcher that retrieves missing or external knowledge via web search;
2. A model developer that generates executable world models (e.g., python code);
3. A testing team, which generates feedback by simulation and unit tests to refine the model's outputs.

Evaluation is done on three benchmarks — Text2World, CWMB, and ByteSized32. The proposed framework achieves good results, and ablation and error analyses further show the complementary benefits of knowledge synthesis and iterative refinement.

**Strengths:**

1. Clear modular design: The three-stage architecture (knowledge synthesis, model generation, evaluation-driven refinement) is conceptually clean and practically effective.
2. Strong performance and comprehensive evaluation: The proposed framework demonstrated better results across three datasets. The authors also provided a deep analysis, including an ablation study, distribution of errors, case study, etc.

**Weaknesses:**

1. Limited novelty in methodology: While the framework design is reasonable and overcomes previous limitations, each component, e.g., web search, sandbox testing, builds on commonly used agentic LLM paradigms. The contribution is incremental and more like engineering integration.
2. Scalability and cost: The approach relies heavily on multiple LLM calls and web queries; runtime and token cost are not deeply analyzed, raising questions about scalability to large domains.
3. Limited base models: The paper only experimented with one base LLM -- GPT 4.1 mini, which is one of the state-of-the-art LLMs. It's unclear whether the complicated framework can also improve less capable LLMs.

**Questions:**

Figure 6 presents the error distribution of the proposed framework. What are the error distributions of baseline approaches? What kind of errors does the proposed framework reduce or increase? This fine-grained comparsion can reveal the strenghs and weaknesses of the proposed method but is missing.

---

> ### Author Response · Authors · 2025-11-26
> **Response to Reviewer aueF (1)**
>
> Dear Reviewer aueF,
>
> Thank you for your thoughtful review and positive feedback on our work! We appreciate your recognition of our modular design and strong performance across multiple benchmarks.
>
> In response to your constructive feedback, we have made several revisions and clarifications in the manuscript, including:
>
> * Clarifications on methodological novelty. (W1.)
>
>
> * A more detailed analysis of the token costs and runtime. (W2, Appendix J, Line 2479)
>
> * Additional experiments using a smaller, open-source Llama-3.1-8B-Instruct model (W3. Table 2)
>
> * Error analysis of baseline methods and comparison (Q1, Appendix E.3 Line 983)
>
>
> We hope that these revisions address your concerns and further clarify the contributions of our work. We look forward to your continued feedback and are happy to provide further clarifications as needed.
>
> ---

---

> ### Author Response · Authors · 2025-11-26
> **Response to Reviewer aueF (2)**
>
> ### W1. Limited methodological novelty; incremental engineering integration
>
>
> **A:** Thank you for raising this concern. We agree that each *individual* component in Agent2World (ReAct-style backbone, web search, sandbox testing, unit tests) builds on commonly used agentic LLM paradigms, and we do **not** claim novelty at the level of these primitives. Our contribution lies instead in the **proposed multi-agent architecture and agent-in-the-loop formulation** tailored to **symbolic world-model generation**, rather than in introducing new low-level tools.
>
> Concretely:
>
> 1. **We do not claim the ReAct-style backbone as our main novelty.** The ReAct-style, tool-using backbone itself is not our primary contribution. We fully acknowledge that this and the individual tools are based on prior work. In fact, we explicitly include a **ReAct-style single-agent baseline (Agent2World-Single)** that uses the *same* backbone and tools. This baseline is meant to represent the “straightforward engineering integration” of a generic agent with web search + sandbox testing. Despite this, it still underperforms our (Agent2World-Multi), suggesting that the performance gains come from the **specific multi-agent roles and interaction protocol we design for world-model generation**, not merely from wiring existing tools together.
>
> 2. **From scripted workflows to proactive agents tailored to world-model generation.** Prior world-model systems are typically based on **scripted draft–repair workflows**: the control flow (“draft → check → patch → repeat”) is fixed by the designer and does not adapt to intermediate feedback. In contrast, Agent2World uses **proactive agents** that decide *when and how* to call tools (web search, simulation, unit tests, play) depending on the current state of the symbolic world model. This shift from hand-designed scripts to adaptive decisions is crucial for discovering missing constraints, dynamics bugs, and commonsense violations, and it goes beyond simply combining web search and sandbox testing in a fixed pipeline.
>
> 3. **A unified, cross-representation framework rather than representation-specific pipelines.** Agent2World is designed as a **unified interface** that works for both PDDL-style and executable-code world models. The same high-level roles (Deep Researcher → Model Developer → Testing Team / Play Agents) and message structure are reused across Text2World, CWMB, and ByteSized32 tasks. This contrasts with existing systems that are tightly coupled to a particular representation (e.g., PDDL-only pipelines or code-only refinement workflows). This cross-representation design is important for two reasons:
>
>    * It shows that the *same* multi-agent framework can handle different symbolic representations without redesigning the pipeline each time;
>    * It provides a **clean, reusable interaction interface** whose traces can be directly leveraged for training world-model agents (see point 4).
>
> 4. **Formulating world-model generation as an agentic MDP and adding a training-based variant.** Beyond the architectural design, we **formalize** symbolic world-model generation as an **agent-in-the-loop MDP**: the Model Developer is a policy, the state encodes the task specification and Testing-Team diagnostics, actions are world-model implementations or patches, and transitions/rewards are produced by simulation and unit tests. This gives a principled way for an LLM agent to *use* simulation testers and play traces to iteratively improve a symbolic world model, instead of treating tools as one-shot or purely heuristic checkers.
> 5. Building on this, in the revised version, we also include a **training-based variant (SFT)** (Section 5.6, Line 446): we roll out Agent2World-Multi, filter trajectories using Testing-Team feedback, and fine-tune the backbone on these interaction traces. This **learning-based Agent2World-Multi (SFT)** improves both accuracy and normalized return over non-SFT baselines, reinforcing that the proposed framework is not just a static engineering integration, but also a **useful learning substrate** for world-model agents.
>
> In summary, while we fully agree that our system stands on well-established agentic components, we view the contribution as more than incremental engineering:
>
> (i) we move from scripted draft–repair workflows to proactive, tool-using agents specifically tailored to symbolic world-model generation,
>
> (ii) we provide a unified cross-representation framework that works for both PDDL and executable code, and
>
> (iii) we introduce an agent-in-the-loop MDP formulation plus a training-based variant that demonstrates how this framework naturally supports learning from Testing-Team feedback.

---

> > ### Author Response · Authors · 2025-11-26
> > **Response to Reviewer aueF (3)**
> >
> > ### W2. Scalability and cost (token/runtime overhead)
> >
> > **A:** Thank you for raising this important concern.
> > In Section 5.5 of the initial manuscript, we already investigated the **specification efficiency** of AGENT2WORLD-Multi, and emphasize that the multi-agent architecture plays a crucial role in the scalability of the approach by introducing specialized agents for each task.
> > Additionally, we conducted a more detailed analysis of **token costs and runtime** in the revised manuscript (see Appendix J, Line 2479).
> > Specifically:
> >
> > 1. The **test time** of Agent2World-Multi is comparable to that of other workflow-based methods. This is largely due to our approach using fewer turns in the refinement stage (e.g., GIF-MCTS uses 10 turns, while ours uses only 3). Moreover, the adaptive agents in Agent2World-Multi could apply early stopping during the testing phase, further optimizing runtime.
> >
> > 2. In terms of **token consumption**, the model developer of Agent2World-Multi consumes fewer tokens compared to other methods. However, it is important to note that the testing stage in Agent2World-Multi is **proactive**—at each turn, we generate **targeted testing cases** and **player agent trajectories**, whereas other methods tend to rely on **static test cases**. This proactive nature leads to a higher token consumption during the testing phase. Despite this, Agent2World-Multi still maintains competitive efficiency relative to existing methods.
> >
> > We hope these explanations address your concerns, and we welcome any further questions.
> >
> >
> > ### W3. Limited base models (only GPT-4.1-mini)
> >
> > **A:** We agree that evaluating only on GPT-4.1-mini in the original submission limited our ability to make claims about generality, especially for less capable backbones. In the initial version, we deliberately fixed **all** methods (baselines and Agent2World) to GPT-4.1-mini with identical decoding settings to isolate the effect of the **agentic framework** from raw model quality.
> >
> > To address your concern, we have added a second set of experiments on **CWMB** using a **smaller open-source backbone**, **Llama-3.1-8B-Instruct**, and updated **Table 2** in the revised manuscript accordingly (Line 280). This backbone is substantially smaller and less capable than GPT-4.1-mini, so it provides a concrete test of whether our “complicated framework” still helps weaker LLMs.
> >
> > The key observations are:
> >
> > * Replacing GPT-4.1-mini with Llama-3.1-8B-Instruct **reduces the absolute performance of all methods**, as expected from a smaller model.
> > * Crucially, the **relative ranking is preserved**: Agent2World-Multi (and especially its SFT variant) still outperforms WorldCoder and GIF-MCTS on **both overall accuracy and normalized return**.
> > * Under Llama-3.1-8B-Instruct, our SFT variant Agent2World-Multi(SFT) improves overall normalized return **from 0.2070 (GIF-MCTS) to 0.3156** and overall accuracy **from 0.2883 to 0.3360** on CWMB, indicating that the framework continues to provide substantial gains even for a weaker base model.
> >
> > We summarize the most relevant part of the new table below (overall metrics on CWMB; full results are reported in Section 4.4, Table 2):
> >
> > | Backbone              | Method                               | Overall Accuracy ↑ | Overall 𝓡 ↑ |
> > | --------------------- | ------------------------------------ | ------------------ | ------------ |
> > | GPT-4.1-mini          | Agent2World-Multi          | 0.5441             | 0.4811       |
> > | Llama-3.1-8B-Instruct | WorldCoder                           | 0.2513             | 0.1642       |
> > | Llama-3.1-8B-Instruct | GIF-MCTS                             | 0.2883             | 0.2070       |
> > | Llama-3.1-8B-Instruct | Agent2World-Multi          | 0.3150             | 0.2296       |
> > | Llama-3.1-8B-Instruct | **Agent2World-Multi(SFT)** | **0.3360**         | **0.3156**   |
> >
> > These results support our claim that **Agent2World is not tied to a single proprietary model**: even when we move to a smaller, open-source, and less capable LLM, the **same multi-agent architecture** consistently yields stronger world models than static workflows and prior agentic baselines.
> >
> > Due to the limited compute and time budget during the rebuttal period, we have run this cross-backbone analysis on CWMB, which is the most computationally demanding benchmark. In the camera-ready version, we plan to extend these experiments to **Text2World** and **ByteSized32** as well, to further validate cross-backbone robustness.

---

> ### Author Response · Authors · 2025-11-26
> **Response to Reviewer aueF (4)**
>
> ### Q1. Error distribution of baselines vs Agent2World
>
> | Method                 | signature-mismatch | schema-mismatch | dynamics-error | non-deterministic | judgment-bug | invariant-violation |
> |------------------------|--------------------|-----------------|----------------|-------------------|--------------|----------------------|
> | WorldCoder (Last Turn) | 0                  | 0               | 3              | 0                 | 2            | 2                    |
> | GIF-MCTS (Last Turn)   | 0                  | 0               | 2              | 0                 | 3            | 2                    |
> | Agent2World-multi (Turn 1)        | 2                  | 1               | 8              | 0                 | 1            | 1                    |
> | Agent2World-multi (Turn 2)        | 2                  | 0               | 6              | 0                 | 2            | 0                    |
> | Agent2World-multi (Turn 3)        | 1                  | 0               | 2              | 0                 | 1            | 1                    |
>
> We conducted additional manual error analysis on the baseline methods. As shown in the table above (and also in Appendix E.3), in the final turn of both WorldCoder and GIF-MCTS, dynamics-error and invariant-violation errors dominate. This suggests that **static test cases** primarily enforce "surface-level" consistency, but are still insufficient in handling fine-grained environment dynamics and long-horizon invariants.
>
> In contrast, Agent2World-Multi starts from a much broader initial state in the first turn: it not only encounters interface and schema errors, but also exhibits a larger number of dynamics-related errors. This indicates that it begins the exploration in a much wider world-model hypothesis space. As the Testing Team conducts multiple rounds of test-repair cycles, the number of dynamics errors and invariant violations significantly decreases, demonstrating the effectiveness of proactive tester agents (Testing Team) in providing feedback.
>
> We also provide a case study on the **Ant-v4** forward-locomotion task in the initial manuscript of our paper (Appendix H, Line 1644)
>
> Specifically:
>
> (i) **State & sensing.** The *Baseline* exposes a flat 27-D observation, whereas we adopt a task-aligned layout that separates positions and velocities and can hide global $(x,y)$ by default. We additionally support contact forces for foot–ground cues. Health checks use the torso $z$ from the split state rather than a fixed slot in the flat vector. State restoration is consistent with each layout: the *Baseline* ingests a 27-D vector directly, whereas ours reconstructs split buffers from the current observation configuration.
>
> (ii) **Dynamics & orientation.** The *Baseline* updates orientation by adding quaternion noise followed by renormalization, and treats actions as noisy joint velocities. In contrast, we integrate damped joint accelerations and update attitude via $\dot{\mathbf{q}}=\tfrac{1}{2}\,{\omega}_q\otimes\mathbf{q}$ with renormalization. This more physically consistent pipeline—enabled by the split-state design—low-passes high-frequency actuation, reduces roll/pitch jitter, and yields more phase-coordinated gaits.
>
> (iii) **Control semantics & reward.** The *Baseline* hard-errors on out-of-range actions and uses a fixed forward-reward weight; forward progress is tracked by an external $x$ variable and reset noise is relatively small. Ours instead clips actions to $[-1,1]$, uses a tunable forward-reward weight, measures progress directly from torso $x$ in the state, and employs a different reset scale; an optional contact-cost term can be included when contact signals are enabled. Together, these choices stabilize the training signal and improve sample efficiency in practice.

---

### Official Review · Reviewer_XwvJ · 2025-10-31

**Soundness:** 2
**Presentation:** 4
**Contribution:** 2
**Rating:** 4
**Confidence:** 4

**Summary:**

This paper presents AGENT2WORLD, a system that tackles the challenge of automatically creating symbolic world models (like environment rules for a planner) just from natural language. Instead of using a fixed, step-by-step script, it uses a flexible team of AI agents. This team includes a deep researcher that browses the web to find missing details, a model developer that actually writes the code for the world model in PDDL or Python, and a Testing Team that runs unit tests and simulations to find bugs. The authors show that this approach works well, setting new state-of-the-art results on three different benchmarks.

**Strengths:**

- The multi-agent framework is well-designed, with clear specialization for each agent (research, development, and testing), which is validated by ablation studies showing each component's contribution.
- Strong empirical results are demonstrated, establishing new state-of-the-art performance across three different benchmarks, which cover both PDDL and executable code representations.

**Weaknesses:**

- Experiments rely exclusively on OpenAI's GPT-4.1-mini as the backbone LLM. It is unclear how dependent the framework's success is on this specific model. Have the authors tested other models to assess the generalizability of the agentic framework, or does the performance heavily rely on the capabilities of GPT-4.1?
- The novelty seems to stem from the composition of existing techniques (ReAct-style agents, RAG via web search, iterative refinement) into a multi-agent pipeline. Is the primary contribution this specific system design, or are there fundamental algorithmic contributions beyond this integration?
- The paper does not include a quantitative comparison against several other recent LLM-based agent or world-modeling frameworks, with the results being compared primarily to task-specific baselines or ablated versions of the system itself.

**Questions:**

See Weaknesses.

---

> ### Author Response · Authors · 2025-11-26
> **Response to Reviewer XwvJ (1)**
>
> Dear Reviewer XwvJ,
>
> Thank you for your thoughtful review and for highlighting both the clear specialization in our multi-agent framework and the strong empirical results. Building on your feedback, we have revised the manuscript and, in the following, address your main concerns:
> - Additional experiments with open-source llama-3.1-8b-instruct model to demonstrate the generalizability (W1, Section 4.2 & Table 2).
> - A more precise clarification of our contributions together with additional training-based experiments that further support our central ideas (W2, Section 5.6).
> - Broader comparisons against recent LLM-based agent and world-modeling frameworks (W3, Table 2).
>
> Please let us know if you have any further questions, and we can provide any additional clarifications to help finalize your assessment and rating of our paper.
>
>
> ---
>
>
> ### W1. Dependence on GPT-4.1-mini and generalizability to other models
>
> **A:** We agree that using a single backbone limits conclusions about generality. Our original experiments intentionally fixed all methods (baselines and Agent2World) to GPT-4.1-mini with identical decoding settings to isolate the effect of the *agentic framework* from raw model quality.
>
> To address your concern, we have added a second set of experiments on CWMB using a smaller open-source backbone, Llama-3.1-8B-Instruct[1], and updated Table 2 accordingly in the revised manuscript.  Concretely:
>
> * Swapping GPT-4.1-mini for Llama-3.1-8B-Instruct reduces absolute scores across *all* methods (as expected from a smaller model), but
> * The relative ranking is preserved: Agent2World-Multi (and its SFT variant) still outperforms WorldCoder and GIF-MCTS in **overall accuracy and normalized return**.
> * With Llama-3.1-8B-Instruct, our SFT variant Agent2World-Multi(SFT) improves overall normalized return **from 0.2070 (GIF-MCTS) to 0.3156** and overall accuracy from **0.2883 to 0.3360** on CWMB.
>
> We summarize the most relevant part of the new table below (overall metrics on CWMB, detailed results are presented in Section 4.4, Line 270):
>
> | Backbone                | Method                 | Overall Accuracy ↑ | Overall 𝓡 ↑ |
> |-------------------------|------------------------|--------------------|-------------|
> | GPT-4.1-mini            | Agent2World-Multi       | 0.5441             | 0.4811      |
> | Llama-3.1-8B-Instruct   | WorldCoder            | 0.2513             | 0.1642      |
> | Llama-3.1-8B-Instruct   | GIF-MCTS              | 0.2883             | 0.2070      |
> | Llama-3.1-8B-Instruct   | Agent2World-Multi       | 0.3150             | 0.2296      |
> | Llama-3.1-8B-Instruct   | **Agent2World-Multi(SFT)** | **0.3360**     | **0.3156**  |
>
> These results support our claim that **Agent2World is not tied to a single proprietary model**: even when we move to a smaller open-source backbone, the *same multi-agent architecture* consistently yields stronger world models than static workflows. In the rebuttal-time budget we were able to run this analysis on CWMB, as it is the most computationally demanding benchmark. We will extend the cross-backbone study to Text2World and ByteSized32 in the camera-ready version.
>
> ---
>
> **References**
>
> [1] Dubey, Abhimanyu, et al. "The llama 3 herd of models." arXiv e-prints (2024): arXiv-2407.

---

> ### Author Response · Authors · 2025-11-26
> **Response to Reviewer XwvJ (2)**
>
> ### W2. The novelty seems to stem from the composition of existing techniques (ReAct-style agents, RAG via web search, iterative refinement) into a multi-agent pipeline. Is the primary contribution this specific system design, or are there fundamental algorithmic contributions beyond this integration?
>
> **A:** Thank you for raising this concern. To answer your question:
>
> 1. What we do not claim: the ReAct-style backbone itself is not our primary contribution. We fully acknowledge that the ReAct-style tool-using backbone and the individual tools (web search, code execution, simulation, unit tests) are all based on prior work. Our ReAct-style single-agent variant (Agent2World-Single) is included as a strong baseline and ablation, not as a novelty claim. While it already outperforms prior workflow-based systems in our setting, we do not present “yet another ReAct variant” as the main contribution of the paper.
> 2. Our view is that the paper makes three related contributions within the setting of symbolic world-model generation, which go beyond a simple composition of tools:
>
>     **a.** **From scripted workflows to proactive agents.** Prior world-model–generation systems are typically based on scripted draft–repair pipelines: the control flow (draft, check, patch, repeat) is hard-coded by the designer and does not adapt to intermediate feedback. Additionally, the test cases are static. In contrast, Agent2World uses proactive agents that decide when and how to call tools (web search, code execution, simulation, unit tests, play) based on the current state of the world model. And using Testing Team to generate adaptive test cases and agent trajectories. This shift from hand-designed workflows to adaptive decision-making is crucial for discovering missing constraints and commonsense violations, and it is exactly what makes the system non-trivial beyond “putting ReAct plus some tools together”.
>
>     **b.** **Agent-in-the-loop world-model generation via simulation-based testing.** Beyond the architecture, we introduce an agent-in-the-loop world-model generation paradigm. The Model Developer interacts with a simulation-based Testing Team (including play agents), which turns rollouts in the environment into structured diagnostics and scalar feedback. We formalize this as an MDP where the Model Developer is the policy, and the Testing Team defines the reward (Section 5.6 Line 446). This is an **algorithmic contribution**: it specifies how an LLM agent should use simulation testers and play traces to iteratively improve a symbolic world model, rather than only using tools for one-shot problem solving.
>
>     **c.** **A clean, unified interface for both PDDL and code-based world models.** A key design choice is a clean, representation-agnostic interface shared by both PDDL-style and executable-code world models: the same roles (Deep Researcher → Model Developer → Testing Team → Play Agents) and the same message structure apply to both. This unified interface is not only conceptually neat; it is what makes the agent-in-the-loop view operational: the interaction traces produced by this interface can be reused directly as training data for world-model agents.
>
> 3. Building on this, we further demonstrate the value of the interface and the agent-in-the-loop formulation by adding a supervised fine-tuning (SFT) variant in the revision (**Section 5.6**): we roll out Agent2World-Multi, filter trajectories using Testing-Team / simulation feedback, and fine-tune the backbone on these traces. This improves both accuracy and normalized return over non-SFT baselines, showing that the same clean interface naturally supports learning, not just inference.
>
> | Method        | Overall Accuracy ↑ | Overall 𝓡 ↑ |
> |--------------|--------------------|-------------|
> | Agent2World-Multi        | 0.3150             | 0.2296      |
> | Agent2World-Multi (SFT)  | 0.3360             | 0.3156      |
> | Δ (absolute) | +0.0210            | +0.0860     |
>
> To directly answer your question:
> The primary contribution is not just a particular system wiring. It consists of
> (i) a move from scripted workflows to proactive agents tailored to symbolic world-model generation (**dynamic test cases and adaptive tool calling**),
> (ii) an agent-in-the-loop, simulation-based paradigm for world-model generation, and
> (iii) a unified interface that works for both PDDL and code-based world models and naturally enables training of world-model agents on the same interaction traces.

---

> ### Author Response · Authors · 2025-11-26
> **Response to Reviewer XwvJ (3)**
>
> ### W3. The paper does not include a quantitative comparison against several other recent LLM-based agent or world-modeling frameworks, with the results being compared primarily to task-specific baselines or ablated versions of the system itself.
>
> **A:** Thank you for pointing this out. We have revised the manuscript to include quantitative comparisons against several additional LLM-based agent and world-modeling baselines.
>
> Concretely, we now incorporate two widely used **test-time scaling** approaches:
>
> 1. **Best-of-N** [1]: This strategy is broadly used to improve LLM reasoning and was recently introduced by Yu et al. [2] for PDDL generation. In our experiments, we further generalize Best-of-N so that it can be applied to **both** PDDL generation and code-based world-model generation.
>
> 2. **Self-consistency** [3]: This method performs multi-sample reasoning followed by a voting step, and is commonly regarded as a particularly effective form of test-time scaling for LLM reasoning.
>
> In addition, we **strengthen the GIF-MCTS baseline** by integrating a **deep research agent**. Specifically, we first let the deep research agent gather information and produce an enhanced research report, and then feed this report into the original GIF-MCTS pipeline (denoted as GIF-MCTS\* in the revised paper).
>
> On CWMB, all of these enhanced LLM-based baselines still underperform our proposed method under the `gpt-4.1-mini` backbone. The table below summarizes the **overall** CWMB metrics, and the detailed results are presented in Table 2 in the revised manuscript:
>
> | Method            | Overall Accuracy ↑ | Overall 𝓡 ↑ |
> | ----------------- | ------------------ | ------------ |
> | Direct Generation | 0.4466             | 0.2620       |
> | Best-of-N         | 0.4257             | 0.3317       |
> | Self-consistency  | 0.4276             | 0.3076       |
> | GIF-MCTS*         | 0.4979             | 0.3649       |
> | **Ours (Multi)**  | **0.5441**         | **0.4811**   |
>
>
> Here, GIF-MCTS\* denotes the enhanced variant of GIF-MCTS equipped with the deep research agent.
>
> ---
>
> **References**
>
> [1] Stiennon, N., et al. *Learning to summarize with human feedback.* NeurIPS, 2020.
> [2] Yu, Z., et al. *Generating symbolic world models via test-time scaling of large language models.* arXiv:2502.04728, 2025.
> [3] Wang, X., et al. *Self-consistency improves chain of thought reasoning in language models.* arXiv:2203.11171, 2022.

---

### Author Response · Authors · 2025-11-26
**Updated Manuscript and Response to All Reviewers**

We sincerely thank all the reviewers for their valuable feedback and constructive comments. We are pleased to note the following positive aspects of our work: **a.** The clear design of our multi-agent framework was appreciated (R#XwvJ, R#aueF, R#m6C3). **b.** Our empirical results were recognized for demonstrating strong performance across multiple benchmarks (R#XwvJ, R#aueF, R#m6C3, R#J92f). **c.** The comprehensive evaluation, including ablation studies and error analysis, was well-received (R#aueF, R#m6C3).

We’ve conducted an **additional set of 5 experiments** (three baselines, one new open-source model, and experiments based on agent-in-the-loop training). We have also included new analysis experiments (Appendix I,J,E.3).
The manuscript has been revised and updated to address the reviewers' comments, with all changes highlighted in **red** in the new PDF. The updates are summarized as follows:

1. **Section 4.1, Line 211**: Added a description of the additional baselines, including testing-time scaling methods (best-of-n, self-consistency, and an enhanced variant of GIF-MCTS with deep research). (R#J92f,R#XwvJ)
2. **Section 4.2, Line 230**: Added implementation details for experiments involving the LLaMA-3.1-8b model.(R#XwvJ,R#aueF)
3. **Table 2, Line 280**: Added new experimental results, including those from the LLaMA 3.1 8b model and new baseline methods. (R#XwvJ,R#aueF)
4. **Section 5.6, Line 453**: Formalized the agent-in-the-loop agent training process.(R#J92f)
5. **Table 5, Line 486**: Improved results for agent-in-the-loop world model agent training.(R#J92f)
6. **Appendix I, Line 2454**: Data contamination analysis.(R#J92f,R#m6C3)
7. **Appendix J, Line 2479**: Efficiency analysis (token and time).(R#J92f,R#aueF,R#m6C3)
8. **Appendix K, Line 2537**: Details on how the training data was constructed. (R#J92f)
9. **Appendix E.3, Line 983**: Distribution of errors across baseline methods and a comparative analysis of our approach versus the baseline methods. (R#aueF)

We have responded to each of the reviewers' queries and comments below and provided clarifications where needed. We would like to express our sincere gratitude to all the reviewers for their valuable input in improving our manuscript. If further clarification is required to help improve our score, please feel free to let us know.

Thank you for your review!

---

### Meta-Review · Area_Chair_EdFq · 2026-01-06

**Summary:**

This paper proposes Agent2World, a novel paradigm that employs autonomous tool-augmented LLM-based agents to generate symbolic world models adaptively. This paper further provides a unified multi-agent framework with specialized agents: (i) a Deep Researcher agent performs knowledge synthesis by web searching to address specification gaps; (ii) a Model Developer implements executable world models; and (iii) a specialized Testing Team conducts evaluation-driven refinement via systematic unit testing and simulation-based validation.

**Reviewer Concerns:**

Reviewers have many concerns, including Limited novelty in methodology,  limited Scalability, and high cost,  limited base models,  and token efficiency, as well as unclear experimental settings.

The authors provide additional clarification, but some of the reviewers' concerns have not been well addressed.

**Reviewer Scores:**

All reviewers gave scores below the acceptance threshold.  After rebuttal,  the Reviewer m6C3 may raise the score from 4 to 6.  The Reviewer J92f chose to keep the original score.  The other reviewers may keep the score.

---

### Decision · Program_Chairs · 2026-01-26

Reject